# A Variational Inequality Perspective on Generative Adversarial Networks

**Gauthier Gidel**[1,*]    **Hugo Berard**[1,3,*]    **Gaëtan Vignoud**[1]    **Pascal Vincent**[1,2,3]

[1]Mila & DIRO, University of Montreal   [2]Canada CIFAR AI Chair
**Simon Lacoste-Julien**[1,2]   [3]Facebook Artificial Intelligence Research

## Abstract

Generative adversarial networks (GANs) form a generative modeling approach known for producing appealing samples, but they are notably difficult to train. One common way to tackle this issue has been to propose new formulations of the GAN objective. Yet, surprisingly few studies have looked at optimization methods designed for this adversarial training. In this work, we cast GAN optimization problems in the general variational inequality framework. Tapping into the mathematical programming literature, we counter some common misconceptions about the difficulties of saddle point optimization and propose to extend techniques designed for variational inequalities to the training of GANs. We apply *averaging*, *extrapolation* and a computationally cheaper variant that we call *extrapolation from the past* to the stochastic gradient method (SGD) and Adam.

## 1 Introduction

Generative adversarial networks (GANs) (Goodfellow et al., 2014) form a generative modeling approach known for producing realistic natural images (Karras et al., 2018) as well as high quality super-resolution (Ledig et al., 2017) and style transfer (Zhu et al., 2017). Nevertheless, GANs are also known to be difficult to train, often displaying an unstable behavior (Goodfellow, 2016). Much recent work has tried to tackle these training difficulties, usually by proposing new formulations of the GAN objective (Nowozin et al., 2016; Arjovsky et al., 2017). Each of these formulations can be understood as a two-player game, in the sense of game theory (Von Neumann and Morgenstern, 1944), and can be addressed as a variational inequality problem (VIP) (Harker and Pang, 1990), a framework that encompasses traditional saddle point optimization algorithms (Korpelevich, 1976).

Solving such GAN games is traditionally approached by running variants of stochastic gradient descent (SGD) initially developed for optimizing supervised neural network objectives. Yet it is known that for some games (Goodfellow, 2016, §8.2) SGD exhibits oscillatory behavior and fails to converge. This oscillatory behavior, which does not arise from stochasticity, highlights a fundamental problem: while a direct application of basic gradient descent is an appropriate method for regular minimization problems, it is *not* a sound optimization algorithm for the kind of two-player games of GANs. This constitutes a fundamental issue for GAN training, and calls for the use of more principled methods with more reassuring convergence guarantees.

**Contributions.**   We point out that multi-player games can be cast as *variational inequality problems* (VIPs) and consequently the same applies to any GAN formulation posed as a minimax or non-zero-sum game. We present two techniques from this literature, namely *averaging* and *extrapolation*, widely used to solve VIPs but which have not been explored in the context of GANs before.[1]

We extend standard GAN training methods such as SGD or Adam into variants that incorporate these techniques (Alg. 4 is new). We also explain that the oscillations of basic SGD for GAN training previously noticed  (Goodfellow, 2016) can be explained by standard variational inequality optimization results and we illustrate how *averaging* and *extrapolation* can fix this issue.

---

*Equal contribution, correspondence to `firstname.lastname@umontreal.ca`.

[1]The preprints for (Mertikopoulos et al., 2019) and (Yazıcı et al., 2019), which respectively explored extrapolation and averaging for GANs, appeared after our initial preprint. See also the related work section §6.

We introduce a technique, called *extrapolation from the past*, that only requires one gradient computation per update compared to *extrapolation* which requires to compute the gradient twice, rediscovering, with a VIP perspective, a particular case of optimistic mirror descent (Rakhlin and Sridharan, 2013). We *prove its convergence* for strongly monotone operators and in the stochastic VIP setting.

Finally, we test these techniques in the context of GAN training. We observe a 4-6% improvement over Miyato et al. (2018) on the inception score and the Fréchet inception distance on the CIFAR-10 dataset using a WGAN-GP (Gulrajani et al., 2017) and a ResNet generator.[2]

**Outline.** §2 presents the background on GAN and optimization, and shows how to cast this optimization as a VIP. §3 presents standard techniques and *extrapolation from the past* to optimize variational inequalities in a batch setting. §4 considers these methods in the *stochastic* setting, yielding three corresponding variants of SGD, and provides their respective convergence rates. §5 develops how to combine these techniques with already existing algorithms. §6 discusses the related work and §7 presents experimental results.

## 2 GAN OPTIMIZATION AS A VARIATIONAL INEQUALITY PROBLEM

### 2.1 GAN FORMULATIONS

The purpose of generative modeling is to generate samples from a distribution $q_{\boldsymbol{\theta}}$ that matches best the true distribution $p$ of the data. The generative adversarial network training strategy can be understood as a *game* between two players called *generator* and *discriminator*. The former produces a sample that the latter has to classify between real or fake data. The final goal is to build a generator able to produce sufficiently realistic samples to fool the discriminator.

In the original GAN paper (Goodfellow et al., 2014), the GAN objective is formulated as a *zero-sum game* where the cost function of the discriminator $D_{\boldsymbol{\varphi}}$ is given by the negative log-likelihood of the binary classification task between real or fake data generated from $q_{\boldsymbol{\theta}}$ by the generator,

$$\min_{\boldsymbol{\theta}} \max_{\boldsymbol{\varphi}} \mathcal{L}(\boldsymbol{\theta}, \boldsymbol{\varphi}) \quad \text{where} \quad \mathcal{L}(\boldsymbol{\theta}, \boldsymbol{\varphi}) \stackrel{\text{def}}{=} -\mathbb{E}_{\mathbf{x} \sim p}[\log D_{\boldsymbol{\varphi}}(\mathbf{x})] - \mathbb{E}_{\mathbf{x}' \sim q_{\boldsymbol{\theta}}}[\log(1 - D_{\boldsymbol{\varphi}}(\mathbf{x}'))]. \quad (1)$$

However Goodfellow et al. (2014) recommends to use in practice a second formulation, called *non-saturating GAN*. This formulation is a *non-zero-sum game* where the aim is to jointly minimize:

$$\mathcal{L}_G(\boldsymbol{\theta}, \boldsymbol{\varphi}) \stackrel{\text{def}}{=} -\mathbb{E}_{\mathbf{x}' \sim q_{\boldsymbol{\theta}}} \log D_{\boldsymbol{\varphi}}(\mathbf{x}') \quad \text{and} \quad \mathcal{L}_D(\boldsymbol{\theta}, \boldsymbol{\varphi}) \stackrel{\text{def}}{=} -\mathbb{E}_{\mathbf{x} \sim p} \log D_{\boldsymbol{\varphi}}(\mathbf{x}) - \mathbb{E}_{\mathbf{x}' \sim q_{\boldsymbol{\theta}}} \log(1 - D_{\boldsymbol{\varphi}}(\mathbf{x}')). \quad (2)$$

The dynamics of this formulation has the same *stationary points* as the zero-sum one (1) but is claimed to provide "much stronger gradients early in learning" (Goodfellow et al., 2014) .

### 2.2 EQUILIBRIUM

The minimax formulation (1) is theoretically convenient because a large literature on games studies this problem and provides guarantees on the existence of equilibria. Nevertheless, practical considerations lead the GAN literature to consider a different objective for each player as formulated in (2). In that case, the *two-player game problem* (Von Neumann and Morgenstern, 1944) consists in finding the following *Nash equilibrium*:

$$\boldsymbol{\theta}^* \in \arg\min_{\boldsymbol{\theta} \in \Theta} \mathcal{L}_G(\boldsymbol{\theta}, \boldsymbol{\varphi}^*) \quad \text{and} \quad \boldsymbol{\varphi}^* \in \arg\min_{\boldsymbol{\varphi} \in \Phi} \mathcal{L}_D(\boldsymbol{\theta}^*, \boldsymbol{\varphi}). \quad (3)$$

Only when $\mathcal{L}_G = -\mathcal{L}_D$ is the game called a *zero-sum game* and (3) can be formulated as a minimax problem. One important point to notice is that the two optimization problems in (3) are *coupled* and have to be considered *jointly* from an optimization point of view.

Standard GAN objectives are non-convex (i.e. each cost function is non-convex), and thus such (pure) equilibria may not exist. As far as we know, not much is known about the existence of these equilibria for non-convex losses (see Heusel et al. (2017) and references therein for some results). In

---

[2]Code available at `https://gauthiergidel.github.io/projects/vip-gan.html`.

our theoretical analysis in §4, our assumptions (monotonicity (24) of the operator and convexity of the constraint set) imply the existence of an equilibrium.

In this paper, we focus on ways to optimize these games, assuming that an equilibrium exists. As is often standard in non-convex optimization, we also focus on finding points satisfying the necessary *stationary conditions*. As we mentioned previously, one difficulty that emerges in the optimization of such games is that the two different cost functions of (3) have to be minimized jointly in $\boldsymbol{\theta}$ and $\boldsymbol{\varphi}$. Fortunately, the optimization literature has for a long time studied so-called *variational inequality problems*, which generalize the stationary conditions for two-player game problems.

### 2.3 VARIATIONAL INEQUALITY PROBLEM FORMULATION

We first consider the local necessary conditions that characterize the solution of the *smooth* two-player game (3), defining *stationary points*, which will motivate the definition of a variational inequality. In the unconstrained setting, a *stationary point* is a couple $(\boldsymbol{\theta}^*, \boldsymbol{\varphi}^*)$ with zero gradient:

$$\|\nabla_{\boldsymbol{\theta}} \mathcal{L}_G(\boldsymbol{\theta}^*, \boldsymbol{\varphi}^*)\| = \|\nabla_{\boldsymbol{\varphi}} \mathcal{L}_D(\boldsymbol{\theta}^*, \boldsymbol{\varphi}^*)\| = 0 \,. \tag{4}$$

When constraints are present,[3] a *stationary point* $(\boldsymbol{\theta}^*, \boldsymbol{\varphi}^*)$ is such that the directional derivative of each cost function is non-negative in any feasible direction (i.e. there is no feasible descent direction):

$$\nabla_{\boldsymbol{\theta}} \mathcal{L}_G(\boldsymbol{\theta}^*, \boldsymbol{\varphi}^*)^\top (\boldsymbol{\theta} - \boldsymbol{\theta}^*) \geq 0 \quad \text{and} \quad \nabla_{\boldsymbol{\varphi}} \mathcal{L}_D(\boldsymbol{\theta}^*, \boldsymbol{\varphi}^*)^\top (\boldsymbol{\varphi} - \boldsymbol{\varphi}^*) \geq 0 \,, \ \forall (\boldsymbol{\theta}, \boldsymbol{\varphi}) \in \Theta \times \Phi. \tag{5}$$

Defining $\boldsymbol{\omega} \stackrel{\text{def}}{=} (\boldsymbol{\theta}, \boldsymbol{\varphi})$, $\boldsymbol{\omega}^* \stackrel{\text{def}}{=} (\boldsymbol{\theta}^*, \boldsymbol{\varphi}^*)$, $\Omega \stackrel{\text{def}}{=} \Theta \times \Phi$, Eq. (5) can be compactly formulated as:

$$F(\boldsymbol{\omega}^*)^\top (\boldsymbol{\omega} - \boldsymbol{\omega}^*) \geq 0 \,, \ \forall \boldsymbol{\omega} \in \Omega \quad \text{where} \quad F(\boldsymbol{\omega}) \stackrel{\text{def}}{=} \begin{bmatrix} \nabla_{\boldsymbol{\theta}} \mathcal{L}_G(\boldsymbol{\theta}, \boldsymbol{\varphi}) & \nabla_{\boldsymbol{\varphi}} \mathcal{L}_D(\boldsymbol{\theta}, \boldsymbol{\varphi}) \end{bmatrix}^\top . \tag{6}$$

These stationary conditions can be generalized to any continuous vector field: let $\Omega \subset \mathbb{R}^d$ and $F : \Omega \to \mathbb{R}^d$ be a continuous mapping. The *variational inequality problem* (Harker and Pang, 1990) (depending on $F$ and $\Omega$) is:

$$\text{find } \boldsymbol{\omega}^* \in \Omega \quad \text{such that} \quad F(\boldsymbol{\omega}^*)^\top (\boldsymbol{\omega} - \boldsymbol{\omega}^*) \geq 0 \,, \ \forall \boldsymbol{\omega} \in \Omega \,. \tag{VIP}$$

We call *optimal set* the set $\Omega^*$ of $\boldsymbol{\omega} \in \Omega$ verifying (VIP). The intuition behind it is that any $\boldsymbol{\omega}^* \in \Omega^*$ is a *fixed point* of the *constrained* dynamic of $F$ (constrained to $\Omega$).

We have thus showed that both saddle point optimization and non-zero sum game optimization, which encompass the large majority of GAN variants proposed in the literature, can be cast as VIPs. In the next section, we turn to suitable optimization techniques for such problems.

## 3 OPTIMIZATION OF VARIATIONAL INEQUALITIES (BATCH SETTING)

Let us begin by looking at techniques that were developed in the optimization literature to solve VIPs. We present the intuitions behind them as well as their performance on a simple bilinear problem (see Fig. 1). Our goal is to provide mathematical insights on *averaging* (§3.1) and *extrapolation* (§3.2) and propose a novel variant of the extrapolation technique that we called *extrapolation from the past* (§3.3). We consider the batch setting, i.e., the operator $F(\boldsymbol{\omega})$ defined in Eq. 6 yields an exact full gradient. We present extensions of these techniques to the stochastic setting later in §4.

The two standard methods studied in the VIP literature are the *gradient method* (Bruck, 1977) and the *extragradient method* (Korpelevich, 1976). The iterates of the basic gradient method are given by $\boldsymbol{\omega}_{t+1} = P_\Omega[\boldsymbol{\omega}_t - \eta F(\boldsymbol{\omega}_t)]$ where $P_\Omega[\cdot]$ is the *projection onto the constraint set* (if constraints are present) associated to (VIP). These iterates are known to converge linearly under an additional assumption on the operator[4] (Chen and Rockafellar, 1997), but oscillate for a bilinear operator as shown in Fig. 1. On the other hand, the *uniform average* of these iterates converge for any bounded monotone operator with a $O(1/\sqrt{t})$ rate (Nedić and Ozdaglar, 2009), motivating the presentation of *averaging* in §3.1. By contrast, the *extragradient method* (extrapolated gradient) does not require any averaging to converge for monotone operators (in the batch setting), and can even converge at the faster $O(1/t)$ rate (Nesterov, 2007). The idea of this method is to compute a lookahead step (see intuition on *extrapolation* in §3.2) in order to compute a more stable direction to follow.

---

[3]An example of constraint for GANs is to clip the parameters of the discriminator (Arjovsky et al., 2017).
[4]Strong monotonicity, a generalization of strong convexity. See §A.

## 3.1 Averaging

More generally, we consider a *weighted averaging* scheme with weights $\rho_t \geq 0$. This *weighted averaging* scheme have been proposed for the first time for (batch) VIP by Bruck (1977),

$$\bar{\boldsymbol{\omega}}_T \stackrel{\text{def}}{=} \frac{\sum_{t=0}^{T-1} \rho_t \boldsymbol{\omega}_t}{S_T}, \quad S_T \stackrel{\text{def}}{=} \sum_{t=0}^{T-1} \rho_t. \tag{7}$$

Averaging schemes can be efficiently implemented in an online fashion noticing that,

$$\bar{\boldsymbol{\omega}}_T = (1 - \tilde{\rho}_T)\bar{\boldsymbol{\omega}}_{T-1} + \tilde{\rho}_T \boldsymbol{\omega}_T \quad \text{where} \quad 0 \leq \tilde{\rho}_T \leq 1. \tag{8}$$

For instance, setting $\tilde{\rho}_T = \frac{1}{T}$ yields *uniform averaging* ($\rho_t = 1$) and $\tilde{\rho}_T = 1 - \beta < 1$ yields *geometric averaging*, also known as *exponential moving averaging* ($\rho_t = \beta^{T-t}$, $1 \leq t \leq T$). Averaging is experimentally compared with the other techniques presented in this section in Fig. 1.

In order to illustrate how averaging tackles the oscillatory behavior in game optimization, we consider a toy example where the discriminator and the generator are linear: $D_{\boldsymbol{\varphi}}(\mathbf{x}) = \boldsymbol{\varphi}^T \mathbf{x}$ and $G_{\boldsymbol{\theta}}(\mathbf{z}) = \boldsymbol{\theta}\mathbf{z}$ (implicitly defining $q_{\boldsymbol{\theta}}$). By substituting these expressions in the WGAN objective,[5] we get the following bilinear objective:

$$\min_{\boldsymbol{\theta} \in \Theta} \max_{\boldsymbol{\varphi} \in \Phi, ||\boldsymbol{\varphi}|| \leq 1} \boldsymbol{\varphi}^T \mathbb{E}[\mathbf{x}] - \boldsymbol{\varphi}^T \boldsymbol{\theta} \mathbb{E}[\mathbf{z}]. \tag{9}$$

A similar task was presented by Nagarajan and Kolter (2017) where they consider a quadratic discriminator instead of a linear one, and show that gradient descent is not necessarily asymptotically stable. The bilinear objective has been extensively used (Goodfellow, 2016; Mescheder et al., 2018; Yadav et al., 2018) to highlight the difficulties of gradient descent for saddle point optimization. Yet, ways to cope with this issue have been proposed decades ago in the context of mathematical programming. For illustrating the properties of the methods of interest, we will study their behavior in the rest of §3 on a simple *unconstrained* unidimensional version of Eq. 9 (this behavior can be generalized to general multidimensional bilinear examples, see §B.3):

$$\min_{\theta \in \mathbb{R}} \max_{\phi \in \mathbb{R}} \quad \theta \cdot \phi \quad \text{and} \quad (\theta^*, \phi^*) = (0, 0). \tag{10}$$

The operator associated with this minimax game is $F(\theta, \phi) = (\phi, -\theta)$. There are several ways to compute the discrete updates of this dynamics. The two most common ones are the *simultaneous* and the *alternating* gradient update rules,

Simultaneous update: $\begin{cases} \theta_{t+1} = \theta_t - \eta\phi_t \\ \phi_{t+1} = \phi_t + \eta\theta_t \end{cases}$, Alternating update: $\begin{cases} \theta_{t+1} = \theta_t - \eta\phi_t \\ \phi_{t+1} = \phi_t + \eta\theta_{t+1} \end{cases}$. (11)

Interestingly, these two choices give rise to completely different behaviors. The norm of the *simultaneous* updates diverges geometrically, whereas the alternating iterates are bounded but do not converge to the equilibrium. As a consequence, their respective uniform average have a different behavior, as highlighted in the following proposition (proof in §B.1 and generalization in §B.3):

**Proposition 1.** *The* simultaneous *iterates diverge geometrically and the* alternating *iterates defined in* (11) *are bounded but do not converge to 0 as*

Simultaneous: $\theta_{t+1}^2 + \phi_{t+1}^2 = (1 + \eta^2)(\theta_t^2 + \phi_t^2)$, Alternating: $\theta_t^2 + \phi_t^2 = \Theta(\theta_0^2 + \phi_0^2)$ (12)

*where* $u_t = \Theta(v_t) \Leftrightarrow \exists \alpha, \beta, t_0 > 0$ *such that* $\forall t \geq t_0, \alpha v_t \leq u_t \leq \beta v_t$.

*The uniform average* $(\bar{\theta}_t, \bar{\phi}_t) \stackrel{\text{def}}{=} \frac{1}{t} \sum_{s=0}^{t-1}(\theta_s, \phi_s)$ *of the* simultaneous *updates (resp. the* alternating *updates) diverges (resp. converges to 0) as,*

Simultaneous: $\bar{\theta}_t^2 + \bar{\phi}_t^2 = \Theta\left(\frac{\theta_0^2 + \phi_0^2}{\eta^2 t^2}(1 + \eta^2)^t\right)$, Alternating: $\bar{\theta}_t^2 + \bar{\phi}_t^2 = \Theta\left(\frac{\theta_0^2 + \phi_0^2}{\eta^2 t^2}\right)$. (13)

This sublinear convergence result, proved in §B, underlines the benefits of averaging when the sequence of iterates is bounded (i.e. for *alternating* update rule). When the sequence of iterates is not bounded (i.e. for *simultaneous* updates) averaging fails to ensure convergence. This theorem also shows how *alternating* updates may have better convergence properties than *simultaneous* updates.

---

[5]Wasserstein GAN (WGAN) proposed by Arjovsky et al. (2017) boils down to the following minimax formulation: $\min_{\boldsymbol{\theta} \in \Theta} \max_{\boldsymbol{\varphi} \in \Phi, ||D_{\boldsymbol{\varphi}}||_L \leq 1} \mathbb{E}_{\mathbf{x} \sim p}[D_{\boldsymbol{\varphi}}(\mathbf{x})] - \mathbb{E}_{\mathbf{x}' \sim q_{\boldsymbol{\theta}}}[D_{\boldsymbol{\varphi}}(\mathbf{x}')]$.

## 3.2 EXTRAPOLATION

Another technique used in the variational inequality literature to prevent oscillations is *extrapolation*. This concept is anterior to the extragradient method since Korpelevich (1976) mentions that the idea of *extrapolated* "prices" to give "stability" had been already formulated by Polyak (1963, Chap. II). The idea behind this technique is to compute the gradient at an (extrapolated) point different from the current point from which the update is performed, stabilizing the dynamics:

$$\text{Compute extrapolated point: } \boldsymbol{\omega}_{t+1/2} = P_\Omega[\boldsymbol{\omega}_t - \eta F(\boldsymbol{\omega}_t)], \tag{14}$$

$$\text{Perform update step: } \boldsymbol{\omega}_{t+1} = P_\Omega[\boldsymbol{\omega}_t - \eta F(\boldsymbol{\omega}_{t+1/2})]. \tag{15}$$

Note that, even in the *unconstrained case*, this method is intrinsically different from Nesterov's momentum[6] (Nesterov, 1983, Eq. 2.2.9) because of this lookahead step for the gradient computation:

$$\text{Nesterov's method: } \boldsymbol{\omega}_{t+1/2} = \boldsymbol{\omega}_t - \eta F(\boldsymbol{\omega}_t), \qquad \boldsymbol{\omega}_{t+1} = \boldsymbol{\omega}_{t+1/2} + \beta(\boldsymbol{\omega}_{t+1/2} - \boldsymbol{\omega}_t). \tag{16}$$

Nesterov's method does not converge when trying to optimize (10). One intuition of why *extrapolation* has better convergence properties than the standard gradient method comes from Euler's integration framework. Indeed, to first order, we have $\boldsymbol{\omega}_{t+1/2} \approx \boldsymbol{\omega}_{t+1} + o(\eta)$ and consequently, the update step (15) can be interpreted as a first order approximation to an *implicit method* step:

$$\text{Implicit step: } \boldsymbol{\omega}_{t+1} = \boldsymbol{\omega}_t - \eta F(\boldsymbol{\omega}_{t+1}). \tag{17}$$

*Implicit methods* are known to be more stable and to benefit from better convergence properties (Atkinson, 2003) than *explicit methods*, e.g., in §B.2 we show that (17) on (10) converges *for any $\eta$*. Though, they are usually not practical since they require to solve a potentially non-linear system at each step. Going back to the simplified WGAN toy example (10) from §3.1, we get the following update rules:

$$\text{Implicit: } \begin{cases} \theta_{t+1} = \theta_t - \eta\phi_{t+1} \\ \phi_{t+1} = \phi_t + \eta\theta_{t+1} \end{cases}, \qquad \text{Extrapolation: } \begin{cases} \theta_{t+1} = \theta_t - \eta(\phi_t + \eta\theta_t) \\ \phi_{t+1} = \phi_t + \eta(\theta_t - \eta\phi_t) \end{cases}. \tag{18}$$

In the following proposition, we see that for $\eta < 1$, the respective convergence rates of the *implicit method* and *extrapolation* are highly similar. Keeping in mind that the latter has the major advantage of being more practical, this proposition clearly underlines the benefits of *extrapolation*. Note that Prop. 1 and 2 generalize to general unconstrained bilinear game (more details and proof in §B.3),

**Proposition 2.** *The squared norm of the iterates $N_t^2 \overset{def}{=} \theta_t^2 + \phi_t^2$, where the update rule of $\theta_t$ and $\phi_t$ are defined in (18), decreases geometrically for any $\eta < 1$ as,*

$$\text{Implicit: } N_{t+1}^2 = \left(1 - \eta^2 + \eta^4 + \mathcal{O}(\eta^6)\right)N_t^2, \quad \text{Extrapolation: } N_{t+1}^2 = (1 - \eta^2 + \eta^4)N_t^2. \tag{19}$$

## 3.3 EXTRAPOLATION FROM THE PAST

One issue with extrapolation is that the algorithm "wastes" a gradient (14). Indeed we need to compute the gradient at two different positions for every single update of the parameters. We thus propose a technique that we call *extrapolation from the past* that only requires a single gradient computation per update. The idea is to store and re-use the extrapolated gradient for the extrapolation:

$$\text{Extrapolation from the past: } \boldsymbol{\omega}_{t+1/2} = P_\Omega[\boldsymbol{\omega}_t - \eta F(\boldsymbol{\omega}_{t-1/2})] \tag{20}$$

$$\text{Perform update step: } \boldsymbol{\omega}_{t+1} = P_\Omega[\boldsymbol{\omega}_t - \eta F(\boldsymbol{\omega}_{t+1/2})] \text{ and store: } F(\boldsymbol{\omega}_{t+1/2}) \tag{21}$$

The same update scheme was proposed by Chiang et al. (2012, Alg. 1) in the context of online convex optimization and generalized by Rakhlin and Sridharan (2013) for general online learning. *Without projection*, (20) and (21) reduce to the optimistic mirror descent described by Daskalakis et al. (2018):

$$\text{Optimistic mirror descent (OMD): } \boldsymbol{\omega}_{t+1/2} = \boldsymbol{\omega}_{t-1/2} - 2\eta F(\boldsymbol{\omega}_{t-1/2}) + \eta F(\boldsymbol{\omega}_{t-3/2}) \tag{22}$$

We rediscovered this technique from a different perspective: it was motivated by VIP and inspired from the extragradient method. Using the VIP point of view, we are able to prove a linear convergence rate for *extrapolation from the past* (see details and proof of Theorem 1 in §B.4). We also provide results for a stochastic version in §4. In comparison to the results from Daskalakis et al. (2018) that

---

[6]Sutskever (2013, §7.2) showed the equivalence between "standard momentum" and Nesterov's formulation.

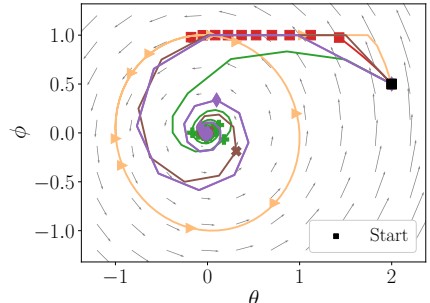

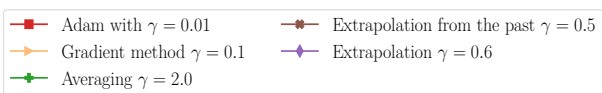

Figure 1: Comparison of the basic gradient method (as well as Adam) with the techniques presented in §3 on the optimization of (9). Only the algorithms advocated in this paper (Averaging, Extrapolation and Extrapolation from the past) converge quickly to the solution. Each marker represents 20 iterations. We compare these algorithms on a non-convex objective in §G.1.

| **Algorithm 1** AvgSGD | **Algorithm 2** AvgExtraSGD | **Algorithm 3** AvgPastExtraSGD |
|---|---|---|
| Let $\boldsymbol{\omega}_0 \in \Omega$ | **for** $t = 0 \dots T - 1$ **do** | Let $\boldsymbol{\omega}_0 \in \Omega$ |
| **for** $t = 0 \dots T - 1$ **do** | $\quad \xi_t, \xi'_t \sim P \quad$ *(mini-batches)* | **for** $t = 0 \dots T - 1$ **do** |
| $\quad \xi_t \sim P \quad$ *(mini-batch)* | $\quad \boldsymbol{d}_t \leftarrow F(\boldsymbol{\omega}_t, \xi_t)$ | $\quad \xi_t \sim P \quad$ *(mini-batch)* |
| $\quad \boldsymbol{d}_t \leftarrow F(\boldsymbol{\omega}_t, \xi_t)$ | $\quad \boldsymbol{\omega}'_t \leftarrow P_\Omega[\boldsymbol{\omega}_t - \eta_t \boldsymbol{d}_t]$ | $\quad \boldsymbol{\omega}'_t \leftarrow P_\Omega[\boldsymbol{\omega}_t - \eta_t \boldsymbol{d}_{t-1}]$ |
| $\quad \boldsymbol{\omega}_{t+1} \leftarrow P_\Omega[\boldsymbol{\omega}_t - \eta_t \boldsymbol{d}_t]$ | $\quad \boldsymbol{d}'_t \leftarrow F(\boldsymbol{\omega}'_t, \xi'_t)$ | $\quad \boldsymbol{d}_t \leftarrow F(\boldsymbol{\omega}'_t, \xi_t)$ |
| **end for** | $\quad \boldsymbol{\omega}_{t+1} \leftarrow P_\Omega[\boldsymbol{\omega}_t - \eta_t \boldsymbol{d}'_t]$ | $\quad \boldsymbol{\omega}_{t+1} \leftarrow P_\Omega[\boldsymbol{\omega}_t - \eta_t \boldsymbol{d}_t]$ |
| Return $\bar{\boldsymbol{\omega}}_T \leftarrow \frac{\sum_{t=0}^{T-1} \eta_t \boldsymbol{\omega}_t}{\sum_{t=0}^{T-1} \eta_t}$ | **end for** | **end for** |
| | Return $\bar{\boldsymbol{\omega}}_T \leftarrow \frac{\sum_{t=0}^{T-1} \eta_t \boldsymbol{\omega}'_t}{\sum_{t=0}^{T-1} \eta_t}$ | Return $\bar{\boldsymbol{\omega}}_T \leftarrow \frac{\sum_{t=0}^{T-1} \eta_t \boldsymbol{\omega}'_t}{\sum_{t=0}^{T-1} \eta_t}$ |

Figure 2: Three variants of SGD computing $T$ updates, using the techniques introduced in §3.

hold only for a bilinear objective, we provide a faster convergence rate (linear vs sublinear) on the last iterate for a general (strongly monotone) operator $F$ and any projection on a convex $\Omega$. One thing to notice is that the operator of a bilinear objective is *not* strongly monotone, but in that case one can use the standard extrapolation method (14) which converges linearly for a (constrained or not) bilinear game (Tseng, 1995, Cor. 3.3).

**Theorem 1** (Linear convergence of *extrapolation from the past*). *If $F$ is $\mu$-strongly monotone (see §A for the definition of strong monotonicity) and $L$-Lipschitz, then the updates (20) and (21) with $\eta = \frac{1}{4L}$ provide linearly converging iterates,*

$$\|\boldsymbol{\omega}_t - \boldsymbol{\omega}^*\|_2^2 \leq \left(1 - \frac{\mu}{4L}\right)^t \|\boldsymbol{\omega}_0 - \boldsymbol{\omega}^*\|_2^2, \quad \forall t \geq 0. \tag{23}$$

## 4 OPTIMIZATION OF VIP WITH STOCHASTIC GRADIENTS

In this section, we consider extensions of the techniques presented in §3 to the context of a *stochastic* operator, i.e., we no longer have access to the exact gradient $F(\boldsymbol{\omega})$ but to an unbiased *stochastic* estimate of it, $F(\boldsymbol{\omega}, \xi)$, where $\xi \sim P$ and $F(\boldsymbol{\omega}) := \mathbb{E}_{\xi \sim P}[F(\boldsymbol{\omega}, \xi)]$. It is motivated by GAN training where we only have access to a finite sample estimate of the expected gradient, computed on a mini-batch. For GANs, $\xi$ is a mini-batch of points coming from the true data distribution $p$ and the generator distribution $q_{\boldsymbol{\theta}}$.

For our analysis, we require at least one of the two following assumptions on the stochastic operator:

**Assumption 1.** *Bounded variance by $\sigma^2$: $\mathbb{E}_\xi[\|F(\boldsymbol{\omega}) - F(\boldsymbol{\omega}, \xi)\|^2] \leq \sigma^2$, $\forall \boldsymbol{\omega} \in \Omega$.*

**Assumption 2.** *Bounded expected squared norm by $M^2$: $\mathbb{E}_\xi[\|F(\boldsymbol{\omega}, \xi)\|^2] \leq M^2$, $\forall \boldsymbol{\omega} \in \Omega$.*

Assump. 1 is standard in stochastic variational analysis, while Assump. 2 is a stronger assumption sometimes made in stochastic convex optimization. To illustrate how strong Assump. 2 is, note that it does not hold for an unconstrained bilinear objective like in our example (10) in §3. It is thus mainly reasonable for bounded constraint sets. Note that in practice we have $\sigma \ll M$.

We now present and analyze three algorithms that are variants of SGD that are appropriate to solve (VIP). The first one Alg. 1 (AvgSGD) is the stochastic extension of the gradient method for solving (VIP); Alg. 2 (AvgExtraSGD) uses *extrapolation* and Alg. 3 (AvgPastExtraSGD) uses *extrapolation from the past*. A fourth variant that re-use the mini-batch for the extrapolation step (ReExtraSGD, Alg. 5) is described in §D. These four algorithms return an *average* of the iterates (typical in stochastic setting). The proofs of the theorems presented in this section are in §F.

To handle constraints such as parameter clipping (Arjovsky et al., 2017), we gave a *projected* version of these algorithms, where $P_\Omega[\omega']$ denotes the projection of $\omega'$ onto $\Omega$ (see §A). Note that when $\Omega = \mathbb{R}^d$, the projection is the identity mapping (unconstrained setting). In order to prove the convergence of these four algorithms, we will assume that $F$ is monotone:

$$(F(\omega) - F(\omega'))^\top (\omega - \omega') \geq 0 \quad \forall \omega, \omega' \in \Omega. \tag{24}$$

If $F$ can be written as (6), it implies that the cost functions are convex.[7] Note however that general GANs parametrized with neural networks lead to non-monotone VIPs.

**Assumption 3.** *F is monotone and $\Omega$ is a compact convex set, such that $\max_{\omega,\omega' \in \Omega} \|\omega - \omega'\|^2 \leq R^2$.*
In that setting the quantity $g(\omega^*) := \max_{\omega \in \Omega} F(\omega)^\top (\omega^* - \omega)$ is well defined and is equal to 0 if and only if $\omega^*$ is a solution of (VIP). Moreover, if we are optimizing a *zero-sum game*, we have $\omega = (\theta, \varphi)$, $\Omega = \Theta \times \Phi$ and $F(\theta, \varphi) = [\nabla_\theta \mathcal{L}(\theta, \varphi) \ -\nabla_\varphi \mathcal{L}(\theta, \varphi)]^\top$. Hence, the quantity $h(\theta^*, \varphi^*) := \max_{\varphi \in \Phi} \mathcal{L}(\theta^*, \varphi) - \min_{\theta \in \Theta} \mathcal{L}(\theta, \varphi^*)$ is well defined and equal to 0 if and only if $(\theta^*, \varphi^*)$ is a *Nash equilibrium* of the game. The two functions $g$ and $h$ are called *merit functions* (more details on the concept of *merit functions* in §C). In the following, we call,

$$\text{Err}(\omega) \stackrel{\text{def}}{=} \begin{cases} \max_{(\theta', \varphi') \in \Omega} \mathcal{L}(\theta, \varphi') - \mathcal{L}(\theta', \varphi) & \text{if } F(\theta, \varphi) = [\nabla_\theta \mathcal{L}(\theta, \varphi) \ -\nabla_\varphi \mathcal{L}(\theta, \varphi)]^\top \\ \max_{\omega' \in \Omega} F(\omega')^\top (\omega - \omega') & \text{otherwise.} \end{cases} \tag{25}$$

**Averaging.** Alg. 1 (AvgSGD) presents the stochastic gradient method with *averaging*, which reduces to the standard (simultaneous) SGD updates for the two-player games used in the GAN literature, but returning an *average* of the iterates.

**Theorem 2.** *Under Assump. 1, 2 and 3, SGD with averaging (Alg. 1) with a constant step-size gives,*

$$\mathbb{E}[\text{Err}(\bar{\omega}_T)] \leq \frac{R^2}{2\eta T} + \eta \frac{M^2 + \sigma^2}{2} \quad \text{where} \quad \bar{\omega}_T \stackrel{\text{def}}{=} \frac{1}{T} \sum_{t=0}^{T-1} \omega_t, \quad \forall T \geq 1. \tag{26}$$

Thm. 2 uses a similar proof as (Nemirovski et al., 2009). The constant term $\eta(M^2 + \sigma^2)/2$ in (26) is called the *variance term*. This type of bound is standard in stochastic optimization. We also provide in §F a similar $\tilde{O}(1/\sqrt{t})$ rate with an extra log factor when $\eta_t = \frac{\eta}{\sqrt{t}}$. We show that this variance term is smaller than the one of *SGD with prediction method* (Yadav et al., 2018) in §E.

**Extrapolations.** Alg. 2 (AvgExtraSGD) adds an extrapolation step compared to Alg. 1 in order to reduce the oscillations due to the game between the two players. A theoretical consequence is that it has a smaller variance term than (26). As discussed previously, Assump. 2 made in Thm. 2 for the convergence of Alg. 1 is very strong in the unbounded setting. One advantage of SGD with *extrapolation* is that Thm. 3 does not require this assumption.

**Theorem 3.** *(Juditsky et al., 2011, Thm. 1) Under Assump. 1 and 3, if $\mathbb{E}_\xi[F]$ is L-Lipschitz, then SGD with extrapolation and averaging (Alg. 2) using a constant step-size $\eta \leq \frac{1}{\sqrt{3}L}$ gives,*

$$\mathbb{E}[\text{Err}(\bar{\omega}_T)] \leq \frac{R^2}{\eta T} + \frac{7}{2}\eta \sigma^2 \quad \text{where} \quad \bar{\omega}_T \stackrel{\text{def}}{=} \frac{1}{T} \sum_{t=0}^{T-1} \omega'_t, \quad \forall T \geq 1. \tag{27}$$

Since in practice $\sigma \ll M$, the variance term in (27) is significantly smaller than the one in (26). To summarize, SGD with *extrapolation* provides better convergence guarantees but requires two gradient computations and samples per iteration. This motivates our new method, Alg. 3 (AvgPastExtraSGD) which uses *extrapolation from the past* and achieves *the best of both worlds* (in theory).

**Theorem 4.** *Under Assump. 1 and 3, if $\mathbb{E}_\xi[F]$ is L-Lipschitz then SGD with extrapolation from the past using a constant step-size $\eta \leq \frac{1}{2\sqrt{3}L}$, gives that the averaged iterates converge as,*

$$\mathbb{E}[\text{Err}(\bar{\omega}_T)] \leq \frac{R^2}{\eta T} + \frac{13}{2}\eta \sigma^2 \quad \text{where} \quad \bar{\omega}_T \stackrel{\text{def}}{=} \frac{1}{T} \sum_{t=0}^{T-1} \omega'_t \quad \forall T \geq 1. \tag{28}$$

The bound is similar to the one provided in Thm. 3 but each iteration of Alg. 3 is computationally half the cost of an iteration of Alg. 2.

---

[7]The convexity of the cost functions in (3) is a necessary condition (not sufficient) for the operator to be monotone. In the context of a zero-sum game, the convexity of the cost functions is a sufficient condition.

## 5 Combining the techniques with established algorithms

In the previous sections, we presented several techniques that converge for stochastic monotone operators. These techniques can be combined in practice with existing algorithms. We propose to combine them to two standard algorithms used for training deep neural networks: the Adam optimizer (Kingma and Ba, 2015) and the SGD optimizer (Robbins and Monro, 1951). For the Adam optimizer, there are several possible choices on how to update the moments. This choice can lead to different algorithms in practice: for example, even in the unconstrained case, our proposed Adam with extrapolation from the past (Alg. 4) is different from Optimistic Adam (Daskalakis et al., 2018) (the moments are updated differently). Note that in the case of a two-player game (3), the previous convergence results can be generalized to gradient updates with a different step-size for each player by simply rescaling the objectives $\mathcal{L}_G$ and $\mathcal{L}_D$ by a different scaling factor. A detailed pseudo-code for Adam with extrapolation step (Extra-Adam) is given in Algorithm 4. Note that our interest regarding this algorithm is practical and that we do not provide any convergence proof.

---

**Algorithm 4** Extra-Adam: proposed Adam with extrapolation step.

---

**input:** step-size $\eta$, decay rates for moment estimates $\beta_1, \beta_2$, access to the stochastic gradients $\nabla\ell_t(\cdot)$ and to the projection $P_\Omega[\cdot]$ onto the constraint set $\Omega$, initial parameter $\boldsymbol{\omega}_0$, averaging scheme $(\rho_t)_{t\geq 1}$

**for** $t = 0 \ldots T - 1$ **do**

    **Option 1: Standard extrapolation.**

        Sample new mini-batch and compute stochastic gradient: $g_t \leftarrow \nabla\ell_t(\boldsymbol{\omega}_t)$

    **Option 2: Extrapolation from the past**

        Load previously saved stochastic gradient: $g_t = \nabla\ell_{t-1/2}(\boldsymbol{\omega}_{t-1/2})$

    Update estimate of first moment for extrapolation: $m_{t-1/2} \leftarrow \beta_1 m_{t-1} + (1-\beta_1)g_t$

    Update estimate of second moment for extrapolation: $v_{t-1/2} \leftarrow \beta_2 v_{t-1} + (1-\beta_2)g_t^2$

    Correct the bias for the moments: $\hat{m}_{t-1/2} \leftarrow m_{t-1/2}/(1-\beta_1^{2t-1}), \hat{v}_{t-1/2} \leftarrow v_{t-1/2}/(1-\beta_2^{2t-1})$

    Perform *extrapolation* step from iterate at time $t$: $\boldsymbol{\omega}_{t-1/2} \leftarrow P_\Omega[\boldsymbol{\omega}_t - \eta\frac{\hat{m}_{t-1/2}}{\sqrt{\hat{v}_{t-1/2}}+\epsilon}]$

    Sample new mini-batch and compute stochastic gradient: $g_{t+1/2} \leftarrow \nabla\ell_{t+1/2}(\boldsymbol{\omega}_{t+1/2})$

    Update estimate of first moment: $m_t \leftarrow \beta_1 m_{t-1/2} + (1-\beta_1)g_{t+1/2}$

    Update estimate of second moment: $v_t \leftarrow \beta_2 v_{t-1/2} + (1-\beta_2)g_{t+1/2}^2$

    Compute bias corrected for first and second moment: $\hat{m}_t \leftarrow m_t/(1-\beta_1^{2t}), \hat{v}_t \leftarrow v_t/(1-\beta_2^{2t})$

    Perform *update* step from the iterate at time $t$: $\boldsymbol{\omega}_{t+1} \leftarrow P_\Omega[\boldsymbol{\omega}_t - \eta\frac{\hat{m}_t}{\sqrt{\hat{v}_t}+\epsilon}]$

**end for**

**Output:** $\boldsymbol{\omega}_{T-1/2}, \boldsymbol{\omega}_T$ or $\bar{\boldsymbol{\omega}}_T = \sum_{t=0}^{T-1} \rho_{t+1}\boldsymbol{\omega}_{t+1/2}/\sum_{t=0}^{T-1}\rho_{t+1}$ (see (8) for online averaging)

---

## 6 Related Work

The extragradient method is a standard algorithm to optimize variational inequalities. This algorithm has been originally introduced by Korpelevich (1976) and extended by Nesterov (2007) and Nemirovski (2004). Stochastic versions of the extragradient have been recently analyzed (Juditsky et al., 2011; Yousefian et al., 2014; Iusem et al., 2017) for stochastic variational inequalities with *bounded constraints*. A linearly convergent variance reduced version of the stochastic gradient method has been proposed by Palaniappan and Bach (2016) for strongly monotone variational inequalities. Extrapolation can also be related to *optimistic methods* (Chiang et al., 2012; Rakhlin and Sridharan, 2013) proposed in the online learning literature (see more details in §3.3). Interesting non-convex results were proved, for a new notion of regret minimization, by Hazan et al. (2017) and in the context of online learning for GANs by Grnarova et al. (2018).

Several methods to stabilize GANs consist in transforming a zero-sum formulation into a more general game that can no longer be cast as a saddle point problem. This is the case of the *non-saturating* formulation of GANs (Goodfellow et al., 2014; Fedus et al., 2018), the DCGANs (Radford et al., 2016), the *gradient penalty*[8] for WGANs (Gulrajani et al., 2017). Yadav et al. (2018) propose an optimization method for GANs based on AltSGD using an additional momentum-based step on the generator. Daskalakis et al. (2018) proposed a method inspired from game theory. Li et al. (2017)

---

[8] The gradient penalty is only added to the discriminator cost function. Since this gradient penalty depends also on the generator, WGAN-GP cannot be cast as a SP problem and is actually a non-zero sum game.

suggest to dualize the GAN objective to reformulate it as a maximization problem and Mescheder et al. (2017) propose to add the norm of the gradient in the objective to get a better signal. Gidel et al. (2019) analyzed a generalization of the bilinear example (9) with a focus put on the effect of momentum on this problem. They do not consider extrapolation (see §B.3 for more details). *Unrolling* steps (Metz et al., 2017) can be confused with extrapolation but is fundamentally different: the perspective is to try to approximate the "true generator objective function" unrolling for $K$ steps the updates of the discriminator and then updating the generator.

Regarding the averaging technique, some recent work appear to have already successfully used *geometric averaging* (7) for GANs in practice, but only briefly mention it (Karras et al., 2018; Mescheder et al., 2018). By contrast, the present work formally motivates and justifies the use of averaging for GANs by relating them to the VIP perspective, and sheds light on its underlying intuitions in §3.1. Subsequent to our first preprint, Yazıcı et al. (2019) explored averaging empirically in more depth, while Mertikopoulos et al. (2019) also investigated extrapolation, providing asymptotic convergence results (i.e. without any rate of convergence) in the context of *coherent saddle point*. The coherence assumption is slightly weaker than monotonicity.

## 7 EXPERIMENTS

Our goal in this experimental section is not to provide new state-of-the art results with architectural improvements or a new GAN formulation, but to show that using the *techniques* (with theoretical guarantees in the monotone case) that we introduced earlier allows us to optimize standard GANs in a better way. These techniques, which are orthogonal to the design of new formulations of GAN optimization objectives, and to architectural choices, can potentially be used for the training of any type of GAN. We will compare the following optimization algorithms: baselines are SGD and Adam using either simultaneous updates on the generator and on the discriminator (denoted **SimAdam** and **SimSGD**) or $k$ updates on the discriminator alternating with 1 update on the generator (denoted **AltSGD**$\{k\}$ and **AltAdam**$\{k\}$).[9] Variants that use *extrapolation* are denoted **ExtraSGD** (Alg. 2) and **ExtraAdam** (Alg. 4). Variants using *extrapolation from the past* are **PastExtraSGD** (Alg. 3) and **PastExtraAdam** (Alg. 4). We also present results using as output the *averaged* iterates, adding **Avg** as a prefix of the algorithm name when we use (uniform) *averaging*.

### 7.1 BILINEAR SADDLE POINT (STOCHASTIC)

We first test the various stochastic algorithms on a simple ($n = 10^3, d = 10^3$) finite sum bilinear objective (a monotone operator) constrained to $[-1, 1]^d$:

$$\frac{1}{n} \sum_{i=1}^{n} \left( \boldsymbol{\theta}^\top \boldsymbol{M}^{(i)} \boldsymbol{\varphi} + \boldsymbol{\theta}^\top \boldsymbol{a}^{(i)} + \boldsymbol{\varphi}^\top \boldsymbol{b}^{(i)} \right) \quad (29)$$

$$\text{solved by } (\boldsymbol{\theta}^*, \boldsymbol{\varphi}^*) \text{ s.t. } \begin{cases} \bar{\boldsymbol{M}} \boldsymbol{\varphi}^* = -\bar{\boldsymbol{a}} \\ \bar{\boldsymbol{M}}^T \boldsymbol{\theta}^* = -\bar{\boldsymbol{b}} \end{cases},$$

where $\bar{\boldsymbol{a}} := \frac{1}{n} \sum_{i=1}^{n} \boldsymbol{a}^{(i)}$, $\bar{\boldsymbol{b}} := \frac{1}{n} \sum_{i=1}^{n} \boldsymbol{b}^{(i)}$ and $\bar{\boldsymbol{M}} := \frac{1}{n} \sum_{i=1}^{n} \boldsymbol{M}^{(i)}$. The matrices $\boldsymbol{M}_{kj}^{(i)}, \boldsymbol{a}_k^{(i)}, \boldsymbol{b}_k^{(i)}$; $1 \le i \le n, 1 \le j, k \le d$ were randomly generated, but ensuring that $(\boldsymbol{\theta}^*, \boldsymbol{\varphi}^*)$ belongs to $[-1, 1]^d$. Results are shown in Fig. 3. We can see that AvgAltSGD1 and AvgPastExtraSGD perform the best on this task.

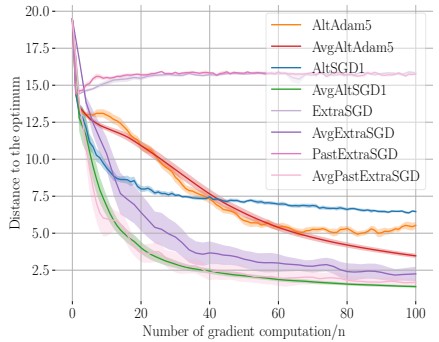

Figure 3: Performance of the considered stochastic optimization algorithms on the bilinear problem (29). Each method uses its respective optimal step-size found by grid-search.

### 7.2 WGAN AND WGAN-GP ON CIFAR10

We evaluate the proposed techniques in the context of GAN training, which is a challenging stochastic optimization problem where the objectives of both players are non-convex. We propose to evaluate the Adam variants of the different optimization algorithms (see Alg. 4 for Adam with *extrapolation*) by training two different architectures on the CIFAR10 dataset (Krizhevsky and Hinton, 2009). First, we consider a constrained zero-sum game by training the DCGAN architecture (Radford et al.,

---

[9]In the original WGAN paper (Arjovsky et al., 2017), the authors use $k = 5$.

| Model | WGAN (DCGAN) | | | WGAN-GP (ResNet) | | |
|---|---|---|---|---|---|---|
| Method | no avg | uniform avg | EMA | no avg | uniform avg | EMA |
| SimAdam | $6.05 \pm .12$ | $5.85 \pm .16$ | $6.08 \pm .10$ | $7.51 \pm .17$ | $7.68 \pm .43$ | $7.60 \pm .17$ |
| AltAdam5 | $5.45 \pm .08$ | $5.72 \pm .06$ | $5.49 \pm .05$ | $7.57 \pm .02$ | $8.01 \pm .05$ | $7.66 \pm .03$ |
| ExtraAdam | $\mathbf{6.38 \pm .09}$ | $\mathbf{6.38 \pm .20}$ | $\mathbf{6.37 \pm .08}$ | $7.90 \pm .11$ | $\mathbf{8.47 \pm .10}$ | $8.13 \pm .07$ |
| PastExtraAdam | $5.98 \pm .15$ | $6.07 \pm .19$ | $6.01 \pm .11$ | $7.84 \pm .06$ | $8.01 \pm .09$ | $7.99 \pm .03$ |
| OptimAdam | $5.74 \pm .10$ | $5.80 \pm .08$ | $5.78 \pm .05$ | $7.98 \pm .08$ | $8.18 \pm .09$ | $8.10 \pm .06$ |

Table 1: Best inception scores (averaged over 5 runs) achieved on CIFAR10 for every considered Adam variant. OptimAdam is the related *Optimistic Adam* (Daskalakis et al., 2018) algorithm. EMA denotes *exponential moving average* (with $\beta = 0.9999$, see Eq. 8). We see that the techniques of extrapolation and averaging consistently enable improvements over the baselines (in italic).

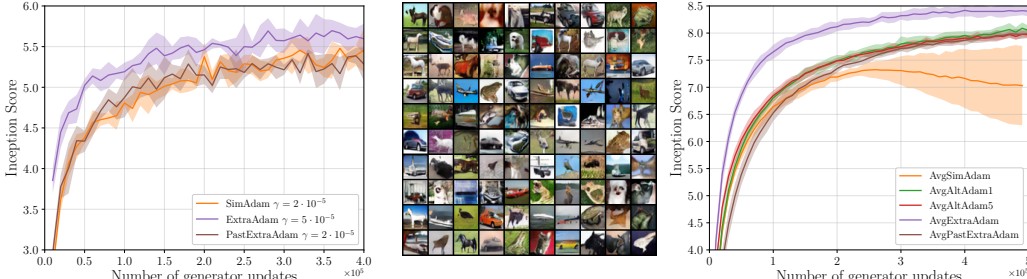

Figure 4: **Left:** Mean and standard deviation of the inception score computed over 5 runs for each method on WGAN trained on CIFAR10. To keep the graph readable we show only SimAdam but AltAdam performs similarly. **Middle:** Samples from a ResNet generator trained with the WGAN-GP objective using AvgExtraAdam. **Right:** WGAN-GP trained on CIFAR10: mean and standard deviation of the inception score computed over 5 runs for each method using the best performing learning rates; all experiments were run on a NVIDIA Quadro GP100 GPU. We see that ExtraAdam converges faster than the Adam baselines.

2016) with the WGAN objective and weight clipping as proposed by Arjovsky et al. (2017). Then, we compare the different methods on a state-of-the-art architecture by training a ResNet with the WGAN-GP objective similar to Gulrajani et al. (2017). Models are evaluated using the inception score (IS) (Salimans et al., 2016) computed on 50,000 samples. We also provide the FID (Heusel et al., 2017) and the details on the ResNet architecture in §G.3.

For each algorithm, we did an extensive search over the hyperparameters of Adam. We fixed $\beta_1 = 0.5$ and $\beta_2 = 0.9$ for all methods as they seemed to perform well. We note that as proposed by Heusel et al. (2017), it is quite important to set different learning rates for the generator and discriminator. Experiments were run with 5 random seeds for 500,000 updates of the generator.

Tab. 1 reports the best IS achieved on these problems by each considered method. We see that the techniques of *extrapolation* and *averaging* consistently enable improvements over the baselines (see §G.5 for more experiments on *averaging*). Fig. 4 shows training curves for each method (for their best performing learning rate), as well as samples from a ResNet generator trained with ExtraAdam on a WGAN-GP objective. For both tasks, using an *extrapolation step* and averaging with Adam (ExtraAdam) outperformed all other methods. Combining ExtraAdam with averaging yields results that improve significantly over the previous state-of-the-art IS (8.2) and FID (21.7) on CIFAR10 as reported by Miyato et al. (2018) (see Tab. 5 for FID). We also observed that methods based on *extrapolation* are less sensitive to learning rate tuning and can be used with higher learning rates with less degradation; see §G.4 for more details.

## 8 CONCLUSION

We newly addressed GAN objectives in the framework of variational inequality. We tapped into the optimization literature to provide more principled techniques to optimize such games. We leveraged these techniques to develop practical optimization algorithms suitable for a wide range of GAN training objectives (including non-zero sum games and projections onto constraints). We experimentally verified that this could yield better trained models, improving the previous state of the art. The presented techniques address a fundamental problem in GAN training in a principled way, and are orthogonal to the design of new GAN architectures and objectives. They are thus likely to be widely applicable, and benefit future development of GANs.

ACKNOWLEDGMENTS.

This research was partially supported by the Canada CIFAR AI Chair Program, the Canada Excellence Research Chair in "Data Science for Realtime Decision-making", by the NSERC Discovery Grant RGPIN-2017-06936 and by a Google Focused Research award. Gauthier Gidel would like to acknowledge Benoît Joly and Florestan Martin-Baillon for bringing a fresh point of view on the proof of Proposition 1.

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

## A    DEFINITIONS

In this section, we recall usual definitions and lemmas from convex analysis. We start with the definitions and lemmas regarding the projection mapping.

### A.1    PROJECTION MAPPING

**Definition 1.** *The projection $P_\Omega$ onto $\Omega$ is defined as,*

$$P_\Omega(\boldsymbol{\omega}') \in \arg\min_{\boldsymbol{\omega}' \in \Omega} \|\boldsymbol{\omega} - \boldsymbol{\omega}'\|_2^2 . \tag{30}$$

When $\Omega$ is a convex set, this projection is unique. This is a consequence of the following lemma that we will use in the following sections: the *non-expansiveness* of the projection onto a convex set.

**Lemma 1.** *Let $\Omega$ a convex set, the projection mapping $P_\Omega : \mathbb{R}^d \to \Omega$ is nonexpansive, i.e.,*

$$\|P_\Omega(\boldsymbol{\omega}) - P_\Omega(\boldsymbol{\omega}')\|_2 \leq \|\boldsymbol{\omega} - \boldsymbol{\omega}'\|_2 , \quad \forall \boldsymbol{\omega}, \boldsymbol{\omega}' \in \Omega . \tag{31}$$

This is standard convex analysis result which can be found for instance in (Boyd and Vandenberghe, 2004). The following lemma is also standard in convex analysis and its proof uses similar arguments as the proof of Lemma 1.

**Lemma 2.** *Let $\boldsymbol{\omega} \in \Omega$ and $\boldsymbol{\omega}^+ \overset{def}{=} P_\Omega(\boldsymbol{\omega} + \boldsymbol{u})$, then for all $\boldsymbol{\omega}' \in \Omega$ we have,*

$$\|\boldsymbol{\omega}^+ - \boldsymbol{\omega}'\|_2^2 \leq \|\boldsymbol{\omega} - \boldsymbol{\omega}'\|_2^2 + 2\boldsymbol{u}^\top(\boldsymbol{\omega}^+ - \boldsymbol{\omega}') - \|\boldsymbol{\omega}^+ - \boldsymbol{\omega}\|_2^2 . \tag{32}$$

*Proof of Lemma 2.* We start by simply developing,

$$\|\boldsymbol{\omega}^+ - \boldsymbol{\omega}'\|_2^2 = \|(\boldsymbol{\omega}^+ - \boldsymbol{\omega}) + (\boldsymbol{\omega} - \boldsymbol{\omega}')\|_2^2 = \|\boldsymbol{\omega} - \boldsymbol{\omega}'\|_2^2 + 2(\boldsymbol{\omega}^+ - \boldsymbol{\omega})^\top(\boldsymbol{\omega} - \boldsymbol{\omega}') + \|\boldsymbol{\omega}^+ - \boldsymbol{\omega}\|_2^2$$
$$= \|\boldsymbol{\omega} - \boldsymbol{\omega}'\|_2^2 + 2(\boldsymbol{\omega}^+ - \boldsymbol{\omega})^\top(\boldsymbol{\omega}^+ - \boldsymbol{\omega}') - \|\boldsymbol{\omega}^+ - \boldsymbol{\omega}\|_2^2 .$$

Then since $\boldsymbol{\omega}^+$ is the projection onto the convex set $\Omega$ of $\boldsymbol{\omega} + \boldsymbol{u}$, we have that $(\boldsymbol{\omega}^+ - (\boldsymbol{\omega} + \boldsymbol{u}))^\top(\boldsymbol{\omega}^+ - \boldsymbol{\omega}') \leq 0 , \; \forall \boldsymbol{\omega}' \in \Omega$, leading to the result of the Lemma.  $\square$

### A.2    SMOOTHNESS AND MONOTONICITY OF THE OPERATOR

Another important property used is the Lipschitzness of an operator.

**Definition 2.** *A mapping $F : \mathbb{R}^p \to \mathbb{R}^d$ is said to be L-Lipschitz if,*

$$\|F(\boldsymbol{\omega}) - F(\boldsymbol{\omega}')\|_2 \leq L\|\boldsymbol{\omega} - \boldsymbol{\omega}'\|_2 , \quad \forall \boldsymbol{\omega}, \boldsymbol{\omega}' \in \Omega . \tag{33}$$

In this paper, we also use the notion of *strong monotonicity*, which is a generalization for operators of the notion of strong convexity. Let us first recall the definition of the latter,

**Definition 3.** *A differentiable function $f : \Omega \to \mathbb{R}$ is said to be $\mu$-strongly convex if*

$$f(\boldsymbol{\omega}) \geq f(\boldsymbol{\omega}') + \nabla f(\boldsymbol{\omega}')^\top(\boldsymbol{\omega} - \boldsymbol{\omega}') + \frac{\mu}{2}\|\boldsymbol{\omega} - \boldsymbol{\omega}'\|_2^2 \quad \forall \boldsymbol{\omega}, \boldsymbol{\omega}' \in \Omega . \tag{34}$$

**Definition 4.** *A function $(\boldsymbol{\theta}, \boldsymbol{\varphi}) \mapsto \mathcal{L}(\boldsymbol{\theta}, \boldsymbol{\varphi})$ is said convex-concave if $\mathcal{L}(\cdot, \boldsymbol{\varphi})$ is convex for all $\boldsymbol{\varphi} \in \Phi$ and $\mathcal{L}(\boldsymbol{\theta}, \cdot)$ is concave for all $\boldsymbol{\theta} \in \Theta$. An $\mathcal{L}$ is said to be $\mu$-strongly convex-concave if $(\boldsymbol{\theta}, \boldsymbol{\varphi}) \mapsto \mathcal{L}(\boldsymbol{\theta}, \boldsymbol{\varphi}) - \frac{\mu}{2}\|\boldsymbol{\theta}\|_2^2 + \frac{\mu}{2}\|\boldsymbol{\varphi}\|_2^2$ is convex-concave.*

If a function $f$ (resp. $\mathcal{L}$) is strongly convex (resp. strongly convex-concave), its gradient $\nabla f$ (resp. $[\nabla_{\boldsymbol{\theta}}\mathcal{L} \; -\nabla_{\boldsymbol{\varphi}}\mathcal{L}]^\top$) is strongly monotone, i.e.,

**Definition 5.** *For $\mu > 0$, an operator $F : \Omega \to \mathbb{R}^d$ is said to be $\mu$-strongly monotone if*

$$(F(\boldsymbol{\omega}) - F(\boldsymbol{\omega}'))^\top(\boldsymbol{\omega} - \boldsymbol{\omega}') \geq \mu\|\boldsymbol{\omega} - \boldsymbol{\omega}'\|_2^2 . \tag{35}$$

# B    GRADIENT METHODS ON UNCONSTRAINED BILINEAR GAMES

In this section, we will prove the results provided in §3, namely Proposition 1, Proposition 2 and Theorem 1. For Proposition 1 and 2, let us recall the context. We wanted to derive properties of some gradient methods on the following simple illustrative example

$$\min_{\theta \in \mathbb{R}} \max_{\phi \in \mathbb{R}} \ \theta \cdot \phi \tag{36}$$

## B.1    PROOF OF PROPOSITION 1

Let us first recall the proposition:

**Proposition' 1.** *The* simultaneous *iterates diverge geometrically and the* alternating *iterates defined in* (11) *are bounded but do not converge to 0 as*

Simultaneous: $\theta_{t+1}^2 + \phi_{t+1}^2 = (1 + \eta^2)(\theta_t^2 + \phi_t^2)$,    Alternating: $\theta_t^2 + \phi_t^2 = \Theta(\theta_0^2 + \phi_0^2)$   (37)

*where* $u_t = \Theta(v_t) \Leftrightarrow \exists \alpha, \beta, t_0 > 0$ *such that* $\forall t \geq t_0, \alpha v_t \leq u_t \leq \beta v_t$.

*The uniform average* $(\bar{\theta}_t, \bar{\phi}_t) \stackrel{def}{=} \frac{1}{t} \sum_{s=0}^{t-1} (\theta_s, \phi_s)$ *of the* simultaneous *updates (resp. the* alternating *updates) diverges (resp. converges to 0) as,*

Simultaneous: $\bar{\theta}_t^2 + \bar{\phi}_t^2 = \Theta\left( \frac{\theta_0^2 + \phi_0^2}{\eta^2 t^2} (1 + \eta^2)^t \right)$,    Alternating: $\bar{\theta}_t^2 + \bar{\phi}_t^2 = \Theta\left( \frac{\theta_0^2 + \phi_0^2}{\eta^2 t^2} \right)$.   (38)

*Proof.* Let us start with the *simultaneous* update rule:

$$\begin{cases} \theta_{t+1} = \theta_t - \eta \phi_t \\ \phi_{t+1} = \phi_t + \eta \theta_t \, . \end{cases} \tag{39}$$

Then we have,

$$\theta_{t+1}^2 + \phi_{t+1}^2 = (\theta_t - \eta \phi_t)^2 + (\phi_t + \eta \theta_t)^2 \tag{40}$$

$$= (1 + \eta^2)(\theta_t^2 + \phi_t^2) \, . \tag{41}$$

The update rule (39) also gives us,

$$\begin{cases} \eta \phi_t = \theta_t - \theta_{t+1} \\ \eta \theta_t = \phi_{t+1} - \phi_t \, . \end{cases} \tag{42}$$

Summing (42) for $0 \leq t \leq T - 1$ to get telescoping sums, we get

$$(\eta^2 T^2)(\bar{\phi}_T^2 + \bar{\theta}_T^2) = (\theta_0 - \theta_T)^2 + (\phi_0 - \phi_T)^2 \tag{43}$$

$$= ((1 + \eta^2)^T + 1)(\theta_0^2 + \phi_0^2) - 2\theta_0 \theta_T - 2\phi_0 \phi_T \tag{44}$$

$$= \Theta\left( (1 + \eta^2)^T ((\theta_0^2 + \phi_0^2)) \right) . \tag{45}$$

Let us continue with the *alternating* update rule

$$\begin{cases} \theta_{t+1} = \theta_t - \eta \phi_t \\ \phi_{t+1} = \phi_t + \eta \theta_{t+1} = \phi_t + \eta(\theta_t - \eta \phi_t) \end{cases} \tag{46}$$

Then we have,

$$\begin{bmatrix} \theta_{t+1} \\ \phi_{t+1} \end{bmatrix} = \begin{bmatrix} 1 & -\eta \\ \eta & 1 - \eta^2 \end{bmatrix} \begin{bmatrix} \theta_t \\ \phi_t \end{bmatrix} . \tag{47}$$

By simple linear algebra, for $\eta < 2$, the matrix $M \stackrel{def}{=} \begin{bmatrix} 1 & -\eta \\ \eta & 1 - \eta^2 \end{bmatrix}$ has two complex conjugate eigenvalues which are

$$\lambda_\pm = 1 - \eta \frac{\eta \pm i \sqrt{4 - \eta^2}}{2} \tag{48}$$

and their squared magnitude is equal to $\det(M) = 1 - \eta^2 + \eta^2 = 1$. We can diagonalize $M$ meaning that there exists $P$ an invertible matrix such that $M = P^{-1} \operatorname{diag}(\lambda_+, \lambda_-)P$. Then, we have

$$\begin{bmatrix} \theta_t \\ \phi_t \end{bmatrix} = M^t \begin{bmatrix} \theta_0 \\ \phi_0 \end{bmatrix} = P^{-1} \operatorname{diag}(\lambda_+^t, \lambda_-^t) P \begin{bmatrix} \theta_0 \\ \phi_0 \end{bmatrix} \tag{49}$$

and consequently,

$$\theta_t^2 + \phi_t^2 = \left\| \begin{bmatrix} \theta_t \\ \phi_t \end{bmatrix} \right\|_{\mathbb{C}}^2 = \left\| P^{-1} \operatorname{diag}(\lambda_+^t, \lambda_-^t) P \begin{bmatrix} \theta_0 \\ \phi_0 \end{bmatrix} \right\|_{\mathbb{C}}^2 \leq \|P^{-1}\| \|P\| (\theta_0^2 + \phi_0^2) \tag{50}$$

where $\| \cdot \|_{\mathbb{C}}$ is the norm in $\mathbb{C}^2$ and $\|P\| := \max_{u \in \mathbb{C}^2} \frac{\|Pu\|_{\mathbb{C}}}{\|u\|_{\mathbb{C}}}$ is the induced matrix norm. The same way we have,

$$\theta_0^2 + \phi_0^2 = \left\| M^{-t} \begin{bmatrix} \theta_t \\ \phi_t \end{bmatrix} \right\|_{\mathbb{C}}^2 = \left\| P^{-1} \operatorname{diag}(\lambda_+^{-t}, \lambda_-^{-t}) P \begin{bmatrix} \theta_t \\ \phi_t \end{bmatrix} \right\|_{\mathbb{C}}^2 \leq \|P^{-1}\| \|P\| (\theta_t^2 + \phi_t^2) \tag{51}$$

Hence, if $\theta_0^2 + \phi_0^2 > 0$, the sequence $(\theta_t, \phi_t)$ is bounded but do not converge to 0. Moreover the update rule gives us,

$$\begin{cases} \eta \phi_t = \theta_t - \theta_{t+1} \\ \eta \theta_t = \phi_t - \phi_{t-1} \end{cases} \Rightarrow \begin{cases} \dfrac{\eta}{T} \sum_{t=0}^{T-1} \phi_t = \dfrac{\theta_0 - \theta_T}{T} \\ \dfrac{\eta}{T} \sum_{t=0}^{T-1} \theta_t = \dfrac{\phi_{T-1} - \phi_0 + \eta \theta_0}{T} \end{cases} \Rightarrow \begin{cases} \bar{\phi}_T = \dfrac{\theta_0 - \theta_T}{\eta T} \\ \bar{\theta}_T = \dfrac{\phi_{T-1} - \phi_0 + \eta \theta_0}{\eta T} \end{cases} \tag{52}$$

Consequently, since $\theta_t^2 + \phi_t^2 = \Theta(\theta_0^2 + \phi_0^2)$,

$$\sqrt{\bar{\theta}_t^2 + \bar{\phi}_t^2} = \Theta \left( \frac{\sqrt{\theta_0^2 + \phi_0^2}}{\eta t} \right) \tag{53}$$

$\square$

## B.2 IMPLICIT AND EXTRAPOLATION METHOD

In this section, we will prove a slightly more precise proposition than Proposition 2,

**Proposition' 2.** *The squared norm of the iterates* $N_t^2 \stackrel{def}{=} \theta_t^2 + \phi_t^2$, *where the update rule of* $\theta_t$ *and* $\phi_t$ *is defined in* (18), *decrease geometrically for any* $0 < \eta < 1$ *as,*[10]

$$\text{Implicit: } N_{t+1}^2 = \frac{N_t^2}{1 + \eta^2}, \quad \text{Extrapolation: } N_{t+1}^2 = (1 - \eta^2 + \eta^4)N_t^2, \quad \forall t \geq 0 \tag{54}$$

*Proof.* Let us recall the update rule for the implicit method

$$\begin{cases} \theta_{t+1} = \theta_t - \eta \phi_{t+1} \\ \phi_{t+1} = \phi_t + \eta \theta_{t+1} \end{cases} \Rightarrow \begin{cases} (1 + \eta^2)\theta_{t+1} = \theta_t - \eta \phi_t \\ (1 + \eta^2)\phi_{t+1} = \phi_t + \eta \theta_t \end{cases} \tag{55}$$

Then,

$$(1 + \eta^2)^2 (\theta_{t+1}^2 + \phi_{t+1}^2) = (\theta_t - \eta \phi_t)^2 + (\phi_t + \eta \theta_t)^2 \tag{56}$$
$$= \theta_t^2 + \phi_t^2 + + \eta^2 (\theta_t^2 + \phi_t^2), \tag{57}$$

implying that

$$\theta_{t+1}^2 + \phi_{t+1}^2 = \frac{\theta_t^2 + \phi_t^2}{1 + \eta^2}, \tag{58}$$

---

[10]Note that the relationship (54) holds actually for *any* $\eta$ for the implicit method, and thus the decrease is geometric for any non-zero step size.

which is valid for *any* $\eta$.

For the extrapolation method, we have the update rule

$$\begin{cases} \theta_{t+1} = \theta_t - \eta(\phi_t + \eta\theta_t) \\ \phi_{t+1} = \phi_t + \eta(\theta_t - \eta\phi_t) \end{cases} \tag{59}$$

Implying that,

$$\theta_{t+1}^2 + \phi_{t+1}^2 = (\theta_t - \eta(\phi_t + \eta\theta_t))^2 + (\phi_t + \eta(\theta_t - \eta\phi_t))^2 \tag{60}$$
$$= \theta_t^2 + \phi_t^2 - 2\eta^2(\theta^2 + \phi^2) + \eta^2((\theta_t - \eta\phi_t)^2 + (\phi_t + \eta\theta_t)^2) \tag{61}$$
$$= (1 - \eta^2 + \eta^4)(\theta_t^2 + \phi_t^2) \tag{62}$$

$\square$

### B.3 GENERALIZATION TO GENERAL UNCONSTRAINED BILINEAR OBJECTIVE

In this section, we will show how to simply extend the study of the algorithm of interest provided in §3 on the general unconstrained bilinear example,

$$\min_{\boldsymbol{\theta} \in \mathbb{R}^d} \max_{\boldsymbol{\varphi} \in \mathbb{R}^p} \quad \boldsymbol{\theta}^\top \boldsymbol{A} \boldsymbol{\varphi} - \boldsymbol{b}^\top \boldsymbol{\theta} - \boldsymbol{c}^\top \boldsymbol{\varphi} \tag{63}$$

where, $\boldsymbol{A} \in \mathbb{R}^{d \times p}$, $\boldsymbol{b} \in \mathbb{R}^d$ and $\boldsymbol{c} \in \mathbb{R}^p$. The only assumption we will make is that this problem is feasible which is equivalent to say that there exists a solution $(\boldsymbol{\theta}^*, \boldsymbol{\varphi}^*)$ to the system

$$\begin{cases} \boldsymbol{A}\boldsymbol{\varphi}^* = \boldsymbol{b} \\ \boldsymbol{A}^\top \boldsymbol{\theta}^* = \boldsymbol{c} \, . \end{cases} \tag{64}$$

In this case, we can re-write (63) as

$$\min_{\boldsymbol{\theta} \in \mathbb{R}^d} \max_{\boldsymbol{\varphi} \in \mathbb{R}^p} \quad (\boldsymbol{\theta} - \boldsymbol{\theta}^*)^\top \boldsymbol{A}(\boldsymbol{\varphi} - \boldsymbol{\varphi}^*) + c \tag{65}$$

where $c := -\boldsymbol{\theta}^{*\top} \boldsymbol{A} \boldsymbol{\varphi}^*$ is a constant that does not depend on $\boldsymbol{\theta}$ and $\boldsymbol{\varphi}$.

First, let us show that we can reduce the study of simultaneous, alternating, extrapolation and implicit updates rules for (63) to the study of the respective unidimensional updates (11) and (18).

This reduction has already been proposed by Gidel et al. (2019). For completeness, we reproduce here similar arguments. The following lemma is a bit more general than the result provided by Gidel et al. (2019). It states that the study of a wide class of *unconstrained* first order method on (63) can be reduced to the study of the method on (36), with potentially rescaled step-sizes.

Before explicitly stating the lemma, we need to introduce a bit of notation to encompass easily our several methods in a unified way. First, we let $\boldsymbol{\omega}_t := (\boldsymbol{\theta}_t, \boldsymbol{\varphi}_t)$, where the index $t$ here is a more general index which can vary more often than the one in §3. For example, for the extrapolation method, we could consider $\boldsymbol{\omega}_1 = \boldsymbol{\omega}'_{0+1/2}$ and $\boldsymbol{\omega}_2 = \boldsymbol{\omega}'_1$, where $\boldsymbol{\omega}'$ was the sequence defined for the extragradient. For the alternated updates, we can consider $\boldsymbol{\omega}_1 = (\boldsymbol{\theta}'_1, \boldsymbol{\varphi}'_0)$ and $\boldsymbol{\omega}_2 = (\boldsymbol{\theta}'_1, \boldsymbol{\varphi}'_1)$ (this also defines $\boldsymbol{\theta}_2 = \boldsymbol{\theta}'_1$), where $\boldsymbol{\theta}'$ and $\boldsymbol{\varphi}'$ were the sequences originally defined for alternated updates. We are thus ready to state the lemma.

**Lemma 3.** *Let us consider the following very general class of first order methods on (63), i.e.,*

$$\begin{aligned} \boldsymbol{\theta}_t &\in \boldsymbol{\theta}_0 + span(F_{\boldsymbol{\theta}}(\boldsymbol{\omega}_0), \dots, F_{\boldsymbol{\theta}}(\boldsymbol{\omega}_t)), \quad \forall t \in \mathbb{N}, \\ \boldsymbol{\varphi}_t &\in \boldsymbol{\varphi}_0 + span(F_{\boldsymbol{\varphi}}(\boldsymbol{\omega}_0), \dots, F_{\boldsymbol{\varphi}}(\boldsymbol{\omega}_t)), \quad \forall t \in \mathbb{N}, \end{aligned} \tag{66}$$

*where $\boldsymbol{\omega}_t := (\boldsymbol{\theta}_t, \boldsymbol{\varphi}_t)$ and $F_{\boldsymbol{\theta}}(\boldsymbol{\omega}_t) := \boldsymbol{A}\boldsymbol{\varphi}_t - \boldsymbol{b}$, $F_{\boldsymbol{\varphi}}(\boldsymbol{\omega}_t) = \boldsymbol{A}^\top \boldsymbol{\theta}_t - \boldsymbol{c}$. Then, we have*

$$\boldsymbol{\theta}_t = \boldsymbol{U}^\top (\tilde{\boldsymbol{\theta}}_t - \boldsymbol{\theta}^*) \quad and \quad \boldsymbol{\varphi}_t = \boldsymbol{V}^\top (\tilde{\boldsymbol{\varphi}}_t - \boldsymbol{\varphi}^*), \tag{67}$$

*where $\boldsymbol{A} = \boldsymbol{U}\boldsymbol{D}\boldsymbol{V}^\top$ (SVD decomposition) and the couples $([\tilde{\boldsymbol{\theta}}_t]_i, [\tilde{\boldsymbol{\varphi}}]_i)_{1 \le i \le r}$ follow the update rule of the same method on a unidimensional problem (36). In particular, for our methods of interest, the couples $([\tilde{\boldsymbol{\theta}}_t]_i, [\tilde{\boldsymbol{\varphi}}]_i)_{1 \le i \le r}$ follow the same update rule with a respective step-size $\sigma_i \eta$, where $\sigma_i$ are the singular values on the diagonal of $\boldsymbol{D}$.*

*Proof.* Our general class of first order methods can be written with the following update rules:

$$\boldsymbol{\theta}_{t+1} = \boldsymbol{\theta}_0 + \sum_{s=0}^{t+1} \lambda_{st} \boldsymbol{A}(\boldsymbol{\varphi}_s - \boldsymbol{\varphi}^*)$$

$$\boldsymbol{\varphi}_{t+1} = \boldsymbol{\varphi}_0 + \sum_{s=0}^{t+1} \mu_{st} \boldsymbol{A}^\top (\boldsymbol{\theta}_s - \boldsymbol{\theta}^*) \,,$$

where $\lambda_{it}, \mu_{it} \in \mathbb{R}$, $0 \le i \le t+1$. We allow the dependence on $t$ for the algorithm coefficients $\lambda$ and $\mu$ (for example, the alternating rule would zero out some of the coefficients depending on whether we are updating $\boldsymbol{\theta}$ or $\boldsymbol{\varphi}$ at the current iteration). Notice also that if both $\lambda_{(t+1)t}$ and $\mu_{(t+1)t}$ are non-zero, we have an *implicit* scheme.

Thus, using the SVD of $\boldsymbol{A} = \boldsymbol{U}\boldsymbol{D}\boldsymbol{V}^\top$, we get

$$\boldsymbol{U}^\top (\boldsymbol{\theta}_{t+1} - \boldsymbol{\theta}^*) = \boldsymbol{U}^\top (\boldsymbol{\theta}_0 - \boldsymbol{\theta}^*) + \sum_{s=0}^{t+1} \lambda_{st} \boldsymbol{D}\boldsymbol{V}^\top (\boldsymbol{\varphi}_s - \boldsymbol{\varphi}^*)$$

$$\boldsymbol{V}^\top (\boldsymbol{\varphi}_{t+1} - \boldsymbol{\varphi}^*) = \boldsymbol{V}^\top (\boldsymbol{\varphi}_0 - \boldsymbol{\varphi}^*) + \sum_{s=0}^{t+1} \mu_{st} \boldsymbol{D}^\top \boldsymbol{U}^\top (\boldsymbol{\theta}_s - \boldsymbol{\theta}^*) \,,$$

which is equivalent to

$$\begin{cases} \tilde{\boldsymbol{\theta}}_{t+1} = \tilde{\boldsymbol{\theta}}_0 + \sum_{s=0}^{t+1} \lambda_{st} \boldsymbol{D}\tilde{\boldsymbol{\varphi}}_s \\[2mm] \tilde{\boldsymbol{\varphi}}_{t+1} = \tilde{\boldsymbol{\varphi}}_0 + \sum_{s=0}^{t+1} \mu_{st} \boldsymbol{D}^\top \tilde{\boldsymbol{\theta}}_s \,, \end{cases} \tag{68}$$

where $\boldsymbol{D}$ is a rectangular matrix with zeros except on a diagonal block of size $r$. Thus, each coordinate of $\tilde{\boldsymbol{\theta}}_{t+1}$ and $\tilde{\boldsymbol{\varphi}}_{t+1}$ are updated independently, reducing the initial problem to $r$ unidimensional problems,

$$\begin{cases} [\tilde{\boldsymbol{\theta}}_{t+1}]_i = [\tilde{\boldsymbol{\theta}}_0]_i + \sum_{s=0}^{t+1} \lambda_{st} \sigma_i [\tilde{\boldsymbol{\varphi}}_s]_i \\[2mm] [\tilde{\boldsymbol{\varphi}}_{t+1}]_i = [\tilde{\boldsymbol{\varphi}}_0]_i + \sum_{s=0}^{t+1} \mu_{st} \sigma_i [\tilde{\boldsymbol{\theta}}_s]_i \end{cases} \quad 1 \le i \le r \,, \tag{69}$$

where $\sigma_1 \ge \ldots \ge \sigma_r > 0$ are the positive diagonal coefficients of $\boldsymbol{D}$.

Finally, for the coordinate $i$ where the diagonal coefficient of $\boldsymbol{D}$ is equal to 0, we can notice that the sequence $([\tilde{\boldsymbol{\theta}}_t]_i, [\tilde{\boldsymbol{\varphi}}_t]_i)$ is constant. Moreover, we have the freedom to chose any $[\boldsymbol{\theta}^*]_i \in \mathbb{R}$ and $[\boldsymbol{\varphi}^*]_i \in \mathbb{R}$ as a coordinate of the solution of (63). We thus set them respectively equal to $[\boldsymbol{\theta}_0]_i$ and $[\boldsymbol{\varphi}_0]_i$. The update rule (69) corresponds to the update rule of the general first order method considered on this proof on the unidimensional problem (36).

Note that the only additional restriction is that the coefficients $(\lambda_{st})$ and $(\sigma_{st})$ (that are the same for $1 \le i \le r$) are rescaled by the singular values of $\boldsymbol{A}$. In practice, for our methods of interest with a step-size $\eta$, it corresponds to the study of $r$ unidimensional problem with a respective step-size $\sigma_i \eta$, $1 \le i \le r$. □

From this lemma, an extension of Proposition 1 and 2 directly follows to the general unconstrained bilinear objective (63). We note

$$N_t^2 := \text{dist}(\boldsymbol{\theta}_t, \Theta^*)^2 + \text{dist}(\boldsymbol{\varphi}_t, \Phi^*)^2 \,, \tag{70}$$

where $(\Theta^*, \Phi^*)$ is the set of solutions of (63). The following corollary is divided in two points, the first point is a result from Gidel et al. (2019) (note that the result on the average is a straightforward extension of the one provided in Proposition 1 and was not provided by Gidel et al. (2019)), the second result is new. Very similar asymptotic upper bounds regarding extrapolation and implicit methods can be derived by Tseng (1995) computing the exact values of the constant $\tau_1$ and $\tau_2$ (and

noticing that $\tau_3 = \infty$) introduced in (Tseng, 1995, Eq. 3 & 4) for the unconstrained bilinear case. However, since Tseng (1995) works in a very general setting, the bound are not as tight as ours and his proof technique is a bit more technical. Our reduction above provides here a simple proof for our simple setting.

**Corollary 1.**
- *Gidel et al. (2019): The* simultaneous *iterates diverge geometrically and the* alternating *iterates are bounded but do not converge to 0 as,*

$$\text{Simultaneous: } N_{t+1}^2 = (1 + (\sigma_{\min}(\boldsymbol{A})\eta)^2)N_t^2\,, \quad \text{Alternating: } N_t^2 = \Theta(N_0^2)\,, \quad (71)$$

*where* $u_t = \Theta(v_t) \Leftrightarrow \exists \alpha, \beta, t_0 > 0$ *such that* $\forall t \geq t_0$, $\alpha v_t \leq u_t \leq \beta v_t$. *The uniform average* $(\bar{\theta}_t, \bar{\phi}_t) \stackrel{def}{=} \frac{1}{t}\sum_{s=0}^{t-1}(\theta_s, \phi_s)$ *of the* simultaneous *updates (resp. the* alternating updates*) diverges (resp. converges to 0) as,*

$$\text{Simultaneous: } \bar{N}_t^2 \leq \Theta\left(\frac{N_0^2}{t^2}(1 + (\sigma_{\min}(\boldsymbol{A})\eta)^2)^t\right)\,, \quad \text{Alternating: } \bar{N}_t^2 = \Theta\left(\frac{N_0^2}{t^2}\right)\,.$$

- Extrapolation *and* Implicit method*: The iterates respectively generated by the update rules* (14) *and* (17) *on a bilinear unconstrained problem* (63) *do converge linearly for any* $0 < \eta < \frac{1}{\sigma_{\max}(\boldsymbol{A})}$ *at a rate,*[11]

$$\text{Implicit: } N_{t+1}^2 \leq \frac{N_t^2}{1 + (\sigma_{\min}(\boldsymbol{A})\eta)^2}\,, \quad \forall t \geq 0 \quad (72)$$

$$\text{Extrapolation: } N_{t+1}^2 \leq (1 - (\sigma_{\min}(\boldsymbol{A})\eta)^2 + (\sigma_{\min}(\boldsymbol{A})\eta)^4)N_t^2\,, \quad \forall t \geq 0\,. \quad (73)$$

*Particularly, for* $\eta = \frac{1}{2\sigma_{\max}(\boldsymbol{A})}$ *we get for the extrapolation method,*

$$\text{Extrapolation: } N_{t+1}^2 \leq (1 - \frac{1}{8\kappa})^t N_0^2\,, \quad \forall t \geq 0\,. \quad (74)$$

*where* $\kappa := \frac{\sigma_{\max}(\boldsymbol{A})^2}{\sigma_{\min}(\boldsymbol{A})^2}$ *is the condition number of* $\boldsymbol{A}^\top \boldsymbol{A}$.

## B.4 EXTRAPOLATION FROM THE PAST FOR STRONGLY CONVEX OBJECTIVES

Let us recall what we call *projected extrapolation form the past*, where we used the notation $\boldsymbol{\omega}_t' = \boldsymbol{\omega}_{t+1/2}$ for compactness,

$$\text{Extrapolation from the past: } \boldsymbol{\omega}_t' = P_\Omega[\boldsymbol{\omega}_t - \eta F(\boldsymbol{\omega}_{t-1}')] \quad (75)$$

$$\text{Perform update step: } \boldsymbol{\omega}_{t+1} = P_\Omega[\boldsymbol{\omega}_t - \eta F(\boldsymbol{\omega}_t')] \text{ and store: } F(\boldsymbol{\omega}_t') \quad (76)$$

where $P_\Omega[\cdot]$ is the projection onto the constraint set $\Omega$. An operator $F : \Omega \to \mathbb{R}^d$ is said to be $\mu$-strongly monotone if

$$(F(\boldsymbol{\omega}) - F(\boldsymbol{\omega}'))^\top (\boldsymbol{\omega} - \boldsymbol{\omega}') \geq \mu \|\boldsymbol{\omega} - \boldsymbol{\omega}'\|_2^2\,. \quad (77)$$

If $F$ is strongly monotone, we can prove the following theorem:

**Theorem' 1.** *If $F$ is $\mu$-strongly monotone (see §A for the definition of strong monotonicity) and $L$-Lipschitz, then the updates* (20) *and* (21) *with* $\eta = \frac{1}{4L}$ *provide linearly converging iterates,*

$$\|\boldsymbol{\omega}_t - \boldsymbol{\omega}^*\|_2^2 \leq \left(1 - \frac{\mu}{4L}\right)^t \|\boldsymbol{\omega}_0 - \boldsymbol{\omega}^*\|_2^2\,, \quad \forall t \geq 0\,. \quad (78)$$

*Proof.* In order to prove this theorem, we will prove a slightly more general result,

$$\|\boldsymbol{\omega}_{t+1} - \boldsymbol{\omega}^*\|_2^2 + \|\boldsymbol{\omega}_{t-1}' - \boldsymbol{\omega}_t'\|_2^2 \leq \left(1 - \frac{\mu}{4L}\right)(\|\boldsymbol{\omega}_t - \boldsymbol{\omega}^*\|_2^2 + \|\boldsymbol{\omega}_{t-1}' - \boldsymbol{\omega}_{t-2}'\|_2^2)\,. \quad (79)$$

with the convention that $\boldsymbol{\omega}_0' = \boldsymbol{\omega}_{-1}' = \boldsymbol{\omega}_{-2}'$. It implies that

$$\|\boldsymbol{\omega}_{t+1} - \boldsymbol{\omega}^*\|_2^2 \leq \|\boldsymbol{\omega}_{t+1} - \boldsymbol{\omega}^*\|_2^2 + \|\boldsymbol{\omega}_{t-1}' - \boldsymbol{\omega}_t'\|_2^2 \leq \left(1 - \frac{\mu}{4L}\right)^t \|\boldsymbol{\omega}_0 - \boldsymbol{\omega}^*\|_2^2\,. \quad (80)$$

Let us first proof three technical lemmas.

---

[11]As before, the inequality (73) for the implicit scheme is actually valid for any step-size.

**Lemma 4.** *If $F$ is $\mu$-strongly monotone, we have*

$$\mu\left(\|\boldsymbol{\omega}_t - \boldsymbol{\omega}^*\|_2^2 - 2\|\boldsymbol{\omega}'_t - \boldsymbol{\omega}_t\|_2^2\right) \le 2F(\boldsymbol{\omega}'_t)^\top(\boldsymbol{\omega}'_t - \boldsymbol{\omega}^*), \quad \forall \boldsymbol{\omega}^* \in \Omega^*. \tag{81}$$

*Proof.* By strong monotonicity and optimality of $\boldsymbol{\omega}^*$,

$$2\mu\|\boldsymbol{\omega}'_t - \boldsymbol{\omega}^*\|_2^2 \le 2F(\boldsymbol{\omega}^*)^\top(\boldsymbol{\omega}'_t - \boldsymbol{\omega}^*) + 2\mu\|\boldsymbol{\omega}'_t - \boldsymbol{\omega}^*\|_2^2 \le 2F(\boldsymbol{\omega}'_t)^\top(\boldsymbol{\omega}'_t - \boldsymbol{\omega}^*) \tag{82}$$

and then we use the inequality $2\|\boldsymbol{\omega}'_t - \boldsymbol{\omega}^*\|_2^2 \ge \|\boldsymbol{\omega}_t - \boldsymbol{\omega}^*\|_2^2 - 2\|\boldsymbol{\omega}'_t - \boldsymbol{\omega}_t\|_2^2$ to get the result claimed. □

**Lemma 5.** *If $F$ is $L$-Lipschitz, we have for any $\boldsymbol{\omega} \in \Omega$,*

$$2\eta_t F(\boldsymbol{\omega}'_t)^\top(\boldsymbol{\omega}'_t - \boldsymbol{\omega}) \le \|\boldsymbol{\omega}_t - \boldsymbol{\omega}\|_2^2 - \|\boldsymbol{\omega}_{t+1} - \boldsymbol{\omega}\|_2^2 - \|\boldsymbol{\omega}'_t - \boldsymbol{\omega}_t\|_2^2 + \eta_t^2 L^2\|\boldsymbol{\omega}'_{t-1} - \boldsymbol{\omega}'_t\|_2^2. \tag{83}$$

*Proof.* Applying Lemma 2 for $(\boldsymbol{\omega}, \boldsymbol{u}, \boldsymbol{\omega}^+, \boldsymbol{\omega}') = (\boldsymbol{\omega}_t, -\eta_t F(\boldsymbol{\omega}'_t), \boldsymbol{\omega}_{t+1}, \boldsymbol{\omega})$ and $(\boldsymbol{\omega}, \boldsymbol{u}, \boldsymbol{\omega}^+, \boldsymbol{\omega}') = (\boldsymbol{\omega}_t, -\eta_t F(\boldsymbol{\omega}'_{t-1}), \boldsymbol{\omega}'_t, \boldsymbol{\omega}_{t+1})$, we get,

$$\|\boldsymbol{\omega}_{t+1} - \boldsymbol{\omega}\|_2^2 \le \|\boldsymbol{\omega}_t - \boldsymbol{\omega}\|_2^2 - 2\eta_t F(\boldsymbol{\omega}'_t)^\top(\boldsymbol{\omega}_{t+1} - \boldsymbol{\omega}) - \|\boldsymbol{\omega}_{t+1} - \boldsymbol{\omega}_t\|_2^2 \tag{84}$$

and

$$\|\boldsymbol{\omega}'_t - \boldsymbol{\omega}_{t+1}\|_2^2 \le \|\boldsymbol{\omega}_t - \boldsymbol{\omega}_{t+1}\|_2^2 - 2\eta_t F(\boldsymbol{\omega}'_{t-1})^\top(\boldsymbol{\omega}'_t - \boldsymbol{\omega}_{t+1}) - \|\boldsymbol{\omega}'_t - \boldsymbol{\omega}_t\|_2^2. \tag{85}$$

Summing (84) and (85) we get,

$$\|\boldsymbol{\omega}_{t+1} - \boldsymbol{\omega}\|_2^2 \le \|\boldsymbol{\omega}_t - \boldsymbol{\omega}\|_2^2 - 2\eta_t F(\boldsymbol{\omega}'_t)^\top(\boldsymbol{\omega}_{t+1} - \boldsymbol{\omega}) \tag{86}$$

$$- 2\eta_t F(\boldsymbol{\omega}'_{t-1})^\top(\boldsymbol{\omega}'_t - \boldsymbol{\omega}_{t+1}) - \|\boldsymbol{\omega}'_t - \boldsymbol{\omega}_t\|_2^2 - \|\boldsymbol{\omega}'_t - \boldsymbol{\omega}_{t+1}\|_2^2 \tag{87}$$

$$= \|\boldsymbol{\omega}_t - \boldsymbol{\omega}\|_2^2 - 2\eta_t F(\boldsymbol{\omega}'_t)^\top(\boldsymbol{\omega}'_t - \boldsymbol{\omega}) - \|\boldsymbol{\omega}'_t - \boldsymbol{\omega}_t\|_2^2 - \|\boldsymbol{\omega}'_t - \boldsymbol{\omega}_{t+1}\|_2^2$$

$$- 2\eta_t(F(\boldsymbol{\omega}'_{t-1}) - F(\boldsymbol{\omega}'_t))^\top(\boldsymbol{\omega}'_t - \boldsymbol{\omega}_{t+1}). \tag{88}$$

Then, we can use the Young's inequality $2a^\top b \le \|a\|_2^2 + \|b\|_2^2$ to get,

$$\|\boldsymbol{\omega}_{t+1} - \boldsymbol{\omega}\|_2^2 \le \|\boldsymbol{\omega}_t - \boldsymbol{\omega}\|_2^2 - 2\eta_t F(\boldsymbol{\omega}'_t)^\top(\boldsymbol{\omega}'_t - \boldsymbol{\omega}) + \eta_t^2\|F(\boldsymbol{\omega}'_{t-1}) - F(\boldsymbol{\omega}'_t)\|_2^2$$

$$+ \|\boldsymbol{\omega}'_t - \boldsymbol{\omega}_{t+1}\|_2^2 - \|\boldsymbol{\omega}'_t - \boldsymbol{\omega}_t\|_2^2 - \|\boldsymbol{\omega}'_t - \boldsymbol{\omega}_{t+1}\|_2^2 \tag{89}$$

$$= \|\boldsymbol{\omega}_t - \boldsymbol{\omega}\|_2^2 - 2\eta_t F(\boldsymbol{\omega}'_t)^\top(\boldsymbol{\omega}'_t - \boldsymbol{\omega}) + \eta_t^2\|F(\boldsymbol{\omega}'_{t-1}) - F(\boldsymbol{\omega}'_t)\|_2^2 - \|\boldsymbol{\omega}'_t - \boldsymbol{\omega}_t\|_2^2$$

$$\le \|\boldsymbol{\omega}_t - \boldsymbol{\omega}\|_2^2 - 2\eta_t F(\boldsymbol{\omega}'_t)^\top(\boldsymbol{\omega}'_t - \boldsymbol{\omega}) + \eta_t^2 L^2\|\boldsymbol{\omega}'_{t-1} - \boldsymbol{\omega}'_t\|_2^2 - \|\boldsymbol{\omega}'_t - \boldsymbol{\omega}_t\|_2^2. \tag{90}$$

□

**Lemma 6.** *For all $t \ge 0$, if we set $\boldsymbol{\omega}'_{-2} = \boldsymbol{\omega}'_{-1} = \boldsymbol{\omega}'_0$ we have*

$$\|\boldsymbol{\omega}'_{t-1} - \boldsymbol{\omega}'_t\|_2^2 \le 4\|\boldsymbol{\omega}_t - \boldsymbol{\omega}'_t\|_2^2 + 4\eta_{t-1}^2 L^2\|\boldsymbol{\omega}'_{t-1} - \boldsymbol{\omega}'_{t-2}\|_2^2 - \|\boldsymbol{\omega}'_{t-1} - \boldsymbol{\omega}'_t\|_2^2. \tag{91}$$

*Proof.* We start with $\|a + b\|_2^2 \le 2\|a\|^2 + 2\|b\|^2$.

$$\|\boldsymbol{\omega}'_{t-1} - \boldsymbol{\omega}'_t\|_2^2 \le 2\|\boldsymbol{\omega}_t - \boldsymbol{\omega}'_t\|_2^2 + 2\|\boldsymbol{\omega}_t - \boldsymbol{\omega}'_{t-1}\|_2^2. \tag{92}$$

Moreover, since the projection is contractive we have that

$$\|\boldsymbol{\omega}_t - \boldsymbol{\omega}'_{t-1}\|_2^2 \le \|\boldsymbol{\omega}_{t-1} - \eta_{t-1}F(\boldsymbol{\omega}'_{t-1}) - \boldsymbol{\omega}_{t-1} - \eta_{t-1}F(\boldsymbol{\omega}'_{t-2})\|_2^2 \tag{93}$$

$$= \eta_{t-1}^2\|F(\boldsymbol{\omega}'_{t-1}) - F(\boldsymbol{\omega}'_{t-2})\|_2^2 \tag{94}$$

$$\le \eta_{t-1}^2 L^2\|\boldsymbol{\omega}'_{t-1} - \boldsymbol{\omega}'_{t-2}\|_2^2. \tag{95}$$

Combining (92) and (95) we get,

$$\|\boldsymbol{\omega}'_{t-1} - \boldsymbol{\omega}'_t\|_2^2 = 2\|\boldsymbol{\omega}'_{t-1} - \boldsymbol{\omega}'_t\|_2^2 - \|\boldsymbol{\omega}'_{t-1} - \boldsymbol{\omega}'_t\|_2^2 \tag{96}$$

$$\le 4\|\boldsymbol{\omega}_t - \boldsymbol{\omega}'_t\|_2^2 + 4\|\boldsymbol{\omega}_t - \boldsymbol{\omega}'_{t-1}\|_2^2 - \|\boldsymbol{\omega}'_{t-1} - \boldsymbol{\omega}'_t\|_2^2 \tag{97}$$

$$\le 4\|\boldsymbol{\omega}_t - \boldsymbol{\omega}'_t\|_2^2 + 4\eta_{t-1}^2 L^2\|\boldsymbol{\omega}'_{t-1} - \boldsymbol{\omega}'_{t-2}\|_2^2 - \|\boldsymbol{\omega}'_{t-1} - \boldsymbol{\omega}'_t\|_2^2. \tag{98}$$

□

**Proof of Theorem 1.** Let $\boldsymbol{\omega}^* \in \Omega^*$ be an optimal point of (VIP). Combining Lemma 4 and Lemma 5 we get,

$$\eta_t \mu \left( \|\boldsymbol{\omega}_t - \boldsymbol{\omega}^*\|_2^2 - 2\|\boldsymbol{\omega}_t' - \boldsymbol{\omega}_t\|_2^2 \right) \leq \|\boldsymbol{\omega}_t - \boldsymbol{\omega}^*\|_2^2 - \|\boldsymbol{\omega}_{t+1} - \boldsymbol{\omega}^*\|_2^2 + \eta_t^2 L^2 \|\boldsymbol{\omega}_{t-1}' - \boldsymbol{\omega}_t'\|_2^2 - \|\boldsymbol{\omega}_t' - \boldsymbol{\omega}_t\|_2^2$$

leading to,

$$\|\boldsymbol{\omega}_{t+1} - \boldsymbol{\omega}^*\|_2^2 \leq (1 - \eta_t \mu) \|\boldsymbol{\omega}_t - \boldsymbol{\omega}^*\|_2^2 + \eta_t^2 L^2 \|\boldsymbol{\omega}_{t-1}' - \boldsymbol{\omega}_t'\|_2^2 - (1 - 2\eta_t \mu)\|\boldsymbol{\omega}_t' - \boldsymbol{\omega}_t\|_2^2 \quad (99)$$

Now using Lemma 6 we get,

$$\|\boldsymbol{\omega}_{t+1} - \boldsymbol{\omega}^*\|_2^2 \leq (1 - \eta_t \mu) \|\boldsymbol{\omega}_t - \boldsymbol{\omega}^*\|_2^2 + \eta_t^2 L^2 (4\eta_{t-1}^2 L^2 \|\boldsymbol{\omega}_{t-1}' - \boldsymbol{\omega}_{t-2}'\|_2^2 - \|\boldsymbol{\omega}_{t-1}' - \boldsymbol{\omega}_t'\|_2^2)$$
$$- (1 - 2\eta_t \mu - 4\eta_t^2 L^2)\|\boldsymbol{\omega}_t' - \boldsymbol{\omega}_t\|_2^2 \quad (100)$$

Now with $\eta_t = \frac{1}{4L} \leq \frac{1}{4\mu}$ we get,

$$\|\boldsymbol{\omega}_{t+1} - \boldsymbol{\omega}^*\|_2^2 \leq \left(1 - \frac{\mu}{4L}\right) \|\boldsymbol{\omega}_t - \boldsymbol{\omega}^*\|_2^2 + \frac{1}{16} \left( \frac{1}{4}\|\boldsymbol{\omega}_{t-1}' - \boldsymbol{\omega}_{t-2}'\|_2^2 - \|\boldsymbol{\omega}_{t-1}' - \boldsymbol{\omega}_t'\|_2^2 \right)$$

Hence, using the fact that $\frac{\mu}{4L} \leq \frac{1}{4}$ we get,

$$\|\boldsymbol{\omega}_{t+1} - \boldsymbol{\omega}^*\|_2^2 + \frac{1}{16}\|\boldsymbol{\omega}_{t-1}' - \boldsymbol{\omega}_t'\|_2^2 \leq \left(1 - \frac{\mu}{4L}\right) \left( \|\boldsymbol{\omega}_t - \boldsymbol{\omega}^*\|_2^2 + \frac{1}{16}\|\boldsymbol{\omega}_{t-1}' - \boldsymbol{\omega}_{t-2}'\|_2^2 \right). \quad (101)$$

$\square$

## C  MORE ON MERIT FUNCTIONS

In this section, we will present how to handle an *unbounded* constraint set $\Omega$ with a more refined *merit function* than (25) used in the main paper. Let $F$ be the continuous operator and $\Omega$ be the constraint set associated with the VIP,

$$\text{find } \boldsymbol{\omega}^* \in \Omega \quad \text{such that} \quad F(\boldsymbol{\omega}^*)^\top (\boldsymbol{\omega} - \boldsymbol{\omega}^*) \geq 0, \quad \forall \boldsymbol{\omega} \in \Omega. \quad \text{(VIP)}$$

When the operator $F$ is monotone, we have that $F(\boldsymbol{\omega}^*)^\top (\boldsymbol{\omega} - \boldsymbol{\omega}^*) \leq F(\boldsymbol{\omega})^\top (\boldsymbol{\omega} - \boldsymbol{\omega}^*), \forall \boldsymbol{\omega}, \boldsymbol{\omega}^*$. Hence, in this case (VIP) implies a stronger formulation sometimes called *Minty variational inequality* (Crespi et al., 2005):

$$\text{find } \boldsymbol{\omega}^* \in \Omega \quad \text{such that} \quad F(\boldsymbol{\omega})^\top (\boldsymbol{\omega} - \boldsymbol{\omega}^*) \geq 0, \quad \forall \boldsymbol{\omega} \in \Omega. \quad \text{(MVI)}$$

This formulation is stronger in the sense that if (MVI) holds for some $\boldsymbol{\omega}^* \in \Omega$, then (VIP) holds too. A *merit function* useful for our analysis can be derived from this formulation. Roughly, a merit function is a convergence measure. More formally, a function $g : \Omega \to \mathbb{R}$ is called a *merit function* if $g$ is non-negative such that $g(\boldsymbol{\omega}) = 0 \Leftrightarrow \boldsymbol{\omega} \in \Omega^*$ (Larsson and Patriksson, 1994). A way to derive a merit function from (MVI) would be to use $g(\boldsymbol{\omega}^*) = \sup_{\boldsymbol{\omega} \in \Omega} F(\boldsymbol{\omega})^\top (\boldsymbol{\omega}^* - \boldsymbol{\omega})$ which is zero if and only if (MVI) holds for $\boldsymbol{\omega}^*$. To deal with unbounded constraint sets (leading to a potentially infinite valued function outside of the optimal set), we use the *restricted merit function* (Nesterov, 2007):

$$\text{Err}_R(\boldsymbol{\omega}_t) \overset{\text{def}}{=} \max_{\boldsymbol{\omega} \in \Omega, \|\boldsymbol{\omega} - \boldsymbol{\omega}_0\| \leq R} F(\boldsymbol{\omega})^\top (\boldsymbol{\omega}_t - \boldsymbol{\omega}). \quad (102)$$

This function acts as merit function for (VIP) on the interior of the open ball of radius $R$ around $\boldsymbol{\omega}_0$, as shown in Lemma 1 of Nesterov (2007). That is, let $\Omega_R \overset{\text{def}}{=} \Omega \cap \{\boldsymbol{\omega} : \|\boldsymbol{\omega} - \boldsymbol{\omega}_0\| < R\}$. Then for any point $\hat{\boldsymbol{\omega}} \in \Omega_R$, we have:

$$\text{Err}_R(\hat{\boldsymbol{\omega}}) = 0 \Leftrightarrow \hat{\boldsymbol{\omega}} \in \Omega^* \cap \Omega_R. \quad (103)$$

The reference point $\boldsymbol{\omega}_0$ is arbitrary, but in practice it is usually the initialization point of the algorithm. $R$ has to be big enough to ensure that $\Omega_R$ contains a solution. $\text{Err}_R$ measures how much (MVI) is violated on the restriction $\Omega_R$. Such merit function is standard in the variational inequality literature. A similar one is used in (Nemirovski, 2004; Juditsky et al., 2011). When $F$ is derived from the gradients (5) of a zero-sum game, we can define a more interpretable merit function. One has to be careful though when extending properties from the minimization setting to the saddle point setting (e.g. the merit function used by Yadav et al. (2018) is vacuous for a bilinear game as explained in App C.2).

In the appendix, we adopt a set of assumptions a little more general than the one in the main paper:

**Assumption 4.**        • *F is* monotone *and* $\Omega$ *is convex and closed.*

- *R is set big enough such that* $R > \|\boldsymbol{\omega}_0 - \boldsymbol{\omega}^*\|$ *and F is a* monotone *operator.*

Contrary to Assumption 3, in Assumption 4 the constraint set in no longer assumed to be bounded. Assumption 4 is implied by Assumption 3 by setting $R$ to the diameter of $\Omega$, and is thus more general.

## C.1    MORE GENERAL MERIT FUNCTIONS

In this appendix, we will note $\mathrm{Err}_R^{(\mathrm{VI})}$ the *restricted merit function* defined in (102). Let us recall its definition,

$$\mathrm{Err}_R^{(\mathrm{VI})}(\boldsymbol{\omega}_t) \stackrel{\mathrm{def}}{=} \max_{\boldsymbol{\omega} \in \Omega, \|\boldsymbol{\omega} - \boldsymbol{\omega}_0\| \leq R} F(\boldsymbol{\omega})^\top (\boldsymbol{\omega}_t - \boldsymbol{\omega}). \tag{104}$$

When the objective is a saddle point problem i.e.,

$$F(\boldsymbol{\theta}, \boldsymbol{\varphi}) = [\nabla_{\boldsymbol{\theta}} \mathcal{L}(\boldsymbol{\theta}, \boldsymbol{\varphi}) \ -\nabla_{\boldsymbol{\varphi}} \mathcal{L}(\boldsymbol{\theta}, \boldsymbol{\varphi})]^\top \tag{105}$$

and $\mathcal{L}$ is *convex-concave* (see Definition 4 in §A), we can use another merit function than (104) on $\Omega_R$ that is more interpretable and more directly related to the cost function of the minimax formulation:

$$\mathrm{Err}_R^{(\mathrm{SP})}(\boldsymbol{\theta}_t, \boldsymbol{\varphi}_t) \stackrel{\mathrm{def}}{=} \max_{\substack{\boldsymbol{\varphi} \in \Phi, \boldsymbol{\theta} \in \Theta \\ \|(\boldsymbol{\theta}, \boldsymbol{\varphi}) - (\boldsymbol{\theta}_0, \boldsymbol{\varphi}_0)\| \leq R}} \mathcal{L}(\boldsymbol{\theta}_t, \boldsymbol{\varphi}) - \mathcal{L}(\boldsymbol{\theta}, \boldsymbol{\varphi}_t). \tag{106}$$

In particular, if the equilibrium $(\boldsymbol{\theta}^*, \boldsymbol{\varphi}^*) \in \Omega^* \cap \Omega_R$ and we have that $\mathcal{L}(\cdot, \boldsymbol{\varphi}^*)$ and $-\mathcal{L}(\boldsymbol{\theta}^*, \cdot)$ are $\mu$-strongly convex (see §A), then the merit function for saddle points upper bounds the distance for $(\boldsymbol{\theta}, \boldsymbol{\varphi}) \in \Omega_R$ to the equilibrium as:

$$\mathrm{Err}_R^{(\mathrm{SP})}(\boldsymbol{\theta}, \boldsymbol{\varphi}) \geq \frac{\mu}{2}(\|\boldsymbol{\theta} - \boldsymbol{\theta}^*\|_2^2 + \|\boldsymbol{\varphi} - \boldsymbol{\varphi}^*\|_2^2). \tag{107}$$

In the appendix, we provide our convergence results with the merit functions (104) and (106), depending on the setup:

$$\mathrm{Err}_R(\boldsymbol{\omega}) \stackrel{\mathrm{def}}{=} \begin{cases} \mathrm{Err}_R^{(\mathrm{SP})}(\boldsymbol{\omega}) & \text{if } F \text{ is a SP operator (5)} \\ \mathrm{Err}_R^{(\mathrm{VI})}(\boldsymbol{\omega}) & \text{otherwise.} \end{cases} \tag{108}$$

## C.2    ON THE IMPORTANCE OF THE MERIT FUNCTION

In this section, we illustrate the fact that one has to be careful when extending results and properties from the minimization setting to the minimax setting (and consequently to the variational inequality setting). Another candidate as a merit function for saddle point optimization would be to naturally extend the suboptimality $f(\boldsymbol{\omega}) - f(\boldsymbol{\omega}^*)$ used in standard minimization (i.e. find $\boldsymbol{\omega}^*$ the minimizer of $f$) to the gap $P(\boldsymbol{\theta}, \boldsymbol{\varphi}) = \mathcal{L}(\boldsymbol{\theta}, \boldsymbol{\varphi}^*) - \mathcal{L}(\boldsymbol{\theta}^*, \boldsymbol{\varphi})$. In a previous analysis of a modification of the stochastic gradient descent (SGD) method for GANs, Yadav et al. (2018) gave their convergence rate on $P$ that they called the "primal-dual" gap. Unfortunately, if we do not assume that the function $\mathcal{L}$ is strongly convex-concave (a stronger assumption defined in §A and which fails for bilinear objective e.g.), $P$ may not be a *merit function*. It can be 0 for a non optimal point, see for instance the discussion on the differences between (106) and $P$ in (Gidel et al., 2017, Section 3). In particular, for the simple 2D bilinear example $\mathcal{L}(\boldsymbol{\theta}, \boldsymbol{\varphi}) = \boldsymbol{\theta} \cdot \boldsymbol{\varphi}$, we have that $\boldsymbol{\theta}^* = \boldsymbol{\varphi}^* = 0$ and thus $P(\boldsymbol{\theta}, \boldsymbol{\varphi}) = 0 \ \ \forall \boldsymbol{\theta}, \boldsymbol{\varphi}$.

## C.3    VARIATIONAL INEQUALITIES FOR NON-CONVEX COST FUNCTIONS

When the cost functions defined in (3) are non-convex, the operator $F$ is no longer monotone. Nevertheless, (VIP) and (MVI) can still be defined, though a solution to (MVI) is less likely to exist. We note that (VIP) is a local condition for $F$ (as only evaluating $F$ at the points $\boldsymbol{\omega}^*$). On the other hand, an appealing property of (MVI) is that it is a global condition. In the context of minimization of a function $f$ for example (where $F = \nabla f$), if $\boldsymbol{\omega}^*$ solves (MVI) then $\boldsymbol{\omega}^*$ is a *global* minimum of $f$ (and not just a stationary point for the solution of (MVI); see Proposition 2.2 from Crespi et al. (2005)).

A less restrictive way to consider variational inequalities in the non-monotone setting is to use a local version of (MVI). If the cost functions are locally convex around the optimal couple $(\boldsymbol{\theta}^*, \boldsymbol{\varphi}^*)$ and if our iterates eventually fall and stay into that neighborhood, then we can consider our restricted merit function $\mathrm{Err}_R(\cdot)$ with a well suited constant $R$ and apply our convergence results for monotone operators.

## D  ANOTHER WAY OF IMPLEMENTING EXTRAPOLATION TO SGD

We now introduce another way to combine extrapolation and SGD. This extension is very similar to AvgExtraSGD Alg. 2, the only difference is that it re-uses the mini-batch sample of the extrapolation step for the update of the current point. The intuition is that it correlates the estimator of the gradient of the extrapolation step and the one of the update step leading to a better correction of the oscillations which are also due to the stochasticity. One emerging issue (for the analysis) of this method is that since $\boldsymbol{\omega}'_t$ depend on $\xi_t$, the quantity $F(\boldsymbol{\omega}'_t, \xi_t)$ is a biased estimator of $F(\boldsymbol{\omega}'_t)$.

---

**Algorithm 5** Re-used mini-batches for stochastic extrapolation (ReExtraSGD)

---

1: Let $\boldsymbol{\omega}_0 \in \Omega$
2: **for** $t = 0 \ldots T - 1$ **do**
3:     Sample $\xi_t \sim P$
4:     $\boldsymbol{\omega}'_t \overset{\text{def}}{=} P_\Omega[\boldsymbol{\omega}_t - \eta_t F(\boldsymbol{\omega}_t, \xi_t)]$               ▷ Extrapolation step
5:     $\boldsymbol{\omega}_{t+1} \overset{\text{def}}{=} P_\Omega[\boldsymbol{\omega}_t - \eta_t F(\boldsymbol{\omega}'_t, \xi_t)]$            ▷ Update step with the **same** sample
6: **end for**
7: Return $\bar{\boldsymbol{\omega}}_T = \sum_{t=0}^{T-1} \eta_t \boldsymbol{\omega}'_t / \sum_{t=0}^{T-1} \eta_t$

---

**Theorem 5.** *Assume that* $\|\boldsymbol{\omega}'_t - \boldsymbol{\omega}_0\| \leq R$, $\forall t \geq 0$ *where* $(\boldsymbol{\omega}'_t)_{t \geq 0}$ *are the iterates of Alg. 5. Under Assumption 1 and 4, for any* $T \geq 1$, *Alg. 5 with constant step-size* $\eta \leq \frac{1}{\sqrt{2}L}$ *has the following convergence properties:*

$$\mathbb{E}[\mathrm{Err}_R(\bar{\boldsymbol{\omega}}_T)] \leq \frac{R^2}{\eta T} + \eta \frac{\sigma^2 + 4L^2(4R^2 + \sigma^2)}{2} \quad where \quad \bar{\boldsymbol{\omega}}_T \overset{\text{def}}{=} \frac{1}{T} \sum_{t=0}^{T-1} \boldsymbol{\omega}'_t.$$

*Particularly,* $\eta_t = \frac{\eta}{\sqrt{T}}$ *gives* $\mathbb{E}[\mathrm{Err}_R(\bar{\boldsymbol{\omega}}_T)] \leq \frac{O(1)}{\sqrt{T}}$.

The assumption that the sequence of the iterates provided by the algorithm is bounded is strong, but has also been made for instance in (Yadav et al., 2018). The proof of this result is provided in §F.

## E  VARIANCE COMPARISON BETWEEN AVGSGD AND SGD WITH PREDICTION METHOD

To compare the variance term of AvgSGD in (26) with the one of the *SGD with prediction method* (Yadav et al., 2018), we need to have the same convergence certificate. Fortunately, their proof can be adapted to our convergence criterion (using Lemma 7 in §F), revealing an extra $\sigma^2/2$ in the variance term from their paper. The resulting variance can be summarized with our notation as $(M^2(1 + L) + \sigma^2)/2$ where the $L$ is the Lipschitz constant of the operator $F$. Since $M \gg \sigma$, their variance term is then $1 + L$ time larger than the one provided by the AvgSGD method.

## F  PROOF OF THEOREMS

This section is dedicated on the proof of the theorems provided in this paper in a slightly more general form working with the merit function defined in (108). First we prove an additional lemma necessary to the proof of our theorems.

**Lemma 7.** *Let $F$ be a monotone operator and let* $(\boldsymbol{\omega}_t), (\boldsymbol{\omega}'_t), (\boldsymbol{z}_t), (\Delta_t), (\xi_t)$ *and* $(\zeta_t)$ *be six random sequences such that, for all* $t \geq 0$

$$2\eta_t F(\boldsymbol{\omega}'_t)^\top (\boldsymbol{\omega}'_t - \boldsymbol{\omega}) \leq N_t - N_{t+1} + \eta_t^2 (M_1(\boldsymbol{\omega}_t, \xi_t) + M_2(\boldsymbol{\omega}'_t, \zeta_t)) + 2\eta_t \Delta_t^\top (\boldsymbol{z}_t - \boldsymbol{\omega}),$$

where $N_t = N(\boldsymbol{\omega}_t, \boldsymbol{\omega}'_{t-1}, \boldsymbol{\omega}'_{t-2}) \geq 0$ and we extend $(\boldsymbol{\omega}'_t)$ with $\boldsymbol{\omega}'_{-2} = \boldsymbol{\omega}'_{-1} = \boldsymbol{\omega}'_0$. Let also assume that with $N_0 \leq R$, $\mathbb{E}[\|\Delta_t\|_2^2] \leq \sigma^2$, $\mathbb{E}[\Delta_t | \boldsymbol{z}_t, \Delta_0, \ldots, \Delta_{t-1}] = 0$, $\mathbb{E}[M_1(\boldsymbol{\omega}_t, \xi_t)] \leq M_1$ and $\mathbb{E}[M_2(\boldsymbol{\omega}'_t, \zeta_t)] \leq M_2$, then,

$$\mathbb{E}[\mathrm{Err}_R(\bar{\boldsymbol{\omega}}_T)] \leq \frac{R^2}{S_T} + \frac{M_1 + M_2 + \sigma^2}{2S_T} \sum_{t=0}^{T-1} \eta_t^2 \tag{109}$$

where $\bar{\boldsymbol{\omega}}_T \overset{def}{=} \sum_{t=0}^{T-1} \eta_t \boldsymbol{\omega}'_t / S_T$ and $S_T \overset{def}{=} \sum_{t=0}^{T-1} \eta_t$.

***Proof of Lemma*** *7.* We sum (7) for $0 \leq t \leq T-1$ to get,

$$2 \sum_{t=0}^{T-1} \eta_t F(\boldsymbol{\omega}'_t)^\top (\boldsymbol{\omega}'_t - \boldsymbol{\omega}) \leq$$

$$\sum_{t=0}^{T-1} \left[ (N_t - N_{t+1}) + \eta_t^2 ((M_1(\boldsymbol{\omega}_t, \xi_t) + M_2(\boldsymbol{\omega}'_t, \zeta_t)) + 2\eta_t \Delta_t^\top (\boldsymbol{z}_t - \boldsymbol{\omega}) \right]. \tag{110}$$

We will then upper bound each sum in the right-hand side,

$$\Delta_t^\top (\boldsymbol{z}_t - \boldsymbol{\omega}) = \Delta_t^\top (\boldsymbol{z}_t - \boldsymbol{u}_t) + \Delta_t^\top (\boldsymbol{u}_t - \boldsymbol{\omega})$$

where $\boldsymbol{u}_{t+1} \overset{def}{=} P_\Omega(\boldsymbol{u}_t - \eta_t \Delta_t)$ and $\boldsymbol{u}_0 \overset{def}{=} \boldsymbol{\omega}_0$. Then,

$$\|\boldsymbol{u}_{t+1} - \boldsymbol{\omega}\|_2^2 \leq \|\boldsymbol{u}_t - \boldsymbol{\omega}\|_2^2 - 2\eta_t \Delta_t^\top (\boldsymbol{u}_t - \boldsymbol{\omega}) + \eta_t^2 \|\Delta_t\|_2^2$$

leading to

$$2\eta_t \Delta_t^\top (\boldsymbol{z}_t - \boldsymbol{\omega}) \leq 2\eta_t \Delta_t^\top (\boldsymbol{z}_t - \boldsymbol{u}_t) + \|\boldsymbol{u}_t - \boldsymbol{\omega}\|_2^2 - \|\boldsymbol{u}_{t+1} - \boldsymbol{\omega}\|_2^2 + \eta_t^2 \|\Delta_t\|_2^2 \tag{111}$$

Then noticing that $\boldsymbol{z}_0 \overset{def}{=} \boldsymbol{\omega}_0$, back to (110) we get a telescoping sum,

$$2 \sum_{t=0}^{T-1} \eta_t F(\boldsymbol{\omega}'_t)^\top (\boldsymbol{\omega}'_t - \boldsymbol{\omega}) \leq 2N_0 + \sum_{t=0}^{T-1} \left[ \eta_t^2 ((M_1(\boldsymbol{\omega}_t, \xi_t) + M_2(\boldsymbol{\omega}'_t, \zeta_t)) + \|\Delta_t\|_2^2) + 2\eta_t \Delta_t^\top (\boldsymbol{z}_t - \boldsymbol{u}_t) \right]. \tag{112}$$

If $F$ is the operator of a convex-concave saddle point (5), we get, with $\boldsymbol{\omega}'_t = (\boldsymbol{\theta}_t, \boldsymbol{\varphi}_t)$

$$\begin{aligned} F(\boldsymbol{\omega}'_t)^\top (\boldsymbol{\omega}'_t - \boldsymbol{\omega}) &\geq \nabla_{\boldsymbol{\theta}} \mathcal{L}(\boldsymbol{\theta}_t, \boldsymbol{\varphi}_t)^\top (\boldsymbol{\theta}_t - \boldsymbol{\theta}) - \nabla_{\boldsymbol{\varphi}} \mathcal{L}(\boldsymbol{\theta}_t, \boldsymbol{\varphi}_t)^\top (\boldsymbol{\varphi}_t - \boldsymbol{\varphi}) \\ &\geq \mathcal{L}(\boldsymbol{\theta}_t, \boldsymbol{\varphi}) - \mathcal{L}(\boldsymbol{\theta}_t, \boldsymbol{\varphi}_t) + \mathcal{L}(\boldsymbol{\theta}_t, \boldsymbol{\varphi}_t) - \mathcal{L}(\boldsymbol{\theta}, \boldsymbol{\varphi}_t) \\ &\qquad \text{(by convexity and concavity)} \\ &= \mathcal{L}(\boldsymbol{\theta}_t, \boldsymbol{\varphi}) - \mathcal{L}(\boldsymbol{\theta}, \boldsymbol{\varphi}_t) \end{aligned}$$

then by convexity of $\mathcal{L}(\cdot, \boldsymbol{\varphi})$ and concavity of $\mathcal{L}(\boldsymbol{\theta}, \cdot)$, we have that,

$$2S_T \sum_{t=0}^{T-1} \frac{\eta_t}{S_T} F(\boldsymbol{\omega}'_t)^\top (\boldsymbol{\omega}'_t - \boldsymbol{\omega}) \geq 2S_T \sum_{t=0}^{T-1} \frac{\eta_t}{S_T} (\mathcal{L}(\boldsymbol{\theta}_t, \boldsymbol{\varphi}) - \mathcal{L}(\boldsymbol{\theta}, \boldsymbol{\varphi}_t)) \geq \bar{2}S_T (\mathcal{L}(\bar{\boldsymbol{\theta}}_t, \boldsymbol{\varphi}) - \mathcal{L}(\bar{\boldsymbol{\theta}}, \boldsymbol{\varphi}_t)) \tag{113}$$

Otherwise if the operator $F$ is just monotone since $F(\boldsymbol{\omega}'_t)^\top (\boldsymbol{\omega}'_t - \boldsymbol{\omega}) \geq F(\boldsymbol{\omega}')^\top (\boldsymbol{\omega}'_t - \boldsymbol{\omega})$ we have that

$$2S_T \sum_{t=0}^{T-1} \eta_t F(\boldsymbol{\omega}'_t)^\top (\boldsymbol{\omega}'_t - \boldsymbol{\omega}) \geq 2S_T \sum_{t=0}^{T-1} \eta_t F(\boldsymbol{\omega}')^\top (\boldsymbol{\omega}'_t - \boldsymbol{\omega}) = 2S_T F(\boldsymbol{\omega}')^\top (\bar{\boldsymbol{\omega}}_t - \boldsymbol{\omega}) \tag{114}$$

In both cases, we can now maximize the left hand side respect to $\boldsymbol{\omega}$ (since the RHS does not depend on $\boldsymbol{\omega}$) to get,

$$2S_T \, \mathrm{Err}_R(\bar{\boldsymbol{\omega}}_t) \leq 2R^2 + \sum_{t=0}^{T-1} \left[ \eta_t^2 ((M_1(\boldsymbol{\omega}_t, \xi_t) + M_2(\boldsymbol{\omega}'_t, \zeta_t)) + \|\Delta_t\|_2^2) + 2\eta_t \Delta_t^\top (\boldsymbol{z}_t - \boldsymbol{u}_t) \right]. \tag{115}$$

Then taking the expectation, since $\mathbb{E}[\Delta_t | \boldsymbol{z}_t, \boldsymbol{u}_t] = \mathbb{E}[\Delta_t | \boldsymbol{z}_t, \Delta_0, \ldots, \Delta_{t-1}] = 0$, $\mathbb{E}_{\zeta_t}[\|\Delta_t\|_2^2] \leq \sigma^2$, $\mathbb{E}_{\xi_t}[M_1(\boldsymbol{\omega}_t, \xi_t)] \leq M_1$ and $\mathbb{E}_{\zeta_t}[M_2(\boldsymbol{\omega}'_t, \zeta_t)] \leq M_2$, we get that,

$$\mathbb{E}[\mathrm{Err}_R(\bar{\boldsymbol{\omega}}_T)] \leq \frac{R^2}{S_T} + \frac{M_1 + M_2 + \sigma^2}{2S_T} \sum_{t=0}^{T-1} \eta_t^2 \tag{116}$$

$$\square$$

### F.1 PROOF OF THM. 2

First let us state Theorem 2 in its general form,

**Theorem' 2.** *Under Assumption 1, 2 and 4, Alg. 1 with constant step-size $\eta$ has the following convergence rate for all $T \geq 1$,*

$$\mathbb{E}[\mathrm{Err}_R(\bar{\boldsymbol{\omega}}_T)] \leq \frac{R^2}{2\eta T} + \eta \frac{M^2 + \sigma^2}{2} \quad where \quad \bar{\boldsymbol{\omega}}_T \overset{def}{=} \frac{1}{T} \sum_{t=0}^{T-1} \boldsymbol{\omega}_t \,. \tag{117}$$

*Particularly, $\eta = \frac{R}{\sqrt{T(M^2+\sigma^2)}}$ gives $\mathbb{E}[\mathrm{Err}_R(\bar{\boldsymbol{\omega}}_T)] \leq \frac{R\sqrt{M^2+\sigma^2}}{\sqrt{T}}$ .*

***Proof of Theorem 2.*** Let any $\boldsymbol{\omega} \in \Omega$ such that $\|\boldsymbol{\omega}_0 - \boldsymbol{\omega}\|_2 \leq R$,

$$
\begin{aligned}
\|\boldsymbol{\omega}_{t+1} - \boldsymbol{\omega}\|_2^2 &= \|P_\Omega(\boldsymbol{\omega}_t - \eta_t F(\boldsymbol{\omega}_t, \xi_t)) - \boldsymbol{\omega}\|_2^2 \\
&\leq \|\boldsymbol{\omega}_t - \eta_t F(\boldsymbol{\omega}_t, \xi_t)) - \boldsymbol{\omega}\|_2^2 \\
&\quad \text{(projections are non-contractive, Lemma 1)} \\
&= \|\boldsymbol{\omega}_t - \boldsymbol{\omega}\|_2^2 - 2\eta_t F(\boldsymbol{\omega}_t, \xi_t)^\top (\boldsymbol{\omega}_t - \boldsymbol{\omega}) + \|\eta_t F(\boldsymbol{\omega}_t, \xi_t)\|_2^2
\end{aligned}
$$

Then we can make appear the quantity $F(\boldsymbol{\omega}_t)^\top (\boldsymbol{\omega}_t - \boldsymbol{\omega})$ on the left-hand side,

$$2\eta_t F(\boldsymbol{\omega}_t)^\top (\boldsymbol{\omega}_t - \boldsymbol{\omega}) \leq \|\boldsymbol{\omega}_t - \boldsymbol{\omega}\|_2^2 - \|\boldsymbol{\omega}_{t+1} - \boldsymbol{\omega}\|_2^2 + \eta_t^2 \|F(\boldsymbol{\omega}_t, \xi_t)\|_2^2 + 2\eta_t (F(\boldsymbol{\omega}_t) - F(\boldsymbol{\omega}_t, \xi_t))^\top (\boldsymbol{\omega}_t - \boldsymbol{\omega}) \tag{118}$$

we can sum (118) for $0 \leq t \leq T - 1$ to get,

$$2 \sum_{t=0}^{T-1} \eta_t F(\boldsymbol{\omega}_t)^\top (\boldsymbol{\omega}_t - \boldsymbol{\omega}) \leq$$

$$\sum_{t=0}^{T-1} \left[ (\|\boldsymbol{\omega}_t - \boldsymbol{\omega}\|^2 - \|\boldsymbol{\omega}_{t+1} - \boldsymbol{\omega}\|^2) + \eta_t^2 \|F(\boldsymbol{\omega}_t, \xi_t)\|_2^2 + 2\eta_t \Delta_t^\top (\boldsymbol{\omega}_t - \boldsymbol{\omega}) \right] \tag{119}$$

where we noted $\Delta_t \overset{def}{=} F(\boldsymbol{\omega}_t) - F(\boldsymbol{\omega}_t, \xi_t)$.
By monotonicity, $F(\boldsymbol{\omega}_t)^\top (\boldsymbol{\omega}_t - \boldsymbol{\omega}) \geq F(\boldsymbol{\omega})^\top (\boldsymbol{\omega}_t - \boldsymbol{\omega})$ we get,

$$2S_T F(\boldsymbol{\omega})^\top (\bar{\boldsymbol{\omega}}_T - \boldsymbol{\omega}) \leq \sum_{t=0}^{T-1} \left[ (\|\boldsymbol{\omega}_t - \boldsymbol{\omega}\|^2 - \|\boldsymbol{\omega}_{t+1} - \boldsymbol{\omega}\|^2) + \eta_t^2 \|F(\boldsymbol{\omega}_t, \xi_t)\|_2^2 + 2\eta_t \Delta_t^\top (\boldsymbol{\omega}_t - \boldsymbol{\omega}) \right] , \tag{120}$$

where $S_T \overset{def}{=} \sum_{t=0}^{T-1} \eta_t$ and $\bar{\boldsymbol{\omega}}_T \overset{def}{=} \frac{1}{S_T} \sum_{t=0}^{T-1} \eta_t \boldsymbol{\omega}_t$.
We will then upper bound each sum in the right hand side,

$$\Delta_t^\top (\boldsymbol{\omega}_t - \boldsymbol{\omega}) = \Delta_t^\top (\boldsymbol{\omega}_t - \boldsymbol{u}_t) + \Delta_t^\top (\boldsymbol{u}_t - \boldsymbol{\omega})$$

where $\boldsymbol{u}_{t+1} \overset{def}{=} P_\Omega(\boldsymbol{u}_t - \eta_t \Delta_t)$ and $\boldsymbol{u}_0 = \boldsymbol{\omega}_0$. Then,

$$\|\boldsymbol{u}_{t+1} - \boldsymbol{\omega}\|_2^2 \leq \|\boldsymbol{u}_t - \boldsymbol{\omega}\|_2^2 - 2\eta_t \Delta_t^\top (\boldsymbol{u}_t - \boldsymbol{\omega}) + \eta_t^2 \|\Delta_t\|_2^2$$

leading to

$$2\eta_t \Delta_t^\top (\boldsymbol{\omega}_t - \boldsymbol{\omega}) \leq 2\eta_t \Delta_t^\top (\boldsymbol{\omega}_t - \boldsymbol{u}_t) + \|\boldsymbol{u}_t - \boldsymbol{\omega}\|_2^2 - \|\boldsymbol{u}_{t+1} - \boldsymbol{\omega}\|_2^2 + \eta_t^2 \|\Delta_t\|_2^2 \tag{121}$$

Then noticing that $\boldsymbol{u}_0 \overset{def}{=} \boldsymbol{\omega}_0$, back to (120) we get a telescoping sum,

$$2S_T F(\boldsymbol{\omega})^\top (\bar{\boldsymbol{\omega}}_T - \boldsymbol{\omega}) \leq 2\|\boldsymbol{\omega}_0 - \boldsymbol{\omega}\|^2 + \sum_{t=0}^{T-1} \eta_t^2 (\|F(\boldsymbol{\omega}_t, \xi_t)\|_2^2 + \|\Delta_t\|_2^2) + 2 \sum_{t=0}^{T-1} \eta_t \Delta_t^\top (\boldsymbol{\omega}_t - \boldsymbol{u}_t)$$

$$\leq 2R + \sum_{t=0}^{T-1} \eta_t^2 (\|F(\boldsymbol{\omega}_t, \xi_t)\|_2^2 + \|\Delta_t\|_2^2) + 2 \sum_{t=0}^{T-1} \eta_t \Delta_t^\top (\boldsymbol{\omega}_t - \boldsymbol{u}_t)$$

Then the right hand side does not depends on $\boldsymbol{\omega}$, we can maximize over $\boldsymbol{\omega}$ to get,

$$2S_T \operatorname{Err}_R(\bar{\boldsymbol{\omega}}_T) \leq 2R + \sum_{t=0}^{T-1} \eta_t^2 (\|F(\boldsymbol{\omega}_t, \xi_t)\|_2^2 + \|\Delta_t\|_2^2) + 2\sum_{t=0}^{T-1} \eta_t \Delta_t^\top (\boldsymbol{\omega}_t - \boldsymbol{u}_t) \tag{122}$$

Noticing that $\mathbb{E}[\Delta_t | \boldsymbol{\omega}_t, \boldsymbol{u}_t] = 0$ (the estimates of $F$ are unbiased), by Assumption 2 $\mathbb{E}[(\|F(\boldsymbol{\omega}_t, \xi_t)\|_2^2] \leq M^2$ and by Assumption 1 $\mathbb{E}[\|\Delta_t\|_2^2] \leq \sigma^2$ we get,

$$\mathbb{E}[\operatorname{Err}_R(\bar{\boldsymbol{\omega}}_T)] \leq \frac{R}{S_T} + \frac{M^2 + \sigma^2}{2S_T} \sum_{t=0}^{T-1} \eta_t^2 \tag{123}$$

particularly for $\eta_t = \eta$ and $\eta_t = \frac{\eta}{\sqrt{t+1}}$ we respectively get,

$$\mathbb{E}[\operatorname{Err}_R(\bar{\boldsymbol{\omega}}_T)] \leq \frac{2R}{\eta T} + \frac{\eta}{2}(M^2 + \sigma^2) \tag{124}$$

and

$$\mathbb{E}[\operatorname{Err}_R(\bar{\boldsymbol{\omega}}_T)] \leq \frac{4R}{\eta\sqrt{T+1}-1} + 2\eta \ln(T+1) \frac{M^2 + \sigma^2}{\sqrt{T+1}-1} \tag{125}$$

$\square$

## F.2 PROOF OF THM. 3

**Theorem' 3.** *Under Assumption 1 and 4, if $\mathbb{E}_\xi[F]$ is L-Lipschitz, then Alg. 2 with a constant step-size $\eta \leq \frac{1}{\sqrt{3}L}$ has the following convergence rate for any $T \geq 1$,*

$$\mathbb{E}[\operatorname{Err}_R(\bar{\boldsymbol{\omega}}_T)] \leq \frac{R^2}{\eta T} + \frac{7}{2}\eta\sigma^2 \quad \text{where} \quad \bar{\boldsymbol{\omega}}_T \stackrel{def}{=} \frac{1}{T} \sum_{t=0}^{T-1} \boldsymbol{\omega}_t'. \tag{126}$$

*Particularly, $\eta = \frac{\sqrt{2}R}{\sigma\sqrt{7T}}$ gives $\mathbb{E}[\operatorname{Err}_R(\bar{\boldsymbol{\omega}}_T)] \leq \frac{\sqrt{14}R\sigma}{\sqrt{T}}$.*

**Proof of Thm. 3.** Let any $\boldsymbol{\omega} \in \Omega$ such that $\|\boldsymbol{\omega}_0 - \boldsymbol{\omega}\|_2 \leq R$. Then, the update rules become $\boldsymbol{\omega}_{t+1} = P_\Omega(\boldsymbol{\omega}_t - \eta_t F(\boldsymbol{\omega}_t', \zeta_t))$ and $\boldsymbol{\omega}_t' = P_\Omega(\boldsymbol{\omega}_t - \eta F(\boldsymbol{\omega}_t, \xi_t))$. We start by applying Lemma 2 for $(\boldsymbol{\omega}, \boldsymbol{u}, \boldsymbol{\omega}', \boldsymbol{\omega}^+) = (\boldsymbol{\omega}_t, -\eta F(\boldsymbol{\omega}_t', \zeta_t), \boldsymbol{\omega}, \boldsymbol{\omega}_{t+1})$ and $(\boldsymbol{\omega}, \boldsymbol{u}, \boldsymbol{\omega}', \boldsymbol{\omega}^+) = (\boldsymbol{\omega}_t, -\eta_t F(\boldsymbol{\omega}_t, \xi_t), \boldsymbol{\omega}_{t+1}, \boldsymbol{\omega}_t')$,

$$\|\boldsymbol{\omega}_{t+1} - \boldsymbol{\omega}\|_2^2 \leq \|\boldsymbol{\omega}_t - \boldsymbol{\omega}\|_2^2 - 2\eta_t F(\boldsymbol{\omega}_t', \zeta_t)^\top (\boldsymbol{\omega}_{t+1} - \boldsymbol{\omega}) - \|\boldsymbol{\omega}_{t+1} - \boldsymbol{\omega}_t\|_2^2$$

$$\|\boldsymbol{\omega}_t' - \boldsymbol{\omega}_{t+1}\|_2^2 \leq \|\boldsymbol{\omega}_t - \boldsymbol{\omega}_{t+1}\|_2^2 - 2\eta_t F(\boldsymbol{\omega}_t, \xi_t)^\top (\boldsymbol{\omega}_t' - \boldsymbol{\omega}_{t+1}) - \|\boldsymbol{\omega}_t' - \boldsymbol{\omega}_t\|_2^2$$

Then, summing them we get

$$\|\boldsymbol{\omega}_{t+1} - \boldsymbol{\omega}\|_2^2 \leq \|\boldsymbol{\omega}_t - \boldsymbol{\omega}\|_2^2 - 2\eta_t F(\boldsymbol{\omega}_t', \zeta_t)^\top (\boldsymbol{\omega}_{t+1} - \boldsymbol{\omega})$$
$$- 2\eta_t F(\boldsymbol{\omega}_t, \xi_t)^\top (\boldsymbol{\omega}_t' - \boldsymbol{\omega}_{t+1}) - \|\boldsymbol{\omega}_t - \boldsymbol{\omega}_t'\|_2^2 - \|\boldsymbol{\omega}_{t+1} - \boldsymbol{\omega}_t'\|_2^2 \tag{127}$$

leading to

$$\|\boldsymbol{\omega}_{t+1} - \boldsymbol{\omega}\|_2^2 \leq \|\boldsymbol{\omega}_t - \boldsymbol{\omega}\|_2^2 - 2\eta_t F(\boldsymbol{\omega}_t', \zeta_t)^\top (\boldsymbol{\omega}_t' - \boldsymbol{\omega})$$
$$+ 2\eta_t (F(\boldsymbol{\omega}_t', \zeta_t) - F(\boldsymbol{\omega}_t, \xi_t))^\top (\boldsymbol{\omega}_t' - \boldsymbol{\omega}_{t+1}) - \|\boldsymbol{\omega}_t - \boldsymbol{\omega}_t'\|_2^2 - \|\boldsymbol{\omega}_{t+1} - \boldsymbol{\omega}_t'\|_2^2$$

Then with $2\boldsymbol{a}^\top \boldsymbol{b} \leq \|\boldsymbol{a}\|_2^2 + \|\boldsymbol{b}\|_2^2$ we get

$$\|\boldsymbol{\omega}_{t+1} - \boldsymbol{\omega}\|_2^2 \leq \|\boldsymbol{\omega}_t - \boldsymbol{\omega}\|_2^2 - 2\eta_t F(\boldsymbol{\omega}_t', \zeta_t)^\top (\boldsymbol{\omega}_t' - \boldsymbol{\omega})$$
$$- \|\boldsymbol{\omega}_t - \boldsymbol{\omega}_t'\|_2^2 + \eta_t^2 \|F(\boldsymbol{\omega}_t', \zeta_t) - F(\boldsymbol{\omega}_t, \xi_t)\|_2^2$$

Using the inequality $\|\boldsymbol{a} + \boldsymbol{b} + \boldsymbol{c}\|_2^2 \leq 3(\|\boldsymbol{a}\|_2^2 + \|\boldsymbol{b}\|_2^2 + \|\boldsymbol{c}\|_2^2)$ we get,

$$\|\boldsymbol{\omega}_{t+1} - \boldsymbol{\omega}\|_2^2 \leq \|\boldsymbol{\omega}_t - \boldsymbol{\omega}\|_2^2 - 2\eta_t F(\boldsymbol{\omega}_t', \zeta_t)^\top (\boldsymbol{\omega}_t' - \boldsymbol{\omega}) - \|\boldsymbol{\omega}_t - \boldsymbol{\omega}_t'\|_2^2$$
$$+ 3\eta_t^2 (\|F(\boldsymbol{\omega}_t) - F(\boldsymbol{\omega}_t, \xi_t)\|_2^2 + \|F(\boldsymbol{\omega}_t') - F(\boldsymbol{\omega}_t', \zeta_t)\|_2^2 + \|F(\boldsymbol{\omega}_t') - F(\boldsymbol{\omega}_t)\|_2^2)$$

Then we can use the $L$-Lipschitzness of $F$ to get,

$$\|\boldsymbol{\omega}_{t+1} - \boldsymbol{\omega}\|_2^2 \leq \|\boldsymbol{\omega}_t - \boldsymbol{\omega}\|_2^2 - 2\eta_t F(\boldsymbol{\omega}'_t, \zeta_t)^\top (\boldsymbol{\omega}'_t - \boldsymbol{\omega}) - \|\boldsymbol{\omega}_t - \boldsymbol{\omega}'_t\|_2^2$$
$$+ 3\eta_t^2 (\|F(\boldsymbol{\omega}_t) - F(\boldsymbol{\omega}_t, \xi_t)\|_2^2 + \|F(\boldsymbol{\omega}'_t) - F(\boldsymbol{\omega}'_t, \zeta_t)\|_2^2 + L^2 \|\boldsymbol{\omega}_t - \boldsymbol{\omega}'_t\|_2^2)$$

As we restricted the step-size to $\eta_t \leq \frac{1}{\sqrt{3}L}$ we get,

$$2\eta_t F(\boldsymbol{\omega}'_t)^\top (\boldsymbol{\omega}'_t - \boldsymbol{\omega}) \leq \|\boldsymbol{\omega}_t - \boldsymbol{\omega}\|_2^2 - \|\boldsymbol{\omega}_{t+1} - \boldsymbol{\omega}\|_2^2 + 2\eta_t (F(\boldsymbol{\omega}'_t) - F(\boldsymbol{\omega}'_t, \zeta_t))^\top (\boldsymbol{\omega}'_t - \boldsymbol{\omega})$$
$$+ 3\eta_t^2 \|F(\boldsymbol{\omega}_t) - F(\boldsymbol{\omega}_t, \xi_t)\|_2^2 + 3\eta_t^2 \|F(\boldsymbol{\omega}'_t) - F(\boldsymbol{\omega}'_t, \zeta_t)\|_2^2$$

We get a particular case of (7) so we can use Lemma 7 where $N_t = \|\boldsymbol{\omega}_t - \boldsymbol{\omega}\|_2^2$, $M_1(\boldsymbol{\omega}_t, \xi_t) = 3\|F(\boldsymbol{\omega}_t) - F(\boldsymbol{\omega}_t, \xi_t)\|_2^2$, $M_2(\boldsymbol{\omega}'_t, \zeta_t) = 3\|F(\boldsymbol{\omega}'_t) - F(\boldsymbol{\omega}'_t, \zeta_t)\|_2^2$, $\Delta_t = F(\boldsymbol{\omega}'_t) - F(\boldsymbol{\omega}'_t, \zeta_t)$ and $z_t = \boldsymbol{\omega}'_t$. By Assumption 1, $M_1 = M_2 = 3\sigma^2$ and by the fact that $\mathbb{E}[F(\boldsymbol{\omega}'_t) - F(\boldsymbol{\omega}'_t, \zeta_t) | \boldsymbol{\omega}'_t, \Delta_0, \ldots, \Delta_{t-1}] = \mathbb{E}[\mathbb{E}[F(\boldsymbol{\omega}'_t) - F(\boldsymbol{\omega}'_t, \zeta_t) | \boldsymbol{\omega}'_t] | \Delta_0, \ldots, \Delta_{t-1}] = 0$ the hypothesis of Lemma 7 hold and we get,

$$\mathbb{E}[\mathrm{Err}_R(\bar{\boldsymbol{\omega}}_T)] \leq \frac{R^2}{S_T} + \frac{7\sigma^2}{2S_T} \sum_{t=0}^{T-1} \eta_t^2 \tag{128}$$

$\square$

### F.3 PROOF OF THM. 4

**Theorem' 4.** *Under Assumption 1, if $\mathbb{E}_\xi[F]$ is L-Lipschitz, then AvgPastExtraSGD (Alg. 3) with a constant step-size $\eta \leq \frac{1}{2\sqrt{3}L}$ has the following convergence rate for any $T \geq 1$,*

$$\mathbb{E}[\mathrm{Err}_R(\bar{\boldsymbol{\omega}}_T)] \leq \frac{R^2}{\eta T} + \frac{13}{2}\eta\sigma^2 \quad \text{where} \quad \bar{\boldsymbol{\omega}}_T \stackrel{def}{=} \frac{1}{T} \sum_{t=0}^{T-1} \boldsymbol{\omega}'_t. \tag{129}$$

*Particularly, $\eta = \frac{\sqrt{2}R}{\sigma\sqrt{13T}}$ gives $\mathbb{E}[\mathrm{Err}_R(\bar{\boldsymbol{\omega}}_T)] \leq \frac{\sqrt{26}R\sigma}{\sqrt{T}}$.*

First let us recall the update rule

$$\begin{cases} \boldsymbol{\omega}_{t+1} = P_\Omega[\boldsymbol{\omega}_t - \eta_t F(\boldsymbol{\omega}'_t, \xi_t)] \\ \boldsymbol{\omega}'_{t+1} = P_\Omega[\boldsymbol{\omega}_{t+1} - \eta_{t+1} F(\boldsymbol{\omega}'_t, \xi_t)]. \end{cases} \tag{130}$$

**Lemma 8.** *We have for any $\boldsymbol{\omega} \in \Omega$,*

$$2\eta F(\boldsymbol{\omega}'_t, \xi_t)^\top (\boldsymbol{\omega}'_t - \boldsymbol{\omega}) \leq \|\boldsymbol{\omega}_t - \boldsymbol{\omega}\|_2^2 - \|\boldsymbol{\omega}_{t+1} - \boldsymbol{\omega}\|_2^2 - \|\boldsymbol{\omega}'_t - \boldsymbol{\omega}_t\|_2^2 + 3\eta_t^2 L^2 \|\boldsymbol{\omega}'_{t-1} - \boldsymbol{\omega}'_t\|_2^2$$
$$+ 3\eta_t^2 \left[ \|F(\boldsymbol{\omega}'_{t-1}, \xi_{t-1}) - F(\boldsymbol{\omega}'_{t-1})\|_2^2 + \|F(\boldsymbol{\omega}'_t) - F(\boldsymbol{\omega}'_t, \xi_t)\|_2^2 \right]. \tag{131}$$

*Proof.* Applying Lemma 2 for $(\boldsymbol{\omega}, \boldsymbol{u}, \boldsymbol{\omega}^+, \boldsymbol{\omega}') = (\boldsymbol{\omega}_t, -\eta_t F(\boldsymbol{\omega}'_t, \xi_t), \boldsymbol{\omega}_{t+1}, \boldsymbol{\omega})$ and $(\boldsymbol{\omega}, \boldsymbol{u}, \boldsymbol{\omega}^+, \boldsymbol{\omega}') = (\boldsymbol{\omega}_t, -\eta_t F(\boldsymbol{\omega}'_{t-1}, \xi_{t-1}), \boldsymbol{\omega}'_t, \boldsymbol{\omega}_{t+1})$, we get,

$$\|\boldsymbol{\omega}_{t+1} - \boldsymbol{\omega}\|_2^2 \leq \|\boldsymbol{\omega}_t - \boldsymbol{\omega}\|_2^2 - 2\eta_t F(\boldsymbol{\omega}'_t, \xi_t)^\top (\boldsymbol{\omega}_{t+1} - \boldsymbol{\omega}) - \|\boldsymbol{\omega}_{t+1} - \boldsymbol{\omega}_t\|_2^2 \tag{132}$$

and

$$\|\boldsymbol{\omega}'_t - \boldsymbol{\omega}_{t+1}\|_2^2 \leq \|\boldsymbol{\omega}_t - \boldsymbol{\omega}_{t+1}\|_2^2 - 2\eta_t F(\boldsymbol{\omega}'_{t-1}, \xi_{t-1})^\top (\boldsymbol{\omega}'_t - \boldsymbol{\omega}_{t+1}) - \|\boldsymbol{\omega}'_t - \boldsymbol{\omega}_t\|_2^2. \tag{133}$$

Summing (132) and (133) we get,

$$\|\boldsymbol{\omega}_{t+1} - \boldsymbol{\omega}\|_2^2 \leq \|\boldsymbol{\omega}_t - \boldsymbol{\omega}\|_2^2 - 2\eta_t F(\boldsymbol{\omega}'_t, \xi_t)^\top (\boldsymbol{\omega}_{t+1} - \boldsymbol{\omega}) \tag{134}$$
$$- 2\eta_t F(\boldsymbol{\omega}'_{t-1}, \xi_{t-1})^\top (\boldsymbol{\omega}'_t - \boldsymbol{\omega}_{t+1}) - \|\boldsymbol{\omega}'_t - \boldsymbol{\omega}_t\|_2^2 - \|\boldsymbol{\omega}'_t - \boldsymbol{\omega}_{t+1}\|_2^2 \tag{135}$$
$$= \|\boldsymbol{\omega}_t - \boldsymbol{\omega}\|_2^2 - 2\eta_t F(\boldsymbol{\omega}'_t, \xi_t)^\top (\boldsymbol{\omega}'_t - \boldsymbol{\omega}) - \|\boldsymbol{\omega}'_t - \boldsymbol{\omega}_t\|_2^2 - \|\boldsymbol{\omega}'_t - \boldsymbol{\omega}_{t+1}\|_2^2 \tag{136}$$
$$- 2\eta_t (F(\boldsymbol{\omega}'_{t-1}, \xi_{t-1}) - F(\boldsymbol{\omega}'_t, \xi_t))^\top (\boldsymbol{\omega}'_t - \boldsymbol{\omega}_{t+1}). \tag{137}$$

Then, we can use the inequality of arithmetic and geometric means $2a^\top b \le \|a\|_2^2 + \|b\|_2^2$ to get,

$$\|\boldsymbol{\omega}_{t+1} - \boldsymbol{\omega}\|_2^2 \le \|\boldsymbol{\omega}_t - \boldsymbol{\omega}\|_2^2 - 2\eta_t F(\boldsymbol{\omega}_t', \xi_t)^\top(\boldsymbol{\omega}_t' - \boldsymbol{\omega}) + \eta_t^2 \|F(\boldsymbol{\omega}_{t-1}', \xi_{t-1}) - F(\boldsymbol{\omega}_t', \xi_t)\|_2^2$$
$$+ \|\boldsymbol{\omega}_t' - \boldsymbol{\omega}_{t+1}\|_2^2 - \|\boldsymbol{\omega}_t' - \boldsymbol{\omega}_t\|_2^2 - \|\boldsymbol{\omega}_t' - \boldsymbol{\omega}_{t+1}\|_2^2 \tag{138}$$
$$= \|\boldsymbol{\omega}_t - \boldsymbol{\omega}\|_2^2 - 2\eta_t F(\boldsymbol{\omega}_t', \xi_t)^\top(\boldsymbol{\omega}_t' - \boldsymbol{\omega}) \tag{139}$$
$$+ \eta_t^2 \|F(\boldsymbol{\omega}_{t-1}', \xi_{t-1}) - F(\boldsymbol{\omega}_t', \xi_t)\|_2^2 - \|\boldsymbol{\omega}_t' - \boldsymbol{\omega}_t\|_2^2. \tag{140}$$

Using the inequality $\|a + b + c\|_2^2 \le 3(\|a\|_2^2 + \|b\|_2^2 + \|c\|_2^2)$ we get,

$$\|F(\boldsymbol{\omega}_{t-1}', \xi_{t-1}) - F(\boldsymbol{\omega}_t', \xi_t)\|_2^2 \le 3(\|F(\boldsymbol{\omega}_{t-1}', \xi_{t-1}) - F(\boldsymbol{\omega}_{t-1}')\|_2^2 + \|F(\boldsymbol{\omega}_{t-1}') - F(\boldsymbol{\omega}_t')\|_2^2$$
$$+ \|F(\boldsymbol{\omega}_t') - F(\boldsymbol{\omega}_t', \xi_t)\|_2^2) \tag{141}$$
$$\le 3(\|F(\boldsymbol{\omega}_{t-1}', \xi_{t-1}) - F(\boldsymbol{\omega}_{t-1}')\|_2^2 + L^2\|\boldsymbol{\omega}_{t-1}' - \boldsymbol{\omega}_t'\|_2^2$$
$$+ \|F(\boldsymbol{\omega}_t') - F(\boldsymbol{\omega}_t', \xi_t)\|_2^2), \tag{142}$$

where we used the $L$-Lipschitzness of $F$ for the last inequality.

Combining (140) with (142) we get,

$$\|\boldsymbol{\omega}_{t+1} - \boldsymbol{\omega}\|_2^2 \le \|\boldsymbol{\omega}_t - \boldsymbol{\omega}\|_2^2 - 2\eta_t F(\boldsymbol{\omega}_t', \xi_t)^\top(\boldsymbol{\omega}_t' - \boldsymbol{\omega}) - \|\boldsymbol{\omega}_t' - \boldsymbol{\omega}_t\|_2^2 + 3\eta_t^2 L^2 \|\boldsymbol{\omega}_{t-1}' - \boldsymbol{\omega}_t'\|_2^2$$
$$+ 3\eta_t^2 \big[\|F(\boldsymbol{\omega}_{t-1}', \xi_{t-1}) - F(\boldsymbol{\omega}_{t-1}')\|_2^2 + \|F(\boldsymbol{\omega}_t') - F(\boldsymbol{\omega}_t', \xi_t)\|_2^2\big]. \tag{143}$$

$\square$

**Lemma 9.** *For all $t \ge 0$, if we set $\boldsymbol{\omega}_{-2}' = \boldsymbol{\omega}_{-1}' = \boldsymbol{\omega}_0'$ we have*

$$\|\boldsymbol{\omega}_{t-1}' - \boldsymbol{\omega}_t'\|_2^2 \le 4\|\boldsymbol{\omega}_t - \boldsymbol{\omega}_t'\|_2^2 + 12\eta_{t-1}^2\big(\|F(\boldsymbol{\omega}_{t-1}', \xi_{t-1}) - F(\boldsymbol{\omega}_{t-1}')\|_2^2 + L^2\|\boldsymbol{\omega}_{t-1}' - \boldsymbol{\omega}_{t-2}'\|_2^2$$
$$+ \|F(\boldsymbol{\omega}_{t-2}') - F(\boldsymbol{\omega}_{t-2}', \xi_{t-2})\|_2^2)\big) - \|\boldsymbol{\omega}_{t-1}' - \boldsymbol{\omega}_t'\|_2^2. \tag{144}$$

*Proof.* We start with $\|a + b\|_2^2 \le 2\|a\|^2 + 2\|b\|^2$.

$$\|\boldsymbol{\omega}_{t-1}' - \boldsymbol{\omega}_t'\|_2^2 \le 2\|\boldsymbol{\omega}_t - \boldsymbol{\omega}_t'\|_2^2 + 2\|\boldsymbol{\omega}_t - \boldsymbol{\omega}_{t-1}'\|_2^2. \tag{145}$$

Moreover, since the projection is contractive we have that

$$\|\boldsymbol{\omega}_t - \boldsymbol{\omega}_{t-1}'\|_2^2 \le \|\boldsymbol{\omega}_{t-1} - \eta_{t-1}F(\boldsymbol{\omega}_{t-1}', \xi_{t-1}) - \boldsymbol{\omega}_{t-1} - \eta_{t-1}F(\boldsymbol{\omega}_{t-2}', \xi_{t-2})\|_2^2 \tag{146}$$
$$= \eta_{t-1}^2\|F(\boldsymbol{\omega}_{t-1}', \xi_{t-1}) - F(\boldsymbol{\omega}_{t-2}', \xi_{t-2})\|_2^2 \tag{147}$$
$$\le 3\eta_{t-1}^2\big(\|F(\boldsymbol{\omega}_{t-1}', \xi_{t-1}) - F(\boldsymbol{\omega}_{t-1}')\|_2^2 + L^2\|\boldsymbol{\omega}_{t-1}' - \boldsymbol{\omega}_{t-2}'\|_2^2$$
$$+ \|F(\boldsymbol{\omega}_{t-2}') - F(\boldsymbol{\omega}_{t-2}', \xi_{t-2})\|_2^2\big). \tag{148}$$

where in the last line we used the same inequality as in (142). Combining (144) and (148) we get,

$$\|\boldsymbol{\omega}_{t-1}' - \boldsymbol{\omega}_t'\|_2^2 = 2\|\boldsymbol{\omega}_{t-1}' - \boldsymbol{\omega}_t'\|_2^2 - \|\boldsymbol{\omega}_{t-1}' - \boldsymbol{\omega}_t'\|_2^2 \tag{149}$$
$$\le 4\|\boldsymbol{\omega}_t - \boldsymbol{\omega}_t'\|_2^2 + 4\|\boldsymbol{\omega}_t - \boldsymbol{\omega}_{t-1}'\|_2^2 - \|\boldsymbol{\omega}_{t-1}' - \boldsymbol{\omega}_t'\|_2^2 \tag{150}$$
$$\le 4\|\boldsymbol{\omega}_t - \boldsymbol{\omega}_t'\|_2^2 + 12\eta_{t-1}^2\big(\|F(\boldsymbol{\omega}_{t-1}', \xi_{t-1}) - F(\boldsymbol{\omega}_{t-1}')\|_2^2 + L^2\|\boldsymbol{\omega}_{t-1}' - \boldsymbol{\omega}_{t-2}'\|_2^2$$
$$+ \|F(\boldsymbol{\omega}_{t-2}') - F(\boldsymbol{\omega}_{t-2}', \xi_{t-2})\|_2^2)\big) - \|\boldsymbol{\omega}_{t-1}' - \boldsymbol{\omega}_t'\|_2^2. \tag{151}$$

$\square$

*Proof of Theorem 4.* Combining Lemma 9 and Lemma 8 we get,

$$2\eta_t F(\boldsymbol{\omega}_t', \xi_t)^\top(\boldsymbol{\omega}_t' - \boldsymbol{\omega}) \le \|\boldsymbol{\omega}_t - \boldsymbol{\omega}\|_2^2 - \|\boldsymbol{\omega}_{t+1} - \boldsymbol{\omega}\|_2^2$$
$$+ 36\eta_t^2\eta_{t-1}^2 L^2\big(\|F(\boldsymbol{\omega}_{t-1}', \xi_{t-1}) - F(\boldsymbol{\omega}_{t-1}')\|_2^2 + L^2\|\boldsymbol{\omega}_{t-1}' - \boldsymbol{\omega}_{t-2}'\|_2^2$$
$$+ \|F(\boldsymbol{\omega}_{t-2}') - F(\boldsymbol{\omega}_{t-2}', \xi_{t-2})\|_2^2)$$
$$- 3\eta_t^2 L^2\|\boldsymbol{\omega}_{t-1}' - \boldsymbol{\omega}_t'\|_2^2 + (12\eta_t^2 L^2 - 1)\|\boldsymbol{\omega}_t' - \boldsymbol{\omega}_t\|_2^2$$
$$+ 3\eta_t^2\big[\|F(\boldsymbol{\omega}_{t-1}', \xi_{t-1}) - F(\boldsymbol{\omega}_{t-1}')\|_2^2 + \|F(\boldsymbol{\omega}_t') - F(\boldsymbol{\omega}_t', \xi_t)\|_2^2\big]. \tag{152}$$

Then for $\eta_t \leq \frac{1}{2\sqrt{3}L}$ we have $36\eta_t^2\eta_{t-1}^2L^4 \leq 3\eta_{t-1}^2L^2$,

$$
\begin{aligned}
2\eta_t F(\boldsymbol{\omega}_t')^\top(\boldsymbol{\omega}_t' - \boldsymbol{\omega}) \leq &\|\boldsymbol{\omega}_t - \boldsymbol{\omega}\|_2^2 - \|\boldsymbol{\omega}_{t+1} - \boldsymbol{\omega}\|_2^2 \\
&+ 3L^2(\eta_{t-1}^2\|\boldsymbol{\omega}_{t-1}' - \boldsymbol{\omega}_{t-2}'\|_2^2 - \eta_t^2\|\boldsymbol{\omega}_{t-1}' - \boldsymbol{\omega}_t'\|_2^2) \\
&+ 2\eta_t(F(\boldsymbol{\omega}_t') - F(\boldsymbol{\omega}_t', \xi_t))^\top(\boldsymbol{\omega}_t' - \boldsymbol{\omega}) \\
&+ 3\eta_t^2\left[\|F(\boldsymbol{\omega}_{t-2}', \xi_{t-2}) - F(\boldsymbol{\omega}_{t-2}')\|_2^2 + 2\|F(\boldsymbol{\omega}_{t-1}', \xi_{t-1}) - F(\boldsymbol{\omega}_{t-1}')\|_2^2\right. \\
&\left. + \|F(\boldsymbol{\omega}_t') - F(\boldsymbol{\omega}_t', \xi_t)\|_2^2\right].
\end{aligned}
\tag{153}
$$

We can then use Lemma 7 where

$$
\begin{aligned}
N_t &= \|\boldsymbol{\omega}_t - \boldsymbol{\omega}\|_2^2 + 3L^3\eta_{t-1}\|\boldsymbol{\omega}_{t-1}' - \boldsymbol{\omega}_{t-2}'\|_2^2, \\
M_1(\boldsymbol{\omega}_t, \xi_t) &= 0 \\
M_2(\boldsymbol{\omega}_t', \xi_t) &= 3\|F(\boldsymbol{\omega}_t') - F(\boldsymbol{\omega}_t', \xi_t)\|_2^2 + 6\|F(\boldsymbol{\omega}_{t-1}') - F(\boldsymbol{\omega}_{t-1}', \xi_{t-1})\|_2^2 \\
&\quad + 3\|F(\boldsymbol{\omega}_{t-2}') - F(\boldsymbol{\omega}_{t-2}', \xi_{t-2})\|_2^2 \\
\Delta_t &= F(\boldsymbol{\omega}_t') - F(\boldsymbol{\omega}_t', \xi_t) \\
\boldsymbol{z}_t &= \boldsymbol{\omega}_t'.
\end{aligned}
$$

By Assumption 1, $M_2 = 12\sigma^2$ and by the fact that $\mathbb{E}[F(\boldsymbol{\omega}_t') - F(\boldsymbol{\omega}_t', \xi_t)|\boldsymbol{\omega}_t', \Delta_0, \ldots, \Delta_{t-1}] = \mathbb{E}[\mathbb{E}[F(\boldsymbol{\omega}_t') - F(\boldsymbol{\omega}_t', \xi_t)|\boldsymbol{\omega}_t']|\Delta_0, \ldots, \Delta_{t-1}] = 0$ the hypothesis of Lemma 7 hold and we get,

$$
\mathbb{E}[\text{Err}_R(\bar{\boldsymbol{\omega}}_T)] \leq \frac{R^2}{S_T} + \frac{13\sigma^2}{2S_T}\sum_{t=0}^{T-1}\eta_t^2
\tag{154}
$$

$\square$

## F.4 PROOF OF THEOREM 5

Theorem 5 has been introduced in §D. This theorem is about Algorithm 5 which consists in another way to implement extrapolation to SGD. Let us first restate this theorem,

**Theorem' 5.** *Assume that $\|\boldsymbol{\omega}_t' - \boldsymbol{\omega}_0\| \leq R$, $\forall t \geq 0$ where $(\boldsymbol{\omega}_t')_{t\geq0}$ are the iterates of Alg. 5. Under Assumption 1 and 4, for any $T \geq 1$, Alg. 5 with constant step-size $\eta \leq \frac{1}{\sqrt{2}L}$ has the following convergence properties:*

$$
\mathbb{E}[\text{Err}_R(\bar{\boldsymbol{\omega}}_T)] \leq \frac{R^2}{\eta T} + \eta\frac{\sigma^2 + 4L^2(4R^2 + \sigma^2)}{2} \quad where \quad \bar{\boldsymbol{\omega}}_T \overset{def}{=} \frac{1}{T}\sum_{t=0}^{T-1}\boldsymbol{\omega}_t'.
$$

*Particularly, $\eta_t = \frac{\eta}{\sqrt{T}}$ gives $\mathbb{E}[\text{Err}_R(\bar{\boldsymbol{\omega}}_T)] \leq \frac{O(1)}{\sqrt{T}}$.*

***Proof of Thm. 5.*** Let any $\boldsymbol{\omega} \in \Omega$ such that $\|\boldsymbol{\omega}_0 - \boldsymbol{\omega}\|_2 \leq R$. Then, the update rules become $\boldsymbol{\omega}_{t+1} = P_\Omega(\boldsymbol{\omega}_t - \eta_t F(\boldsymbol{\omega}_t', \xi_t))$ and $\boldsymbol{\omega}_t' = P_\Omega(\boldsymbol{\omega}_t - \eta F(\boldsymbol{\omega}_t, \xi_t))$. We start the same way as the proof of Thm. 3 by applying Lemma 2 for $(\boldsymbol{\omega}, \boldsymbol{u}, \boldsymbol{\omega}', \boldsymbol{\omega}^*) = (\boldsymbol{\omega}_t, -\eta F(\boldsymbol{\omega}_t', \xi_t), \boldsymbol{\omega}, \boldsymbol{\omega}_{t+1})$ and $(\boldsymbol{\omega}, \boldsymbol{u}, \boldsymbol{\omega}', \boldsymbol{\omega}^+) = (\boldsymbol{\omega}_t, -\eta_t F(\boldsymbol{\omega}_t, \xi_t), \boldsymbol{\omega}_{t+1}, \boldsymbol{\omega}_t')$,

$$
\begin{aligned}
\|\boldsymbol{\omega}_{t+1} - \boldsymbol{\omega}\|_2^2 &\leq \|\boldsymbol{\omega}_t - \boldsymbol{\omega}\|_2^2 - 2\eta_t F(\boldsymbol{\omega}_t', \xi_t)^\top(\boldsymbol{\omega}_{t+1} - \boldsymbol{\omega}) - \|\boldsymbol{\omega}_{t+1} - \boldsymbol{\omega}_t\|_2^2 \\
\|\boldsymbol{\omega}_t' - \boldsymbol{\omega}_{t+1}\|_2^2 &\leq \|\boldsymbol{\omega}_t - \boldsymbol{\omega}_{t+1}\|_2^2 - 2\eta_t F(\boldsymbol{\omega}_t, \xi_t)^\top(\boldsymbol{\omega}_t' - \boldsymbol{\omega}_{t+1}) - \|\boldsymbol{\omega}_t' - \boldsymbol{\omega}_t\|_2^2
\end{aligned}
$$

Then, summing them we get

$$
\begin{aligned}
\|\boldsymbol{\omega}_{t+1} - \boldsymbol{\omega}\|_2^2 \leq &\|\boldsymbol{\omega}_t - \boldsymbol{\omega}\|_2^2 - 2\eta_t F(\boldsymbol{\omega}_t', \xi_t)^\top(\boldsymbol{\omega}_{t+1} - \boldsymbol{\omega}) \\
&- 2\eta_t F(\boldsymbol{\omega}_t, \xi_t)^\top(\boldsymbol{\omega}_t' - \boldsymbol{\omega}_{t+1}) - \|\boldsymbol{\omega}_t' - \boldsymbol{\omega}_t\|_2^2 - \|\boldsymbol{\omega}_{t+1} - \boldsymbol{\omega}_t'\|_2^2 \quad (155)
\end{aligned}
$$

leading to

$$
\begin{aligned}
\|\boldsymbol{\omega}_{t+1} - \boldsymbol{\omega}\|_2^2 \leq &\|\boldsymbol{\omega}_t - \boldsymbol{\omega}\|_2^2 - 2\eta_t F(\boldsymbol{\omega}_t', \xi_t)^\top(\boldsymbol{\omega}_t' - \boldsymbol{\omega}) \\
&+ 2\eta_t(F(\boldsymbol{\omega}_t', \xi_t) - F(\boldsymbol{\omega}_t, \xi_t))^\top(\boldsymbol{\omega}_t' - \boldsymbol{\omega}_{t+1}) - \|\boldsymbol{\omega}_t' - \boldsymbol{\omega}_t\|_2^2 - \|\boldsymbol{\omega}_{t+1} - \boldsymbol{\omega}_t'\|_2^2
\end{aligned}
$$

Then with $2\boldsymbol{a}^\top \boldsymbol{b} \le \|\boldsymbol{a}\|_2^2 + \|\boldsymbol{b}\|_2^2$ we get

$$\|\boldsymbol{\omega}_{t+1} - \boldsymbol{\omega}\|_2^2 \le \|\boldsymbol{\omega}_t - \boldsymbol{\omega}\|_2^2 - 2\eta_t F(\boldsymbol{\omega}_t', \xi_t)^\top (\boldsymbol{\omega}_t' - \boldsymbol{\omega})$$
$$+\eta_t^2 \|F(\boldsymbol{\omega}_t', \xi_t) - F(\boldsymbol{\omega}_t, \xi_t)\|_2^2 - \|\boldsymbol{\omega}_t' - \boldsymbol{\omega}_t\|_2^2$$

Using the Lipschitz assumption we get

$$\|\boldsymbol{\omega}_{t+1} - \boldsymbol{\omega}\|_2^2 \le \|\boldsymbol{\omega}_t - \boldsymbol{\omega}\|_2^2 - 2\eta_t F(\boldsymbol{\omega}_t', \xi_t)^\top (\boldsymbol{\omega}_t' - \boldsymbol{\omega}) + (\eta_t^2 L^2 - 1)\|\boldsymbol{\omega}_t - \boldsymbol{\omega}_t'\|_2^2$$

Then we add $2\eta_t F(\boldsymbol{\omega}_t')^\top (\boldsymbol{\omega}_t' - \boldsymbol{\omega})$ in both sides to get,

$$2\eta_t F(\boldsymbol{\omega}_t')^\top (\boldsymbol{\omega}_t' - \boldsymbol{\omega}) \le \|\boldsymbol{\omega}_t - \boldsymbol{\omega}\|_2^2 - \|\boldsymbol{\omega}_{t+1} - \boldsymbol{\omega}\|_2^2$$
$$- 2\eta_t (F(\boldsymbol{\omega}_t', \xi_t) - F(\boldsymbol{\omega}_t'))^\top (\boldsymbol{\omega}_t' - \boldsymbol{\omega}) + (\eta_t^2 L^2 - 1)\|\boldsymbol{\omega}_t - \boldsymbol{\omega}_t'\|_2^2 \quad (156)$$

Here, unfortunately we cannot use Lemma 7 because $F(\boldsymbol{\omega}_t', \xi_t)$ is biased. We will then deal with the quantity $A = (F(\boldsymbol{\omega}_t', \xi_t) - F(\boldsymbol{\omega}'))^\top (\boldsymbol{\omega}_t' - \boldsymbol{\omega})$ . We have that,

$$A = (F(\boldsymbol{\omega}_t', \xi_t) - F(\boldsymbol{\omega}_t, \xi_t))^\top (\boldsymbol{\omega} - \boldsymbol{\omega}_t') + (F(\boldsymbol{\omega}_t) - F(\boldsymbol{\omega}_t'))^\top (\boldsymbol{\omega} - \boldsymbol{\omega}_t')$$
$$+ (F(\boldsymbol{\omega}_t, \xi_t) - F(\boldsymbol{\omega}_t))^\top (\boldsymbol{\omega}_t - \boldsymbol{\omega}_t') + (F(\boldsymbol{\omega}_t, \xi_t) - F(\boldsymbol{\omega}_t))^\top (\boldsymbol{\omega} - \boldsymbol{\omega}_t)$$
$$\le 2L\|\boldsymbol{\omega}_t' - \boldsymbol{\omega}_t\|_2 \|\boldsymbol{\omega}_t' - \boldsymbol{\omega}\|_2 + \|F(\boldsymbol{\omega}_t, \xi_t) - F(\boldsymbol{\omega}_t)\|\|\boldsymbol{\omega}_t' - \boldsymbol{\omega}_t\|_2$$
$$+ (F(\boldsymbol{\omega}_t, \xi_t) - F(\boldsymbol{\omega}_t))^\top (\boldsymbol{\omega} - \boldsymbol{\omega}_t)$$
$$\text{(Using Cauchy-Schwarz and the } L\text{-Lip of } F)$$

Then using $2\|a\|\|b\| \le \delta\|a\|_2^2 + \frac{1}{\delta}\|b\|_2^2$, for $\delta = 4$,

$$-2\eta_t (F(\boldsymbol{\omega}_t', \xi_t) - F(\boldsymbol{\omega}_t'))^\top (\boldsymbol{\omega}_t' - \boldsymbol{\omega}) \le \frac{1}{2}\|\boldsymbol{\omega}_t' - \boldsymbol{\omega}_t\|^2 + 8\eta_t^2 L^2 \|\boldsymbol{\omega}_t' - \boldsymbol{\omega}\|_2^2$$
$$+ 4\eta_t^2 \|F(\boldsymbol{\omega}_t, \xi_t) - F(\boldsymbol{\omega}_t)\|_2^2$$
$$+ \frac{1}{4}\|\boldsymbol{\omega}_t' - \boldsymbol{\omega}_t\|_2^2 + 2\eta_t (F(\boldsymbol{\omega}_t, \xi_t) - F(\boldsymbol{\omega}_t))^\top (\boldsymbol{\omega} - \boldsymbol{\omega}_t)$$

leading to,

$$2\eta_t F(\boldsymbol{\omega}_t')^\top (\boldsymbol{\omega}_t' - \boldsymbol{\omega}) \le \|\boldsymbol{\omega}_t - \boldsymbol{\omega}\|_2^2 - \|\boldsymbol{\omega}_{t+1} - \boldsymbol{\omega}\|_2^2 + 2\eta_t (F(\boldsymbol{\omega}_t, \xi_t) - F(\boldsymbol{\omega}_t))^\top (\boldsymbol{\omega} - \boldsymbol{\omega}_t)$$
$$+ (\eta_t^2 L^2 - \frac{1}{4})\|\boldsymbol{\omega}_t - \boldsymbol{\omega}_t'\|_2^2$$
$$+ 4\eta_t^2 (2L^2 \|\boldsymbol{\omega}_t' - \boldsymbol{\omega}\|_2^2 + \|F(\boldsymbol{\omega}_t, \xi_t) - F(\boldsymbol{\omega}_t)\|_2^2)$$

If one assumes finally that $\|\boldsymbol{\omega}_t' - \boldsymbol{\omega}_0\|_2 \le R$ (assumption of the theorem) and that $\eta_t \le \frac{1}{2L}$ we get,

$$2\eta_t F(\boldsymbol{\omega}_t')^\top (\boldsymbol{\omega}_t' - \boldsymbol{\omega}) \le \|\boldsymbol{\omega}_t - \boldsymbol{\omega}\|_2^2 - \|\boldsymbol{\omega}_{t+1} - \boldsymbol{\omega}\|_2^2 + 2\eta_t (F(\boldsymbol{\omega}_t, \xi_t) - F(\boldsymbol{\omega}_t))^\top (\boldsymbol{\omega} - \boldsymbol{\omega}_t)$$
$$+ 4\eta_t^2 (4L^2 R^2 + \|F(\boldsymbol{\omega}_t, \xi_t) - F(\boldsymbol{\omega}_t)\|_2^2)$$

where we used that $\|\boldsymbol{\omega}_t' - \boldsymbol{\omega}\|_2 \le \|\boldsymbol{\omega}_t' - \boldsymbol{\omega}_0\|_2 + \|\boldsymbol{\omega}_0 - \boldsymbol{\omega}\|_2 \le 2R$. Once again this equation is a particular case of Lemma 7 where $N_t = \|\boldsymbol{\omega}_t - \boldsymbol{\omega}\|_2^2$, $M_1(\boldsymbol{\omega}_t, \xi_t) = 4(4L^2 R^2 + \|F(\boldsymbol{\omega}_t, \xi_t) - F(\boldsymbol{\omega}_t)\|_2^2)$, $M_2(\boldsymbol{\omega}_t', \zeta_t) = 0$, $\boldsymbol{z}_t = \boldsymbol{\omega}_t$ and $\Delta_t = F(\boldsymbol{\omega}_t, \xi_t) - F(\boldsymbol{\omega}_t)$. By Assumption 1 $\mathbb{E}[M_1(\boldsymbol{\omega}_t, \xi_t)] \le 16L^2 R^2 + 4\sigma^2$ and $\mathbb{E}[\Delta_t|\boldsymbol{\omega}_t, \Delta_0, \dots, \Delta_{t-1}] = \mathbb{E}[\mathbb{E}[\Delta_t|\boldsymbol{\omega}_t]|\Delta_0, \dots, \Delta_{t-1}] = 0$ so we can use Lemma 7 and get,

$$\mathbb{E}[\text{Err}_R(\bar{\boldsymbol{\omega}}_T)] \le \frac{R^2}{S_T} + \frac{\sigma^2 + 16L^2 R^2 + 4\sigma^2}{2S_T} \sum_{t=0}^{T-1} \eta_t^2 . \quad (157)$$

$\square$

## G  ADDITIONAL EXPERIMENTAL RESULTS

### G.1  TOY NON-CONVEX GAN (2D AND DETERMINISTIC)

We now consider a task similar to (Mescheder et al., 2018) where the discriminator is linear $D_{\boldsymbol{\varphi}}(\boldsymbol{\omega}) = \boldsymbol{\varphi}^T \boldsymbol{\omega}$, the generator is a Dirac distribution at $\boldsymbol{\theta}$, $q_{\boldsymbol{\theta}} = \delta_{\boldsymbol{\theta}}$ and the distribution we try to match is also a Dirac at $\boldsymbol{\omega}^*$, $p = \delta_{\boldsymbol{\omega}^*}$. The minimax formulation from Goodfellow et al. (2014) gives:

$$\min_{\boldsymbol{\theta}} \max_{\boldsymbol{\varphi}} - \log\left(1 + e^{-\boldsymbol{\varphi}^T \boldsymbol{\omega}^*}\right) - \log\left(1 + e^{\boldsymbol{\varphi}^T \boldsymbol{\theta}}\right) \tag{158}$$

Note that as observed by Nagarajan and Kolter (2017), this objective is concave-concave, making it hard to optimize. We compare the methods on this objective where we take $\boldsymbol{\omega}^* = -2$, thus the position of the equilibrium is shifted towards the position $(\boldsymbol{\theta}, \boldsymbol{\varphi}) = (-2, 0)$. The convergence and the gradient vector field are shown in Figure 5. We observe that depending on the initialization, some methods can fail to converge but *extrapolation* (18) seems to perform better than the other methods.

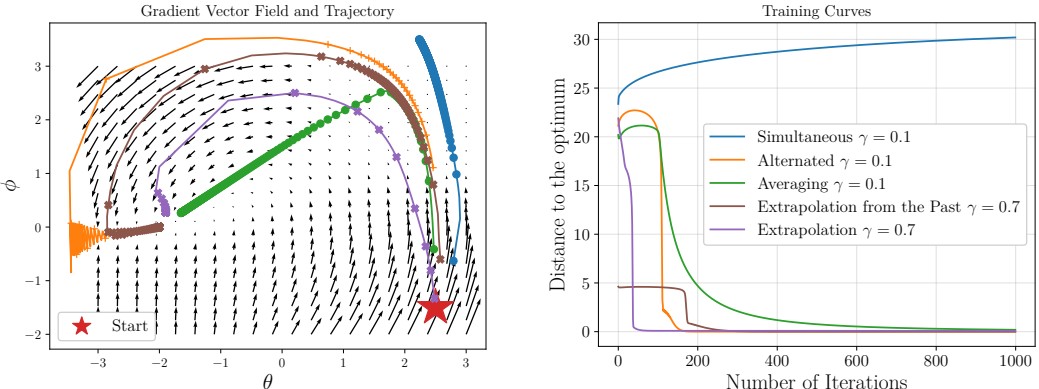

Figure 5: Comparison of five algorithms (described in Section 3) on the non-convex GAN objective (158), using the optimal step-size for each method. **Left**: The gradient vector field and the dynamics of the different methods. **Right**:The distance to the optimum as a function of the number of iterations.

### G.2  DCGAN WITH WGAN-GP OBJECTIVE

In addition to the results presented in section §7.2, we also trained the DCGAN architecture with the WGAN-GP objective. The results are shown in Table 3. The best results are achieved with *uniform averaging* of AltAdam5. However, its iterations require to update the discriminator 5 times for every generator update. With a small drop in best final score, ExtraAdam can train WGAN-GP significantly faster (see Fig. 6 right) as the discriminator and generator are updated only twice.

### G.3  FID SCORES FOR RESNET ARCHITECTURE WITH WGAN-GP OBJECTIVE

In addition to the inception scores, we also computed the FID scores (Heusel et al., 2017) using 50,000 samples for the ResNet architecture with the WGAN-GP objective; the results are presented in Table 5. We see that the results and conclusions are similar to the one obtained from the inception scores, adding an extrapolation step as well as using Exponential Moving Average (EMA) consistently improves the FID scores. However, contrary to the results from the inception score, we observe that uniform averaging does not necessarily improve the performance of the methods. This could be due to the fact that the samples produced using uniform averaging are more blurry and FID is more sensitive to blurriness; see §G.3 for more details about the effects of uniform averaging.

| Generator |
|---|
| *Input: $z \in \mathbb{R}^{128} \sim \mathcal{N}(0, I)$* |
| Linear $128 \to 512 \times 4 \times 4$ |
| Batch Normalization |
| ReLU |
| transposed conv. (kernel: $4\times4$, $512 \to 256$, stride: 2, pad: 1) |
| Batch Normalization |
| ReLU |
| transposed conv. (kernel: $4\times4$, $256 \to 128$, stride: 2, pad: 1) |
| Batch Normalization |
| ReLU |
| transposed conv. (kernel: $4\times4$, $128 \to 3$, stride: 2, pad: 1) |
| $Tanh(\cdot)$ |
| **Discriminator** |
| *Input: $x \in \mathbb{R}^{3\times32\times32}$* |
| conv. (kernel: $4\times4$, $1 \to 64$; stride: 2; pad:1) |
| LeakyReLU (negative slope: 0.2) |
| conv. (kernel: $4\times4$, $64 \to 128$; stride: 2; pad:1) |
| Batch Normalization |
| LeakyReLU (negative slope: 0.2) |
| conv. (kernel: $4\times4$, $128 \to 256$; stride: 2; pad:1) |
| Batch Normalization |
| LeakyReLU (negative slope: 0.2) |
| Linear $128 \times 4 \times 4 \times 4 \to 1$ |

Table 2: DCGAN architecture used for our CIFAR-10 experiments. When using the gradient penalty (WGAN-GP), we remove the Batch Normalization layers in the discriminator.

| Model | WGAN-GP (DCGAN) | |
|---|---|---|
| Method | no averaging | uniform avg |
| SimAdam | *$6.00 \pm .07$* | $6.01 \pm .08$ |
| AltAdam5 | *$6.25 \pm .05$* | **$6.51 \pm .05$** |
| ExtraAdam | $6.22 \pm .04$ | $6.35 \pm .05$ |
| PastExtraAdam | $6.27 \pm 0.06$ | $6.23 \pm 0.13$ |

Table 3: Best inception scores (averaged over 5 runs) achieved on CIFAR10 for every considered Adam variant. We see that the techniques of extrapolation and averaging consistently enable improvements over the baselines (in italic).

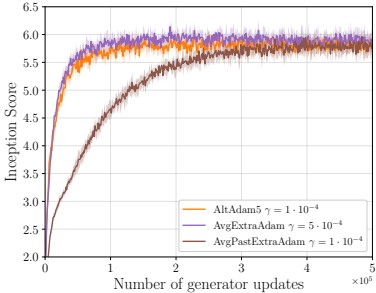 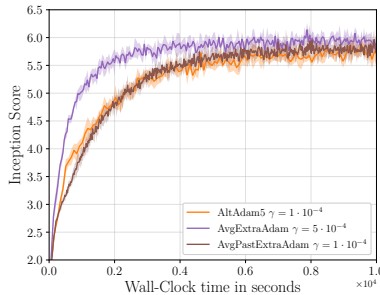

Figure 6: DCGAN architecture with WGAN-GP trained on CIFAR10: mean and standard deviation of the inception score computed over 5 runs for each method using the best performing learning rate plotted over number of generator updates (**Left**) and wall-clock time (**Right**); all experiments were run on a NVIDIA Quadro GP100 GPU. We see that ExtraAdam converges faster than the Adam baselines.

| **Generator** |
| :---: |
| *Input:* $z \in \mathbb{R}^{128} \sim \mathcal{N}(0, I)$ |
| Linear $128 \to 128 \times 4 \times 4$ |
| ResBlock $128 \to 128$ |
| ResBlock $128 \to 128$ |
| ResBlock $128 \to 128$ |
| Batch Normalization |
| ReLU |
| transposed conv. (kernel: $3 \times 3$, $128 \to 3$, stride: 1, pad: 1) |
| $Tanh(\cdot)$ |
| **Discriminator** |
| *Input:* $x \in \mathbb{R}^{3 \times 32 \times 32}$ |
| ResBlock $3 \to 128$ |
| ResBlock $128 \to 128$ |
| ResBlock $128 \to 128$ |
| ResBlock $128 \to 128$ |
| Linear $128 \to 1$ |

Table 4: ResNet architecture used for our CIFAR-10 experiments. When using the gradient penalty (WGAN-GP), we remove the Batch Normalization layers in the discriminator.

| Model | WGAN-GP (ResNet) | | |
| :--- | :---: | :---: | :---: |
| Method | no averaging | uniform avg | EMA |
| SimAdam | *23.74 ± 2.79* | 26.29 ± 5.56 | 21.89 ± 2.51 |
| AltAdam5 | *21.65 ± 0.66* | 19.91 ± 0.43 | 20.69 ± 0.37 |
| ExtraAdam | 19.42 ± 0.15 | 18.13 ± 0.51 | **16.78 ± 0.21** |
| PastExtraAdam | 19.95 ± 0.38 | 22.45 ± 0.93 | 17.85 ± 0.40 |
| OptimAdam | *18.88 ± 0.55* | 21.23 ± 1.19 | 16.91 ± 0.32 |

Table 5: Best FID scores (averaged over 5 runs) achieved on CIFAR10 for every considered Adam variant. OptimAdam is the related *Optimistic Adam* (Daskalakis et al., 2018) algorithm. We see that the techniques of extrapolation and EMA consistently enable improvements over the baselines (in italic).

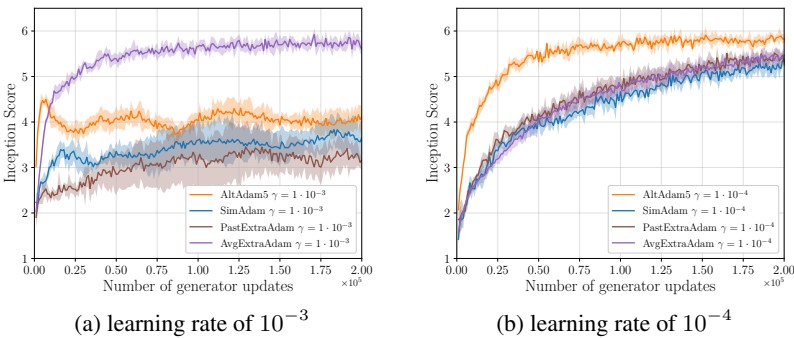

(a) learning rate of $10^{-3}$          (b) learning rate of $10^{-4}$

Figure 7: Inception score on CIFAR10 for WGAN-GP (DCGAN) over number of generator updates for different learning rates. We can see that AvgExtraAdam is less sensitive to the choice of learning rate.

### G.4 COMPARISON OF THE METHODS WITH THE SAME LEARNING RATE

In this section, we compare how the methods presented in §7 perform with the same step-size. We follow the same protocol as in the experimental section §7, we consider the DCGAN architecture with WGAN-GP experiment described in App §G.2. In Figure 7 we plot the inception score provided by each training method as a function of the number of generator updates. Note that these plots advantage **AltAdam5** a bit because each iteration of this algorithm is a bit more costly (since it perform 5 discriminator updates for each generator update). Nevertheless, the goal of this experiment is not to show that **AltAdam5** is faster but to show that **ExtraAdam** is less sensitive to the choice of learning rate and can be used with higher learning rates with less degradation.

In Figure 8, we compare the sample quality on the DCGAN architecture with the WGAN-GP objective of **AltAdam5** and **AvgExtraAdam** for different step-sizes. We notice that for **AvgExtraAdam**, the sample quality does not significantly change whereas the sample quality of **AltAdam5** seems to be really sensitive to step-size tunning.

We think that robustness to step-size tuning is a key property for an optimization algorithm in order to save as much time as possible to tune other hyperparameters of the learning procedure such as regularization.

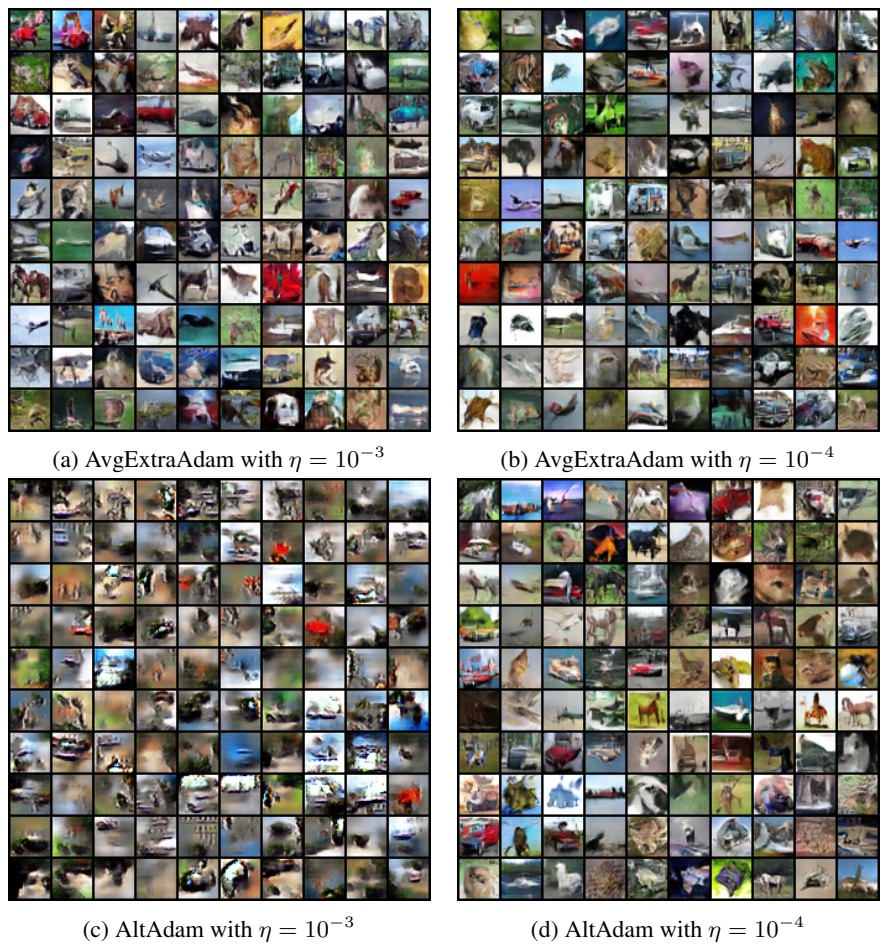

(a) AvgExtraAdam with $\eta = 10^{-3}$      (b) AvgExtraAdam with $\eta = 10^{-4}$

(c) AltAdam with $\eta = 10^{-3}$      (d) AltAdam with $\eta = 10^{-4}$

Figure 8: Comparison of the samples quality on the WGAN-GP (DCGAN) experiment for different methods and learning rate $\eta$.

### G.5 COMPARISON OF THE METHODS WITH AND WITHOUT UNIFORM AVERAGING

In this section, we compare how uniform averaging affect the performance of the methods presented in §7. We follow the same protocol as in the experimental section §7, we consider the DCGAN architecture with the WGAN and weight clipping objective as well as the WGAN-GP objective. In Figure 9 and 10, we plot the inception score provided by each training method as a function of the number of generator updates with and without uniform averaging.

We notice that uniform averaging seems to improve the inception score, nevertheless it looks like the sample are a bit more blurry (see Figure 11). This is confirmed by our result (Figure 12) on the Fréchet Inception Distance (FID) which is more sensitive to blurriness. A similar observation about FID was made in §G.3.

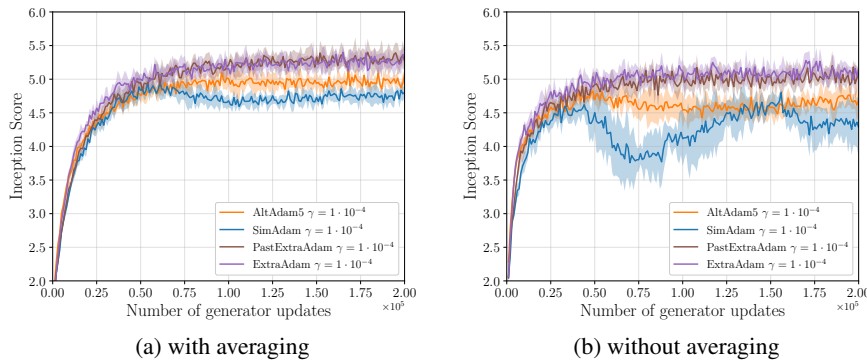

(a) with averaging  (b) without averaging

Figure 9: Inception Score on CIFAR10 for WGAN over number of generator updates with and without averaging. We can see that averaging improve the inception score.

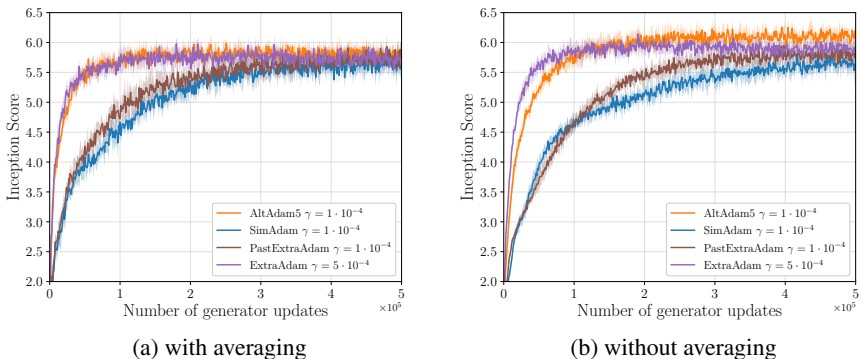

(a) with averaging  (b) without averaging

Figure 10: Inception score on CIFAR10 for WGAN-GP (DCGAN) over number of generator updates

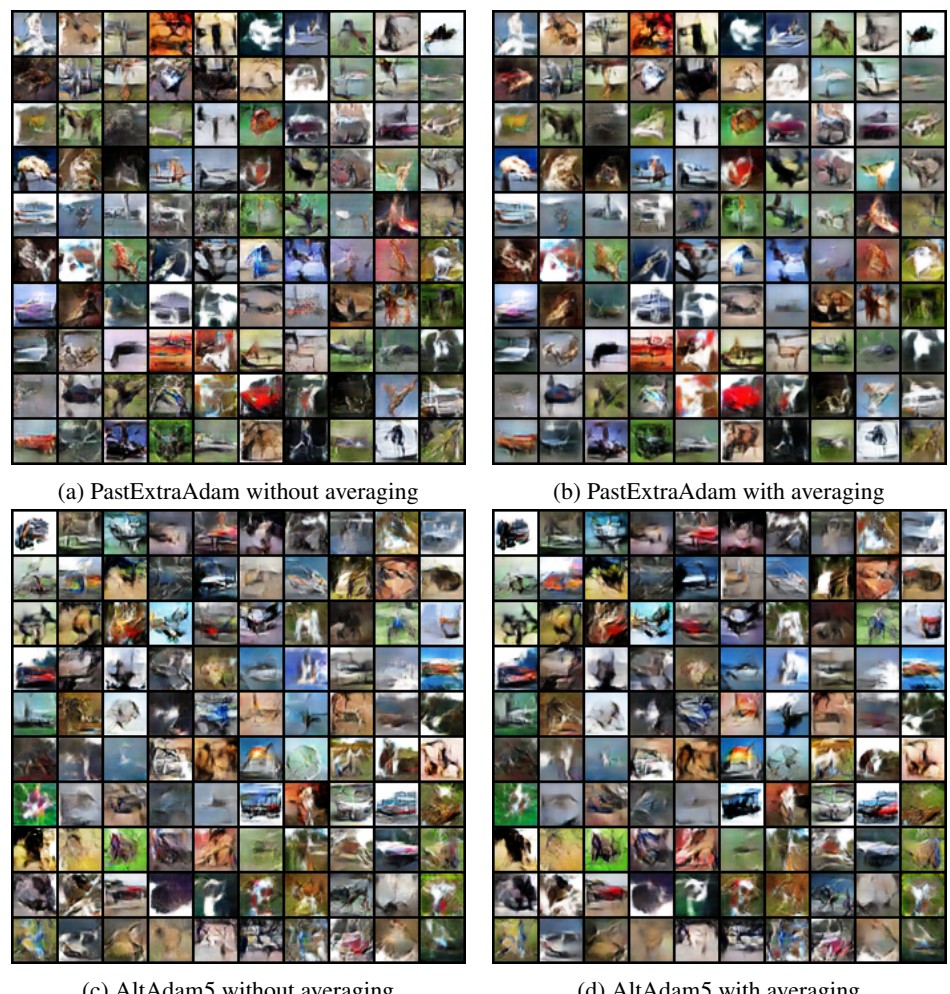

(a) PastExtraAdam without averaging  (b) PastExtraAdam with averaging

(c) AltAdam5 without averaging  (d) AltAdam5 with averaging

Figure 11: Comparison of the samples of a WGAN trained with the different methods with and without averaging. Although averaging improves the inception score, the samples seem more blurry

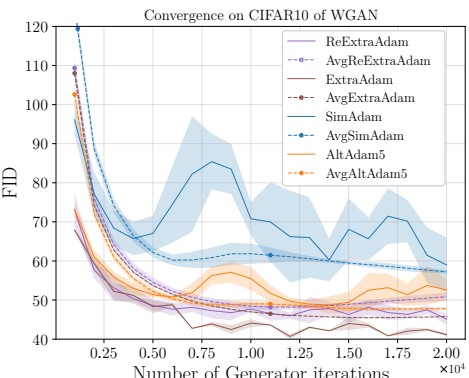

Figure 12: The Fréchet Inception Distance (FID) from Heusel et al. (2017) computed using 50,000 samples, on the WGAN experiments. ReExtraAdam refers to Alg. 5 introduced in §D. We can see that averaging performs worse than when comparing with the Inception Score. We observed that the samples generated by using averaging are a little more blurry and that the FID is more sensitive to blurriness, thus providing an explanation for this observation.

## H  HYPERPARAMETERS

| **(DCGAN) WGAN Hyperparameters** | |
|---|---|
| Batch size | $= 64$ |
| Number of generator update | $= 500,000$ |
| Adam $\beta_1$ | $= 0.5$ |
| Adam $\beta_2$ | $= 0.9$ |
| Weight clipping for the discriminator | $= 0.01$ |
| Learning rate for generator | $= 2 \times 10^{-5}$ (for Adam1, Adam5, PastExtraAdam, OptimisticAdam) |
| | $= 5 \times 10^{-5}$ (for ExtraAdam) |
| Learning rate for discriminator | $= 2 \times 10^{-4}$ (for Adam1, Adam5, PastExtraAdam, OptimisticAdam) |
| | $= 5 \times 10^{-4}$ (for ExtraAdam) |
| $\beta$ for EMA | $= 0.999$ |

| **(DCGAN) WGAN-GP Hyperparameters** | |
|---|---|
| Batch size | $= 64$ |
| Number of generator update | $= 500,000$ |
| Adam $\beta_1$ | $= 0.5$ |
| Adam $\beta_2$ | $= 0.9$ |
| Gradient penalty | $= 10$ |
| Learning rate for generator | $= 1 \times 10^{-4}$ (for Adam1, Adam5, PastExtraAdam, OptimisticAdam) |
| | $= 5 \times 10^{-4}$ (for ExtraAdam) |
| Learning rate for discriminator | $= 1 \times 10^{-4}$ (for Adam1, Adam5, PastExtraAdam, OptimisticAdam) |
| | $= 5 \times 10^{-4}$ (for ExtraAdam) |
| $\beta$ for EMA | $= 0.999$ |

| **(ResNet) WGAN-GP Hyperparameters** | |
|---|---|
| Batch size | $= 64$ |
| Number of generator update | $= 500,000$ |
| Adam $\beta_1$ | $= 0.5$ |
| Adam $\beta_2$ | $= 0.9$ |
| Gradient penalty | $= 10$ |
| Learning rate for generator | $= 2 \times 10^{-5}$ (for Adam1, Adam5, PastExtraAdam, OptimisticAdam) |
| | $= 5 \times 10^{-5}$ (for ExtraAdam) |
| Learning rate for discriminator | $= 2 \times 10^{-4}$ (for Adam1, Adam5, PastExtraAdam, OptimisticAdam) |
| | $= 5 \times 10^{-4}$ (for ExtraAdam) |
| $\beta$ for EMA | $= 0.9999$ |

