# OpenReview forum: "A Variational Inequality Perspective on Generative Adversarial Networks"
_ICLR.cc/2019/Conference_

### Official Review · AnonReviewer2 · 2018-11-01
**Good review on algorithms for VIs in the context of GANs**

**Rating:** 7
**Confidence:** 4

**Review:**

Overall, the paper is well-written and of high quality, therefore I recommend acceptance.

Pros:
+ The work gives an accessible but still rigorous introduction to the literature on VIs which I find highly valuable, as it creates a bridge between the classical mathematical programming literature and applications in AI.

+ The theory for optimization of VIs with stochastic gradients (though only in monotone setting) was very interesting to me and contains some novel results (Theorem 2, Theorem 4)

Cons:
- I'm a bit skeptical about the experiments on GANs. They indicate that for the specific choice of architectures and hyper-parameters "ExtraAdam" works better, but the chosen architectures are not state-of-the art. What would convince me if the algorithm can be used to improve a current best inception score of 8.2 reached with SNGANs. Also with WGAN-GP, scores of ~7.8 are reported which are much higher than the 6.4 reported in the paper. But I understand that producing state-of-the-art inception scores is not the focus of the paper, therefore I would suggest that the authors release an implementation of the proposed new optimizers (ExtraAdam) for a popular DL framework (e.g. pytorch) such that practitioners working with GANs can quickly try them out in a "plug-and-play" fashion.

- Proposition 2 is a bit misleading. While for \eta \in (0, 1) implicit and extrapolation are similar, adding the remark that implicit method is stable for any \eta > 0 (and therefore can lead to an arbitrary fast convergence) would give a more balanced view. Right now, only the advantages of extrapolation method and disadvantages of implicit method are mentioned which I find unfair for the implicit method.

- The theory is presented for variational inequalities with monotone operators. For clarity it should be mentioned that GANs parametrized with neural nets lead to non-monotone VIs. A provably convergent algorithm for that setting is still an open problem, no?

---

> ### Author Response · Authors · 2018-11-17
> **Many thanks for the constructive comments. **New experimental results****
>
> The authors first would like to thank AnonReviewer2 for his thorough evaluation and interesting remarks. In the following, we try to address as clearly as possible the points risen by AnonReviewer2.
>
> “I'm a bit skeptical about the experiments on GANs.:”
> As suggested, we tried to train a WGAN-GP with a ResNet architecture. The new results have been included in the updated version of the paper (see the experimental section 7.2, table 1 and figure 4 for the new results). After few days of experiments, we were able to obtain state-of-the art results on this architecture using ExtraAdam with averaging. As developed in our new experimental section (see the revised paper), we are not claiming that our principled methods are the solution to the many challenges of practical GAN optimization ( it is possible that a very fine-grained hyperparameter tuning on a standard method may give similar results) but after spending a similar limited time budget on optimizing the hyperparameters of each algorithm it looks clear to us that for this task the ExtraAdam method is much more robust to hyperparameter tuning, i.e, it yields reasonably good results for a large range of step-sizes.
> The code in pytorch containing all the algorithms presented in the paper as well as the exact experimental setup is ready and will be released after the anonymity period due to the reviewing process.
>
> “Proposition 2 is a bit misleading.”
> Our goal was not to weaken the value of implicit methods. When a closed form for the implicit updates is known, this method is very effective, but unfortunately for neural network optimization we are not aware of any practical way to implement the implicit steps. More precisely, an implicit step is equivalent to computing a minimization step of the original objective with a l2 regularization (see [1] for more details on implicit SGD and its applications), this subproblem is supposed to be simpler because of the strong convexity of the l2 regularization. Unfortunately, for neural networks, the optimization problem remains non-convex for any small step-size (which are the step-sizes of interest). Hence, we considered that implicit steps were prohibitively expensive for our applications of interest.
>
> “The theory is presented for variational inequalities with monotone operators.”
> As we mentioned it in the second paragraph of Sec. 2.2 “Standard GAN objectives are non-convex (i.e. each cost function is non-convex),”, meaning that they are non-monotone since as we explain it right after the definition of monotonicity, “If F can be written as (6), it implies that the cost functions are convex.“. We added a clarification in the paper right after the definition of monotonicity (page 7) stating that “GANs parametrized with neural networks lead to non-monotone VIPs” to clarify this. For further discussion about the extension of the VI to non-monotone operators we refer the reviewer to App. C.3.
>
> “A provably convergent algorithm for that setting is still an open problem, no?*”
> To our knowledge, general convergence results in the context of general non-monotone VIPs is still an open question. The only partial results we are aware of are mentioned in our related work section:
> -“for a new notion of regret minimization, by Hazan et al. (2017) and in the context of GANs by Grnarova et al. (2018)“
> -“ Mertikopoulos et al. (2018) also independently explored extrapolation providing asymptotic convergence results (i.e. without any rate of convergence) in the context of coherent saddle point. The coherence assumption is slightly weaker than monotonicity”.
>
>
>
> [1] TOULIS, Panagiotis, AIROLDI, Edoardo, et RENNIE, Jason. Statistical analysis of stochastic gradient methods for generalized linear models. In : International Conference on Machine Learning. 2014.

---

### Official Review · AnonReviewer1 · 2018-11-02
**A new perspective on optimization problems arising in GANs which helps provide insights into why averaging helps, why certain type of updates are bad, and how extrapolation can be used to obtain even better solvers.**

**Rating:** 8
**Confidence:** 3

**Review:**

This paper looks at solving optimization problems that arise in GANs, via a variational inequality perspective (VIP). VIP entails solving an optimization problem that is related to the first order condition of the optimization problem that we wish to solve. VIP have been very successful in solving min-max style problems. Given that, GAN formulations tend to be min-max style problems (though not necessarily 0 sum) the VIP perspective is very natural, though under-explored in machine learning. Two techniques that have been widely used to solve VIP problems are averaging and extragradient methods. The authors look at a simple GAN setup where both the generator and the discriminator are linear models. In this case two kinds of gradient updates can be derived. First are simultaneous updates, and the other is alternated updates. The authors show that simultaneous updates are not even bounded and diverge to infinity, whereas alternated updates are more stable and stay bounded, but need not necessarily converge. However, I think this behaviour is limited to only linear discriminator/generator and might not extend beyond the linear case. The second key idea is the use of extra-gradient updates. Extra-gradient updates perform an "extra" or fake gradient step to get to a new point, and then kind of retracks back and perform a gradient step using the gradient step obtained from the "extra step".  This extra-gradient method is a close approximation to Euler's method, though far more computationally efficient.  However, the extragradient step requires one to calculate gradient twice, which can be expensive in large models. For this reason, the authors suggest using gradients from past as the "extragradient" in the extragradient method.

For strongly-monotone operators (a generalization of strongly-convex functions) extrapolation updates are shown to have linear convergence.  Furthermore, the authors show that using extrapolation and averaging under the assumption that the operator is monotonic, and using constant step size SGD the rates of convergence are better than the rates obtained using plain SGD with averaging but without extrapolation. Authors also show how one can use these ideas using other first order methods such as ADAM instead of SGD. Experiments are shown on the DCGAN architecture.

On the whole this is a really nice paper, that shows how standard ideas from VIP can be useful for training GANs. I recommend acceptance

---

> ### Author Response · Authors · 2018-11-17
> **Thank you for the knowledgeable review**
>
> The authors first would like to thank AnonReviewer1 for his meticulous analysis and his insightful comments.
>
> “I think this behaviour is limited to only linear discriminator/generator and might not extend beyond the linear case.”
> This behavior is a local behavior (and then can be true for a globally non-monotone operator), i.e, if the objective is bilinear in a neighborhood of an (local) equilibrium this behavior is true in that neighborhood. It means that the iterates of the simultaneous method will be expelled from this neighborhood geometrically, the ones of the alternated method will stay in the neighborhood but will not converge to the equilibrium. On the contrary the averaged iterates  and the iterate of the extragradient method will converge to the equilibrium. We also think that these results could be generalized to any game which is locally Hamiltonian (see [1]) around the (local) equilibrium.
>
> “Experiments are shown on the DCGAN architecture. “
> As suggested by AnonReviewer2, we tried our methods to train a ResNet architecture with the WGAN-GP objective (see the experimental section 7.2, table 1 and figure 4 for the new results). After few days of experiments, we were able to match current the state-of-the art results of ~8.2 on this architecture by using ExtraAdam with averaging (and without using spectral normalization). Contrary to the previous experiments of the paper, the hyperparameter search was less exhaustive (due to time reason) but a similar time budget was spent for fine tuning each algorithm. We observed that with quite few hyperparameter tuning it was possible to match the state of the art with ExtraAdam. We also observed that ExtraAdam is less sensitive to the choice of learning rate, making the hyperparameter tuning easier and enabling the use of higher learning rate.
>
> [1] Balduzzi, D., Racaniere, S., Martens, J., Foerster, J., Tuyls, K., & Graepel, T. The Mechanics of n-Player Differentiable Games. ICML 2018

---

### Official Review · AnonReviewer3 · 2018-11-05
**Principled optimization for GANS**

**Rating:** 8
**Confidence:** 3

**Review:**

Summary:
The authors take a variational inequality perspective to the study of the saddle point problem that defines a GAN. By doing so, they are able to profit from the corresponding literature and propose a few methods that are variants of SGD. The authors show in a simple example (a bilinear function) these exhibit better performance than Adam and a basic gradient method. After showing theoretical guarantees of these methods (linear convergence) the authors propose to combine them with existing techniques, and show in fact this leads to better results.

Evaluation
This is a very good paper and I cannot but recommend its acceptance:
It is clear and well written.
It has the right level of balance between theory and experiments.
Theoretical results are far from trivial.
I haven't seen something similar.
The authors's do not make overstatements: they do not claim to have solved the GAN problem, but they do report improvements which are due to a thorough analysis (see above points). These results are much appreciated.

---

> ### Author Response · Authors · 2018-11-17
> **Thank you for the positive comments**
>
> The authors first would like to thank AnonReviewer3 for his careful evaluation and his detailed comments.
> We would like to point out that we addressed the points raised by AnonReviewer2  in our updated version, particularly, we tried our methods to train a ResNet architecture with the WGAN-GP objective (see the experimental section 7.2, table 1 and figure 4 for the new results).  After few days of experiments, we were able to match the current state-of-the art results of 8.2 on this architecture by using ExtraAdam with averaging (and without using spectral normalization). A similar time budget was spent for fine tuning each algorithm (SimAdam, AltAdam1, AltAdam5, ExtraAdam). We observed that with quite few hyperparameter tuning it was possible to match the state of the art with ExtraAdam. We also observed that ExtraAdam is less sensitive to the choice of learning rate, making the hyperparameter tuning easier and enabling the use of higher learning rate.

---

### Public Comment · (anonymous) · 2018-11-13
**an interesting perspective and missing important references**

It is an interesting perspective for training GAN.

I would like to point out several important references that the paper is missing regarding the theoretical contributions of this work.

1. the linear convergence for strongly monotone VI has been proved by Nesterov in 2011, though for a different algorithm.
Yurii Nesterov and Laura Scrimali. Solving strongly monotone variational and quasi-variational inequalities. Discrete and Continuous Dynamical Systems - A, 2011.

2. the idea of using one gradient in the extragradient method has been used in online optimization algorithms, e.g.,
 Chiang, C.K., Yang, T., Lee, C.J., Mahdavi, M., Lu, C.J., Jin, R., Zhu, S.: Online optimization with gradual variations. In: COLT 2012.

---

> ### Author Response · Authors · 2018-11-17
> **Response to "an interesting perspective and missing important references"**
>
> Hello,
> first of all we would like to thank this anonymous reader for his interest on the paper. We agree that [Chiang et al. 2012] is a relevant and we will consider to incorporate it in the revision.
>
> However, note that we already mentioned in our paper a more general or a more seminal related work:
>
> 1.
> We are aware of existing convergence proof for strongly monotone VI: We actually mention in Section 3 a seminal work on strongly monotone VIPs: “These iterates are known to converge linearly under an additional assumption on the operator\footnote{ Strong monotonicity, a generalization of strong convexity. See §A.} (Chen and Rockafellar, 1997)”.
>
> As you pointed out, Nesterov and Scrimali (2011) consider another algorithm. The Forward-Backward algorithm presented in (Chen and Rockafellar, 1997)  is another denomination (a bit more general though) for what we called “gradient method” (in Section 3). The Forward-Backward algorithm is more related to our work than Nesterov and Scrimali’s method is. Moreover, the proof of linear convergence of what we called “extrapolation from the past” algorithm (Theorem 1) is non trivial and, to our knowledge, does not directly extend from any existing work.
>
> 2.
> We mention right after (21), “ This update scheme can be related to the optimistic mirror descent (Rakhlin and Sridharan, 2013)”. Rakhlin and Sridharan (2013) explain in the beginning of Section 2 that “[they] exhibit a Mirror Descent type method which can be seen as a generalization of the recent algorithm of [9]” [9] being (Chiang et al. 2012).
>
> As developed in our paper right after the definition of “extrapolation from the past” (and pointed out by the anonymous reviewers) we are bringing a new perspective on this method: “However our technique comes from a different perspective, it was motivated by VIP and inspired from the extragradient method” and “ Using the VIP point of view we are able to prove a linear convergence rate for a projected version of the extrapolation from the past (see details and proof of Theorem 1 in §B.3). We also extend these results to the stochastic operator setting in §4”.

---

### Public Comment · (anonymous) · 2018-12-19
**Some concerns**

I have some concerns about this paper:

(1) In the theory side, this paper assumes that the loss function F is \mu-strongly monotone (or equivalently convex-concave), which seems to be a very unrealistic assumption as GANs are highly non-convex. Besides, many prior works on GANs have analyzed the convergence (w.r.t. regret bound) based on the convex-concave assumption, such as [1,2], which seems to make the theoretical contribution of this work even more limited. Finally, the authors argued that higher variance in the regret bounds implies worse convergence behavior, which is not quite convincing to me, as their actual convergence rates are the same though.

(2) In the algorithm side, this paper proposed to use extrapolation from the past, which is exactly the same with [3] in the unconstrained case. The authors argued that their proposed technique also works in the *constrained* case, but from the nonexpansive property of projection, it seems to be very straightforward (or trivial) to extend the unconstrained case to the constrained one.

(3) In the experimental side, it seems that the proposed "extrapolation from the past" algorithm does not improve the performance in the most time, by comparing "PastExtraAdam" with "SimAdam" or "AltAdam" in Table 1, Figure 5 and Figure 6. Does it contradict to your theory that  "extrapolation from the past" has smaller variance term in the regret bound, which implies a better convergence behavior?


[1] https://openreview.net/pdf?id=Skj8Kag0Z (Stabilizing Adversarial Nets with Prediction Methods)
[2] https://arxiv.org/pdf/1705.07215.pdf (On Convergence and Stability of GANs)
[3] https://openreview.net/forum?id=SJJySbbAZ (Training GANs with Optimism)

---

### Meta-Review · Area_Chair1 · 2018-12-15
**Unanimous Accept by Reviewers.**

**Confidence:** 5
**Recommendation:** Accept (Poster)

**Metareview:**

The paper presents a variational inequality perspective on the optimization problem arising in GANs. Convergence of stochastic gradient descent methods (averaging and extragradient variants) is given under monotonicity (or convex) assumptions. In particular, binlinear saddle point problem is carefully studied with batch and stochastic algorithms. Experiments on CIFAR10 with WGAN etc. show that the proposed averaging and extrapolation techniques improve the GAN training in such a nonconvex optimization practices.

General convergence results in the context of general non-monotone VIPs is still an open problem for future exploration. The questions raised by the reviewers are well answered. The reviewers unanimously accept the paper for ICLR publication.